# The Complexity of Bayesian Network Learning: Revisiting the Superstructure

**Robert Ganian and Viktoriia Korchemna**
Algorithms and Complexity Group, TU Wien
{rganian,vkorchemna}@ac.tuwien.ac.at

## Abstract

We investigate the parameterized complexity of Bayesian Network Structure Learning (BNSL), a classical problem that has received significant attention in empirical but also purely theoretical studies. We follow up on previous works that have analyzed the complexity of BNSL w.r.t. the so-called *superstructure* of the input. While known results imply that BNSL is unlikely to be fixed-parameter tractable even when parameterized by the size of a vertex cover in the superstructure, here we show that a different kind of parameterization—notably by the size of a feedback edge set—yields fixed-parameter tractability. We proceed by showing that this result can be strengthened to a localized version of the feedback edge set, and provide corresponding lower bounds that complement previous results to provide a complexity classification of BNSL w.r.t. virtually all well-studied graph parameters.

We then analyze how the complexity of BNSL depends on the representation of the input. In particular, while the bulk of past theoretical work on the topic assumed the use of the so-called *non-zero representation*, here we prove that if an *additive representation* can be used instead then BNSL becomes fixed-parameter tractable even under significantly milder restrictions to the superstructure, notably when parameterized by the treewidth alone. Last but not least, we show how our results can be extended to the closely related problem of Polytree Learning.

## 1   Introduction

Bayesian networks are among the most prominent graphical models for probability distributions. The key feature of Bayesian networks is that they represent conditional dependencies between random variables via a directed acyclic graph; the vertices of this graph are the variables, and an arc $ab$ means that the distribution of variable $b$ depends on the value of $a$. One beneficial property of Bayesian networks is that they can be used to infer the distribution of random variables in the network based on the values of the remaining variables.

The problem of constructing a Bayesian network with an optimal network structure is NP-hard, and remains NP-hard even on highly restricted instances [4]. This initial negative result has prompted an extensive investigation of the problem's complexity, with the aim of identifying new tractable fragments as well as the boundaries of its intractability [26, 32, 27, 22, 11, 6, 19]. The problem—which we simply call BAYESIAN NETWORK STRUCTURE LEARNING (BNSL)—can be stated as follows: given a set of $V$ of variables (represented as vertices), a family $\mathcal{F}$ of *score functions* which assign each variable $v \in V$ a score based on its *parents*, and a target value $\ell$, determine if there exists a directed acyclic graph over $V$ that achieves a total score of at least $\ell^1$.

---

[1]Formal definitions are provided in Section 2. We consider the decision version of BNSL for complexity-theoretic reasons only; all of the provided algorithms are constructive and can output a network as a witness.

35th Conference on Neural Information Processing Systems (NeurIPS 2021).

To obtain a more refined understanding of the complexity of BNSL, past works have analyzed the problem not only in terms of classical complexity but also from the perspective of *parameterized complexity* [9, 5]. In parameterized complexity analysis, the tractability of problems is measured with respect to the input size $n$ and additionally with respect to a specified numerical *parameter $k$*. In particular, a problem that is NP-hard in the classical sense may—depending on the parameterization used—be *fixed-parameter tractable* (FPT), which is the parameterized analogue of polynomial-time tractability and means that a solution can be found in time $f(k) \cdot n^{\mathcal{O}(1)}$ for some computable function $f$, or W[1]-*hard*, which rules out fixed-parameter tractability under standard complexity assumptions. The use of parameterized complexity as a refinement of classical complexity is becoming increasingly common and has been employed not only for BNSL [26, 32, 27], but also for numerous other problems arising in the context of neural networks and artificial intelligence [13, 41, 10, 16].

Unfortunately, past complexity-theoretic works have shown that BNSL is a surprisingly difficult problem. In particular, not only is the problem NP-hard, but it remains NP-hard even when asking for the existence of extremely simple networks such as directed paths [29] and is W[1]-hard when parameterized by the *vertex cover number* of the network [27]. In an effort to circumvent these lower bounds, several works have proposed to instead consider restrictions to the so-called *superstructure*, which is a graph that, informally speaking, captures all potential dependencies between variables [42, 35]. Ordyniak and Szeider [32] studied the complexity of BNSL when parameterized by the structural properties of the superstructure, and showed that parameterizing by the *treewidth* [36] of the superstructure is sufficient to achieve a weaker notion of tractability called XP-*tractability*. However, they also proved that BNSL remains W[1]-hard when parameterized by the treewidth of the superstructure [32, Theorem 3].

**Contribution.** Up to now, no "implicit" restrictions of the superstructure were known to lead to a fixed-parameter algorithm for BNSL alone. More precisely, the only known fixed-parameter algorithms for the problem require that we place explicit restrictions on either the sought-after network or the parent sets on the input: BNSL is known to be fixed-parameter tractable when parameterized by the number of arcs in the target network [22], the treewidth of an "*extended superstructure graph*" which also bounds the maximum number of parents a variable can have [26], or the number of parent set candidates plus the treewidth of the superstructure [32]. Moreover, a closer analysis of the reduction given by Ordyniak and Szeider [32, Theorem 3] reveals that BNSL is also W[1]-hard when parameterized by the *treedepth*, *pathwidth*, and even the *vertex cover number* of the superstructure alone. The vertex cover number is equal to the vertex deletion distance to an edgeless graph, and hence their result essentially rules out the use of the vast majority of graph parameters; among others, any structural parameter based on vertex deletion distance.

As our first conceptual contribution, we show that a different kind of graph parameters—notably, parameters that are based on edge deletion distance—give rise to fixed-parameter algorithms for BNSL in its full generality, without requiring any further explicit restrictions on the target network or parent sets. Our first result in this direction concerns the *feedback edge number* (fen), which is the minimum number of edges that need to be deleted to achieve acyclicity. In Theorem 3 we show not only that BNSL is fixed-parameter tractable when parameterized by the fen of the superstructure, but also provide a polynomial-time preprocessing algorithm that reduces any instance of BNSL to an equivalent one whose number of variables is linear in the fen (i.e., a *kernelization* [9, 5]).

Since fen is a highly "restrictive" parameter—its value can be large even on simple superstructures such as collections of disjoint cycles—we proceed by asking whether it is possible to lift fixed-parameter tractability to a more relaxed way of measuring distance to acyclicity. For our second result, we introduce the *local feedback edge number* (lfen), which intuitively measures the maximum edge deletion distance to acyclicity for cycles intersecting any particular vertex in the superstructure. In Theorem 6, we show that BNSL is also fixed-parameter tractable when prameterized by lfen; we also show that this comes at the cost of BNSL not admitting any polynomial-time preprocessing procedure akin to Theorem 3 when parameterized by lfen. We conclude our investigation in the direction of parameters based on edge deletion distance by showing that BNSL parameterized by *treecut width* [28, 45, 14], a recently discovered edge-cut based counterpart to treewidth, remains W[1]-hard (Theorem 9). An overview of these complexity-theoretic results is provided in Figure 1.

As our second conceptual contribution, we show that BNSL becomes significantly easier when one can use an *additive representation* of the scores rather than the *non-zero representation* that was considered in the vast majority of complexity-theoretic works on BNSL to date [26, 32, 27, 22, 11, 19].

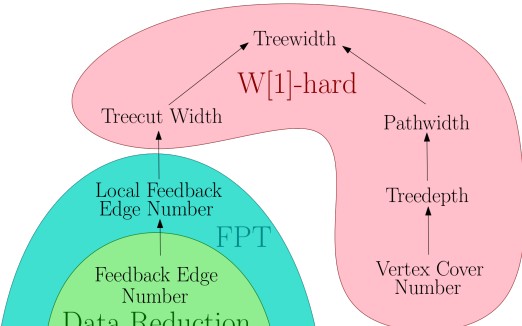

Figure 1: The complexity landscape of BNSL with respect to parameterizations of the super-structure. Arrows point from more restrictive parameters to less restrictive ones. Results for the three graph parameters on the left side follow from this paper, while all other W[1]-hardness results follow from the reduction by Ordyniak and Szeider [32, Theorem 3].

The additive representation is inspired by known heuristics for BNSL [40, 39] and utilizes a succinct encoding of the score function which assumes that the scores for parent sets can be decomposed into a sum of the scores of individual variables in the parent set; a discussion and formal definitions are provided in Section 2. In Theorem 11, we show that if the additive representation can be used, BNSL becomes fixed-parameter tractable when parameterized by the treewidth of the superstructure (and hence under every parameterization depicted in Figure 1). Motivated by the empirical usage of the additive representation, we also consider the case where we additionally impose a bound $q$ on the number of parents a vertex can accept; we show that the result of Theorem 11 also covers this case if $q$ is taken as an additional parameter, and otherwise rule out fixed-parameter tractability using an intricate reduction (Theorem 12).

For our third and final conceptual contribution, we show how our results can be adapted for the emergent problem of POLYTREE LEARNING (PL), a variant of BNSL where we require that the network forms a polytree. The crucial advantage of such networks is that they allow for a more efficient solution of the inference task [34, 23], and the complexity of PL has been studied in several works [21, 19, 38]. We show that all our results for BNSL can be adapted to PL, albeit in some cases it is necessary to perform non-trivial modifications. Furthermore, we observe that unlike BNSL, PL becomes polynomial-time tractable when the additive representation is used (Observation 16); this matches the "naive" expectation that learning simple networks would be easier than BNSL in its full generality. As our concluding result, we show that this expectation is in fact not always validated: while PL was recently shown to be W[1]-hard when parameterized by the number of so-called *dependent vertices* [21], in Theorem 17 we prove that BNSL is fixed-parameter tractable under that same parameterization.

## 2   Preliminaries

For an integer $i$, we let $[i] = \{1, 2, \ldots, i\}$ and $[i]_0 = [i] \cup \{0\}$. We denote by $\mathbb{N}$ the set of natural numbers, by $\mathbb{N}_0$ the set $\mathbb{N} \cup \{0\}$. We refer to the handbook by Diestel [8] for standard terminology on directed as well as undirected graphs. The *skeleton* (sometimes called the *underlying undirected graph*) of a directed graph (a *digraph*) $D = (V, A)$ is the undirected graph $G' = (V, E)$ such that $vw \in E$ if $vw \in A$ or $wv \in A$. A digraph is a *polytree* if its skeleton is a forest.

When comparing two numerical parameters $\alpha, \beta$ of graphs, we say that $\alpha$ is more *restrictive* than $\beta$ if there exists a function $f$ such that $\beta(G) \leq f(\alpha(G))$ holds for every graph $G$. We refer to the standard sources for the fundamentals of parameterized complexity, including the definitions of *fixed-parameter tractability*, *parameterized reductions*, W[1]-*hardness* and *treewidth* [5, 9, 31].

**Problem Definitions.**   Let $V$ be a set of vertices and $\mathcal{F} = \{ f_v : 2^{V \setminus \{v\}} \to \mathbb{N}_0 \mid v \in V \}$ be a family of *local score functions*. For a digraph $D = (V, A)$, we define its score as follows: $\mathtt{score}(D) = \sum_{v \in V} f_v(P_D(v))$, where $P_D(v)$ is the set of vertices of $D$ with an outgoing arc to $v$ (i.e., the *parent set* of $v$ in $D$). We can now formalize our problem of interest [32, 22].

---

BAYESIAN NETWORK STRUCTURE LEARNING (BNSL)

Input:      A set $V$ of vertices, a family $\mathcal{F}$ of local score functions, and an integer $\ell$.
Question:   Does there exist an acyclic digraph $D = (V, A)$ such that $\mathtt{score}(D) \geq \ell$?

---

POLYTREE LEARNING (PL) is defined analogously, with the only difference that there $D$ is addition-ally required to be a polytree [21]. We call $D$ a *solution* for the given instance.

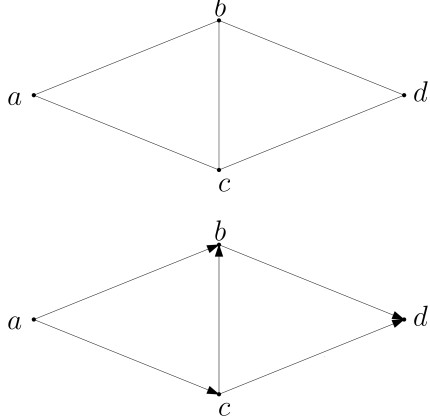

Figure 2: Example of a superstructure graph (on the top) and a suitable solution DAG (on the bottom) when:
$f_a(\{b\}) = f_a(\{c\}) = 1, f_a(\{b,c\}) = 2;$
$f_b(\{a\}) = f_b(\{c\}) = 1, f_b(\{a,c\}) = 3;$
$f_c(\{a\}) = 3, f_c(\{b\}) = 2;$
$f_d(\{b,c\}) = 1;$
$l = 6.$
Note that in the depicted DAG the scores of $b$ and $c$ are equal to 3, the score of $d$ is equal to 1. Parent set of $a$ is empty; as we assume the non-zero representation and $f_a(\emptyset)$ is not specified in the input, we conclude that $f_a(\emptyset) = 0$. Therefore the total score is $0 + 3 + 3 + 1 = 7 \geq 6 = l$.

Since both $V$ and $\mathcal{F}$ are assumed to be given on the input of our problems, an issue that arises here is that an explicit representation of $\mathcal{F}$ would be exponentially larger than $|V|$. A common way to potentially circumvent this is to use a *non-zero representation* of the family $\mathcal{F}$, i.e., where we only store values for $f_v(P)$ that are different than zero. This model has been used in the vast majority of works studying the complexity of BNSL and PL [26, 32, 27, 22, 19, 21]. Let $\Gamma_f(v)$ be the collection of candidate parents of $v$ which yield a non-zero score; formally, $\Gamma_f(v) = \{ Z \mid f_v(Z) \neq 0 \}$, and the input size $|\mathcal{I}|$ of an instance $\mathcal{I} = (V, \mathcal{F}, \ell)$ is simply defined as $|V| + \ell + \sum_{v \in V, P \in \Gamma_f(v)} |P|^2$.

A natural way to think about and exploit the structure of inter-variable dependencies of an instance $\mathcal{I}$ is to consider its *superstructure graph* $G_\mathcal{I} = (V, E)$, where $ab \in E$ if $a$ occurs in at least one candidate set in $\Gamma_f(b)$ (or vice-versa). An example is provided in Figure 2.

Naturally, families of local score functions may be exponentially larger than $|V|$ even when stored using the non-zero representation. In this paper, we also consider a second representation of $\mathcal{F}$ which is guaranteed to be polynomial in $|V|$: in the *additive representation*, we require that for every vertex $v \in V$ and set $Q = \{q_1, \ldots, q_m\} \subseteq V \setminus \{v\}$, $f_v(Q) = f_v(\{q_1\}) + \cdots + f_v(\{q_m\})$. Hence, each cost function $f_v$ can be fully characterized by storing at most $|V|$-many entries of the form $f_v(x) := f_v(\{x\})$ for each $x \in V \setminus \{v\}$. To avoid overfitting, one may optionally impose an additional constraint: an upper bound $q$ on the size of any parent set in the solution.

While not every family of local score functions admits an additive representation, the additive model is similar in spirit to the models used by some practical algorithms for BNSL. For instance, the algorithms of Scanagatta, de Campos, Corani and Zaffalon [40, 39], which can process BNSL instances with up to thousands of variables, approximate the real score functions by adding up the known score functions for two parts of the parent set and applying a small, logarithmic correction. Both of these algorithms also use the aforementioned bound $q$ for the parent set size. In spite of this connection to practice and the representation's streamlined nature, we are not aware of any prior works that considered the additive representation in complexity-theoretic studies of BNSL and PL. The superstructure graph for the additive representation can be defined in an analogous way as for the non-zero representation: an edge $uv$ simply captures a "suspected dependencey" between variables $u$ and $v$ (i.e., one receives a positive score for depending on the other).

To distinguish between these models, we use $\text{BNSL}^{\neq 0}$, $\text{BNSL}^+$, and $\text{BNSL}^+_\leq$ to denote BAYESIAN NETWORK STRUCTURE LEARNING with the non-zero representation, the additive representation, and the additive representation and the parent set size bound $q$, respectively (and analogously for PL).

In our algorithmic results, we will often use $G = (V, E)$ to denote the superstructure graph of the input instance $\mathcal{I}$. Without any loss of generality, we will also assume that $G$ is connected.

**Graph Parameters Based on Edge Cuts.** Traditionally, the bulk of graph-theoretic research on structural parameters has focused on parameters that guarantee the existence of small vertex separators in the graph; these are inherently tied to the theory of *graph minors* [37, 36] and the vertex deletion

---

[2]We remark that the non-zero representation could be strengthened even further by omitting each parent set $Z$ of $v$ which contains a proper subset $Z'$ such that $f_v(Z) \leq f_v(Z')$. This preprocessing step, however, does not have an impact on any of the results presented in this paper.

distance. This approach gives rise not only to the classical notion of treewidth, but also to its well-known restrictions and refinements such as *pathwidth* [37], *treedepth* [30] and the *vertex cover number* [12, 25]. The vertex cover number is the most restrictive parameter in this hierarchy.

However, there are numerous problems of interest that remain intractable even when parameterized by the vertex cover number. A recent approach developed for attacking such problems has been to consider parameters that guarantee the existence of small edge cuts in the graph; these are typically based on the edge deletion distance or, more broadly, tied to the theory of *graph immersions* [45, 28]. The parameter of choice for the latter is *treecut width* (tcw) [45, 28, 14, 15], a counterpart to treewidth which has been successfully used to tackle some problems that remained intractable when parameterized by the vertex cover number [17].

On the other hand, the by far most prominent parameter based on edge deletion distance is the *feedback edge number* of a connected graph $G = (V, E)$, which is the minimum cardinality of a set $F \subseteq E$ of edges (called the *feedback edge set*) such that $G - F$ is acyclic. The feedback edge number can be computed in quadratic time and has primarily been used to obtain fixed-parameter algorithms and polynomial kernels for problems where other parameterizations failed [17, 2, 1, 44].

Up to now, these were the only two edge-cut based graph parameters that have been considered in the broader context of algorithm design. This situation could be seen as rather unstisfactory in view of the large gap between the complexity of the richer class of graphs of bounded treecut width, and the significantly simpler class of graphs of bounded feedback edge number—for instance, the latter class is not even closed under disjoint union. Here, we propose a new parameter that lies "between" the feedback edge number and treecut width, and which can be seen as a localized relaxation of the feedback edge number: instead of measuring the total size of the feedback edge set, it only measures how many feedback edges can "locally interfere with" any particular part of the graph.

Formally, for a connected graph $G = (V, E)$ and a spanning tree $T$ of $G$, let the *local feedback edge set* at $v \in V$ be $E_{\mathrm{loc}}^T(v) = \{uw \in E \setminus E(T) \mid$ the unique path between $u$ and $w$ in $T$ contains $v\}$. The *local feedback edge number of* $(G, T)$ (denoted $\mathrm{lfen}(G, T)$) is then equal to $\max_{v \in V} |E_{\mathrm{loc}}^T(v)|$, and the *local feedback edge number of* $G$ is simply the smallest local feedback edge number among all possible spanning trees of $G$, i.e., $\mathrm{lfen}(G) = \min_{T \text{ is a spanning tree of } G} \mathrm{lfen}(G, T)$.

It is not difficult to show that the local feedback edge number is "sandwiched" between the feedback edge number and treecut width. We also show that computing it is FPT.

**Proposition 1.** *For every graph $G$, $\mathrm{tcw}(G) \leq \mathrm{lfen}(G) + 1$ and $\mathrm{lfen}(G) \leq \mathrm{fen}(G)$.*

**Theorem 2.** *The problem of determining whether $\mathrm{lfen}(G) \leq k$ for an input graph $G$ parameterized by an integer $k$ is fixed-parameter tractable. Moreover, if the answer is positive, we may also output a spanning tree $T$ such that $\mathrm{lfen}(G, T) \leq k$ as a witness.*

# 3 Solving BNSL$^{\neq 0}$ with Parameters Based on Edge Cuts.

In this section we provide tractability and lower-bound results for BNSL$^{\neq 0}$ from the viewpoint of superstructure parameters based on edge cuts. Together with the previous lower bound that rules out fixed-parameter algorithms based on all vertex-separator parameters [32, Theorem 3], the results presented here provide a comprehensive picture of the complexity of BNSL$^{\neq 0}$ with respect to superstructure parameterizations.

**Using the Feedback Edge Number for BNSL$^{\neq 0}$.** We say that two instances $\mathcal{I}, \mathcal{I}'$ of BNSL are *equivalent* if (1) they are either both Yes-instances or both No-instances, and furthermore (2) a solution to one instance can be transformed into a solution to the other instance in polynomial time. Our aim here is to prove the following theorem:

**Theorem 3.** *There is an algorithm which takes as input an instance $\mathcal{I}$ of BNSL$^{\neq 0}$ whose superstructure has $\mathrm{fen}$ $k$, runs in time $\mathcal{O}(|\mathcal{I}|^2)$, and outputs an equivalent instance $\mathcal{I}' = (V', \mathcal{F}', \ell')$ of BNSL$^{\neq 0}$ such that $|V'| \leq 16k$.*

In parameterized complexity theory, such data reduction algorithms with performance guarantees are called *kernelization algorithms* [9, 5]. These may be applied as a polynomial-time preprocessing step before, e.g., more computationally expensive methods are used. The fixed-parameter tractability of BNSL$^{\neq 0}$ when parameterized by the $\mathrm{fen}$ of the superstructure follows as an immediate corollary of Theorem 3 (one may solve $\mathcal{I}$ by, e.g., exhaustively looping over all possible DAGs on $V'$ via a

brute-force procedure). We also note that even though the number of variables of the output instance is polynomial in the parameter $k$, the instance $\mathcal{I}'$ need not have size polynomial in $k$.

We begin our path towards a proof of Theorem 3 by computing a feedback edge set $E_F$ of $G$ of size $k$ in time $\mathcal{O}(|\mathcal{I}|^2)$ by, e.g., Prim's algorithm. Let $T$ be the spanning tree of $G$, $E_F = E(G) \setminus E(T)$. The algorithm will proceed by the recursive application of certain reduction rules, which are polynomial-time operations that alter ("simplify") the input instance in a certain way. A reduction rule is *safe* if it outputs an instance which is equivalent to the input instance. We start by describing a rule that will be used to prune $T$ until all leaves are incident to at least one edge in $E_F$.

**Reduction Rule 1.** *Let $v \in V$ be a vertex and let $Q$ be the set of neighbors of $v$ with degree 1 in $G$. We construct a new instance $\mathcal{I}' = (V', \mathcal{F}', \ell)$ by setting:* **1.** $V' := V \setminus Q$; **2.** $\Gamma_{f'}(v) := \{\emptyset\} \cup \{ (P \setminus Q) \mid P \in \Gamma_f(v) \}$; **3.** *for all $w \in V' \setminus \{v\}$, $f'_w = f_w$;* **4.** *for every $P' \in \Gamma_{f'}(v)$:*

$$f'_v(P') := \max_{P : P \setminus Q = P'} \Big( f_v(P) + \sum_{v_{in} \in P \cap Q} f_{v_{in}}(\emptyset) + \sum_{v_{out} \in Q \setminus P} \max(f_{v_{out}}(\emptyset), f_{v_{out}}(v)) \Big).$$

**Lemma 4.** *Reduction Rule 1 is safe.*

Observe that the superstructure graph $G'$ obtained after applying one step of Reduction Rule 1 is simply $G - Q$; after its exhaustive application we obtain an instance $\mathcal{I}$ such that all the leaves of the tree $T$ are endpoints of $E_F$. Our next step is to get rid of long paths in $G$ whose internal vertices have degree 2. We note that this step is more complicated than in typical kernelization results using feedback edge set as the parameter, since a directed path $Q$ in $G$ can serve multiple "roles" in a hypothetical solution $D$ and our reduction gadget needs to account for all of these. Intuitively, $Q$ may or may not appear as a directed path in $D$ (which impacts what other arcs can be used in $D$ due to acyclicity), and in addition the total score achieved by $D$ on the internal vertices of $Q$ needs to be preserved while taking into account whether the endpoints of $Q$ have a neighbor in the path or not. Because of this (and unlike in many other kernelization results of this kind [17, 43, 15]), we will not be replacing $Q$ merely by a shorter path, but by a more involved gadget. An illustration is provided in Figure 3.

**Reduction Rule 2.** *Let $a, b_1, \ldots, b_m, c$ be a path in $G$ such that for each $i \in [m]$, $b_i$ has degree precisely 2, and let $P = \{b_1, \ldots, b_m\}$. We construct a new instance $\mathcal{I}' = (V', \mathcal{F}', \ell)$ as follows:*

1. $V' := V \cup \{b\} \setminus \{b_2 \ldots b_{m-1}\}$;

2. $\Gamma_{f'}(b) = \{B \cup \{b_1, b_m\} \mid B \subseteq \{a, c\}\}$ *where $f'_b(B \cup \{b_1, b_m\})$ is equal to the maximum score that can be achieved by $P$ if $B$ are used as parents;*

3. *The scores for $a$ and $c$ are obtained from $\mathcal{F}$ by simply adding $b$ to any parent set containing either $b_1$ or $b_m$;*

4. $\Gamma_{f'}(b_1)$ *contains only $\{a, b, b_m\}$ with score equal to the maximum score that can be achieved by $P$ if $a$ is used as a parent but there is no path from $a$ to $b_m$ via $P$;*

5. $\Gamma_{f'}(b_m)$ *contains only $\{c, b, b_1\}$ with score equal to the maximum score that can be achieved by $P$ if $c$ is used as a parent but there is no path from $c$ to $b_1$ via $P$;*

6. *for all $w \in V' \setminus \{a, b_1, b, b_m, c\}$, $f'_w = f_w$.*

**Lemma 5.** *Reduction Rule 2 is safe.*

*Proof of Theorem 3.* We begin by exhaustively applying Reduction Rule 1 on an instance whose superstructure graph has a feedback edge set of size $k$, which results in an instance with the same feedback edge set but whose spanning tree $T$ has at most $2k$ leaves. It follows that there are at most $2k$ vertices with a degree greater than 2 in $T$.

Let us now "mark" all the vertices that either are endpoints of the edges in $E_F$ or have a degree greater then 2 in $T$; the total number of marked vertices is upper-bounded by $4k$. We now proceed to the exhaustive application of Reduction Rule 2, which will only be triggered for sufficiently long paths in $T$ that connect two marked vertices but contain no marked vertices on its internal vertices; there are at most $4k$ such paths due to the tree structure of $T$. Reduction Rule 2 will replace each such path with

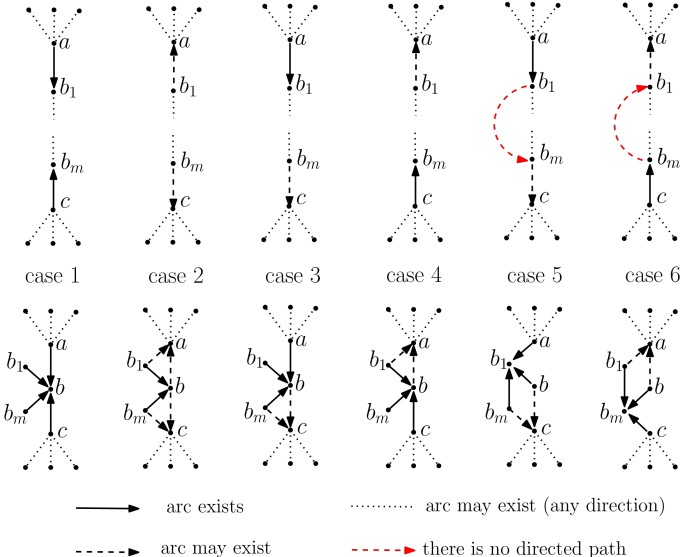

Figure 3:
Top: The six possible solution scenarios that may arise when dealing with long paths.
Bottom: The corresponding arcs in the gadget after the application of Reduction Rule 2.

a set of 3 vertices, and therefore after its exhaustive application we obtain an equivalent instance with at most $4k + 4k \cdot 3 = 16k$ vertices, as desired. Correctness follows from the safeness of Reduction Rules 1, 2, and the runtime bound follows by observing that the total number of applications of each rule as well as the runtime of each rule are upper-bounded by a linear function of the input size. $\square$

As an immediate corollary of Theorem 3, we can apply a standard brute-force branching procedure [33] to solve BNSL in time $n^{\mathcal{O}(1)} + 2^{\mathcal{O}(k)}$.

**Fixed-Parameter Tractability of $\mathrm{BNSL}^{\neq 0}$ using the Local Feedback Edge Number.** Our aim here will be to lift the fixed-parameter tractability of $\mathrm{BNSL}^{\neq 0}$ established by Theorem 3 by relaxing the parameterization to lfen. In particular, we will prove:

**Theorem 6.** $\mathrm{BNSL}^{\neq 0}$ *is fixed-parameter tractable when parameterized by the local feedback edge number of the superstructure.*

Since fen is a more restrictive parameter than lfen, this results in a strictly larger class of instances being identified as tractable. However, the means we will use to establish Theorem 6 will be fundamentally different: we will not use a polynomial-time data reduction algorithm as the one provided in Theorem 3, but instead apply a dynamic programming approach. Since the kernels constructed by Theorem 3 contain only polynomially-many variables w.r.t. fen, that result is incomparable to Theorem 6. In fact one can use standard techniques to prove that, under well-established complexity assumptions, a data reduction result such as the one provided in Theorem 3 *cannot* exist for lfen.

**Theorem 7.** *Unless* $\mathsf{NP} \subseteq \mathsf{co\text{-}NP/poly}$*, there is no polynomial-time algorithm which takes as input an instance $\mathcal{I}$ of $\mathrm{BNSL}^{\neq 0}$ whose superstructure has* lfen $k$ *and outputs an equivalent instance* $\mathcal{I}' = (V', \mathcal{F}', \ell')$ *of $\mathrm{BNSL}^{\neq 0}$ such that $|V'| \in k^{\mathcal{O}(1)}$. In particular, $\mathrm{BNSL}^{\neq 0}$ does not admit a polynomial kernel when parameterized by* lfen*.*

Towards proving Theorem 6, assume that we are given an instance $\mathcal{I} = (V, \mathcal{F}, \ell)$ of $\mathrm{BNSL}^{\neq 0}$ with connected superstructure graph $G = (V, E)$. Let $T$ be a fixed rooted spanning tree of $G$ such that $\mathrm{lfen}(G, T) = \mathrm{lfen}(G) = k$, denote the root by $r$. For $v \in V(T)$, let $T_v$ be the subtree of $T$ rooted at $v$, let $V_v = V(T_v)$, and let $\bar{V}_v = N_G(V_v) \cup V_v$. We define the *boundary* $\delta(v)$ of $v$ to be the set of endpoints of all edges in $G$ with precisely one endpoint in $V_v$ (observe that the boundary can never have a size of 1). Notice that $|\delta(v)| \leq 2k + 2$. We can now proceed to a definition of the records that will be used in our dynamic program. Intuitively, these records will be computed in a leaf-to-root fashion and will store at each vertex $v$ information about the best score that can be achieved by a partial solution that intersects the subtree rooted at $v$.

Let $R$ be a binary relation on $\delta(v)$ and $s$ an integer. For $s \in \mathbb{Z}$, we say that $(R : s)$ is a *record* for a vertex $v$ if and only if there exists a DAG $D$ on $\bar{V}_v$ such that (1) $w \in V_v$ for each arc $uw \in A(D)$, (2) $ab \in R$ if there exists an $a$-$b$ path in $D$, and (3) $s$ is the total score achieved by $D$ on vertices in $V_v$.

The records $(R, s)$ where $s$ is maximal for fixed $R$ are called *valid*. Denote the set of all valid records for $v$ by $\mathcal{R}(v)$, and note that $|\mathcal{R}(v)| \leq 2^{\mathcal{O}(k^2)}$.

Observe that if $v_i$ is a closed child of $v$, then $\mathcal{R}(v_i)$ consists of precisely two valid records: one for $R = \emptyset$ and one for $R = \{vv_i\}$. Moreover, the root $r$ of $T$ has only a single valid record $(\emptyset : s_{\mathcal{I}})$, where $s_{\mathcal{I}}$ is the maximum score that can be achieved by a solution in $\mathcal{I}$. The following lemma lies at the heart of our result and shows how we can compute our records in a leaf-to-root fashion along $T$.

**Lemma 8.** *Let $v \in V(G)$ have $m$ children in $T$ where $m > 0$, and assume we have computed $\mathcal{R}(v_i)$ for each child $v_i$ of $v$. Then $\mathcal{R}(v)$ can be computed in time at most $m \cdot |\Gamma_f(v)| \cdot 2^{\mathcal{O}(k^3)}$.*

To prove Theorem 6, we start by invoking Theorem 2 to obtain a spanning tree $T$ and then compute the records $\mathcal{R}(v)$ for each leaf of $T$ via exhaustive branching. We then apply Lemma 8 to propagate our record sets towards the root $r$ of $T$; once we obtain $\mathcal{R}(r)$, we can output a solution in constant time. The runtime of the dynamic programming procedure used in the proof of Theorem 6 is upper-bounded by $|\mathcal{I}|^3 \cdot 2^{\mathcal{O}(k^3)}$.

For our final result for this section, recall that lfen lies between fen and treecut width in the parameter hierarchy (see Proposition 1). Since BNSL$^{\neq 0}$ is FPT when parameterized by lfen, the next step would be to ask whether this tractability result can be lifted to treecut width. Below, we answer this question negatively via a reduction from the W[1]-hard MULTICOLORED CLIQUE problem [9, 5].

**Theorem 9.** BNSL$^{\neq 0}$*is* W[1]*-hard when parameterized by the treecut width of the superstructure.*

## 4 Additive Scores and Treewidth

While the previous section focused on the complexity of BNSL when the non-zero representation was used (i.e., BNSL$^{\neq 0}$), here we turn our attention to the complexity of the problem with respect to the additive representation. Recall from Subsection 2 that there are two variants of interest for this representation: BNSL$^+$ and BNSL$^+_{\leq}$. We begin by showing that, unsurprisingly, both of these are NP-hard. We do so by reducing from the classical MINIMUM FEEDBACK ARC SET problem [20, 7].

**Theorem 10.** BNSL$^+$ *is* NP*-hard. Moreover,* BNSL$^+_{\leq}$ *is* NP*-hard for every $q \geq 3$.*

While the use of the additive representation did not affect the classical complexity of BNSL, it makes a significant difference in terms of parameterized complexity. Indeed, in contrast to BNSL$^{\neq 0}$:

**Theorem 11.** BNSL$^+$ *is* FPT *when parameterized by the treewidth of the superstructure. Moreover,* BNSL$^+_{\leq}$ *is* FPT *when parameterized by $q$ plus the treewidth of the superstructure.*

*Proof Sketch.* As noted in the preliminaries, due to space constraints we refer to the usual books for a definition of *treewidth* and *nice tree-decompositions* [9, 5]. We begin by applying Bodlaender's algorithm [3, 24] to compute a nice tree-decomposition $(\mathcal{T}, \chi)$ of $G_{\mathcal{I}}$ of width $k = \text{tw}(G_{\mathcal{I}})$, whose vertices are called *nodes*. To prove the theorem, we will design a leaf-to-root dynamic programming algorithm which will compute and store a set of records at each node of $T$, whereas once we ascertain the records for $r$ we will have the information required to output a correct answer. Intuitively, the records will store all information about each possible set of arcs between vertices in each *bag*, along with relevant connectivity information provided by arcs between all vertices that are either in the current bag $t$ or in some descendant of $t$ (we denote the set of these vertices $\chi_t^{\downarrow}$), and information about the partial score. When solving BNSL$^+_{\leq}$, the records will also keep track of parent set sizes.

Formally, the records will have the following structure. For a node $t$, let $S(t) = \{(\text{loc}, \text{con}, \text{inn}) \mid \text{loc}, \text{con} \subseteq A_{\chi(t)}, \text{inn} : \chi(t) \to [q]_0\}$ be the set of *snapshots* of $t$. The record $\mathcal{R}_t$ of $t$ is then a mapping from $S(t)$ to $\mathbb{N}_0 \cup \{\bot\}$. Observe that $|S(t)| \leq 4^{k^2}(q+1)^k$. To introduce the semantics of our records, let $\Upsilon_t$ be the set of all directed acyclic graphs over the vertex set $\chi_t^{\downarrow}$ with maximal in-degree at most $q$, and let $D_t = (\chi_t^{\downarrow}, A)$ be a directed acyclic graph in $\Upsilon_t$. We say that the *snapshot of $D_t$ in $t$* is the tuple $(\alpha, \beta, p)$ where $\alpha = A \cap A_{\chi(t)}$, $\beta = \text{Con}(\chi(t), D_t)$ and $p$ (which is only used for BNSL$^+_{\leq}$) specifies numbers of parents of vertices from $\chi(t)$ in $D$, i.e., $p(v) = |\{w \in \chi_t^{\downarrow} | wv \in A\}|$, $v \in \chi(t)$. Now, for each snapshot $(\text{loc}, \text{con}, \text{inn}) \in S(t)$ we set $\mathcal{R}_t(\text{loc}, \text{con}, \text{inn}) = \bot$ if there exists no DAG in $\Upsilon_t$ with $(\text{loc}, \text{con}, \text{inn})$ as its snapshot, and otherwise we set $\mathcal{R}_t(\text{loc}, \text{con}, \text{inn})$ to the highest score that can be achieved by such a DAG.

The algorithm computes these records in a leaf-to-root fashion while traversing $\mathcal{T}$, which can be achieved in time at most $2^{\mathcal{O}(k^2)} \cdot q^{\mathcal{O}(k)} \cdot n$, where $n$ is the input size. Once we reach the root node $r$, we use the fact that $\chi(r) = \emptyset$ by the definition of nice tree-decompositions to simply check if $\mathcal{R}_r(\emptyset, \emptyset, \emptyset) \geq \ell$; the algorithm then outputs "Yes" if and only if this is the case. $\square$

This completely resolves the parameterized complexity of $\text{BNSL}^+$ w.r.t. all parameters depicted on Figure 1. However, the same is not true for $\text{BNSL}^+_{\leq}$: while a careful analysis of the algorithm provided in the proof of Theorem 11 reveals that $\text{BNSL}^+_{\leq}$ is XP-tractable when parameterized by the treewidth of the superstructure alone, it is not yet clear whether it is FPT—in other words, do we need to parameterize by both $q$ and treewidth to achieve fixed-parameter tractability? We conclude this section by answering this question affirmatively via an involved two-step reduction from a variant of the W[1]-hard MULTIDIMENSIONAL SUBSET SUM problem [18, 15].

**Theorem 12.** $\text{BNSL}^+_{\leq}$ *is* W[1]*-hard when parameterized by the treewidth of the superstructure.*

## 5 Implications for Polytree Learning

Here, we discuss how the results of Sections 3 and 4 can be adapted to POLYTREE LEARNING (PL).

**Theorem 3: Data Reduction.** Recall that the proof of Theorem 3 used two data reduction rules. While Reduction Rule 1 carries over to $\text{PL}^{\neq 0}$, Reduction Rule 2 has to be completely redesigned to preserve the (non-)existence of undirected paths between $a$ and $c$. By doing so, we obtain:

**Theorem 13.** *There is an algorithm which takes as input an instance $\mathcal{I}$ of $\text{PL}^{\neq 0}$ whose superstructure has feedback edge number $k$, runs in time $\mathcal{O}(|\mathcal{I}|^2)$, and outputs an equivalent instance $\mathcal{I}' = (V', \mathcal{F}', \ell')$ of $\text{PL}^{\neq 0}$ such that $|V'| \leq 24k$.*

**Theorem 6: Fixed-parameter tractability.** Analogously to $\text{BNSL}^{\neq 0}$ a data reduction procedure as the one provided in Theorem 13 does not exist for $\text{PL}^{\neq 0}$ parametrized by lfen unless NP $\subseteq$ co-NP/poly, since the lower-bound result provided in Theorem 7 can be straightforwardly adapted to $\text{PL}^{\neq 0}$. But similarly as for BNSL we can provide an FPT algorithm using the same ideas as in the proof of Theorem 6. The algorithm proceeds by dynamic programming on the spanning tree $T$ of $G$ with $\text{lfen}(G, T) = \text{lfen}(G) = k$. The records will, however, need to be modified: for each vertex $v$, instead of the path-connectivity relation on $\delta(v)$, we store connected components of the *inner boundary* $\delta(v) \cap V_v$ and incoming arcs to $T_v$. This yields:

**Theorem 14.** $\text{PL}^{\neq 0}$ *is fixed-parameter tractable when parameterized by the local feedback edge number of the superstructure.*

As for treecut width, we remark that a recent reduction for $\text{PL}^{\neq 0}$ [21, Theorem 4.2] immediately implies that the problem is W[1]-hard when parameterized by the treecut width.

**Theorem 11: Additive Representation.** We remark that, like $\text{BNSL}^+$ and $\text{BNSL}^+_{\leq}$, a simple reduction shows that $\text{PL}^+_{\leq}$ is NP-hard for a fixed value of $q$, in this case $q = 1$. Moreover, the dynamic programming algorithm for $\text{BNSL}^+_{\leq}$ parameterized by treewidth and $q$ can be adapted to also solve $\text{PL}^+_{\leq}$. The algorithm runs in time at most $2^{\mathcal{O}(k^2)} \cdot q^{\mathcal{O}(k)} \cdot |\mathcal{I}|$.

**Theorem 15.** $\text{PL}^+_{\leq}$ *is* FPT *when parameterized by $q$ plus the treewidth of the superstructure.*

The situation is, however, completely different for $\text{PL}^+$: unlike $\text{BNSL}^+$, this problem is in fact polynomial-time tractable. Indeed, it admits a simple reduction to the classical minimum edge-weighted spanning tree problem.

**Observation 16.** $\text{PL}^+$ *is polynomial-time tractable.*

This coincides with the intuitive expectation that learning simple, more restricted networks could be easier than learning general networks. We conclude our exposition with an example showcasing that this is not true in general when comparing PL to BNSL. Grüttemeier et al. [21] recently showed that $\text{PL}^{\neq 0}$ is W[1]-hard when parameterized by the number of *dependent vertices*, which are vertices with non-empty sets of candidate parents in the non-zero representation. For $\text{BNSL}^{\neq 0}$ we can show:

**Theorem 17.** $\text{BNSL}^{\neq 0}$ *is fixed-parameter tractable when parameterized by the number of dependent vertices.*

## 6   Concluding Remarks

Our results provide a new set of tractability results that counterbalance the previously established algorithmic lower bounds for BAYESIAN NETWORK STRUCTURE LEARNING and POLYTREE LEARNING on "simple" superstructures. In particular, even though the problems remain $\mathsf{W}[1]$-hard when parameterized by the vertex cover number of the superstructure [32, 21], we obtained fixed-parameter tractability and a data reduction procedure using the feedback edge number and its localized version. Together with our lower-bound result for treecut width, this completes the complexity map for BNSL w.r.t. virtually all commonly considered graph parameters of the superstructure. Moreover, we showed that if the input is provided with an additive representation instead of the non-zero representation considered in previous theoretical works, the problems admit a dynamic programming algorithm which guarantees fixed-parameter tractability w.r.t. the treewidth of the superstructure. We remark that all of our results assume that the score functions are provided explicitly; future work could also consider the behavior of the problem when these functions are supplied by a suitably defined oracle.

This theoretical work follows up on previous complexity studies of the considered problems, and as such we do not claim any immediate practical applications of the results. That being said, it would be interesting to see if the polynomial-time data reduction procedure introduced in Theorem 3 could be adapted and streamlined to allow for a speedup of previously introduced heuristics for the problem [40, 39], at least for some sets of instances. Finally, we believe that the *local feedback edge number* can be used to push the boundaries of tractability for other problems of interest as well.

**Acknowledgments.**   The authors acknowledge support by the Austrian Science Fund (FWF, projects P31336 and Y1329).

**Funding Transparency Statement.**   Funding in direct support of this work: FWF Project P31336, FWF Project Y1329.

**Declaration of Competing Interests.**   None of the authors have financial relationships with entities that could potentially be perceived to influence the content of the submitted work.

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
