graph terminology. In this paper, we will consider directed as well as undirected simple graphs. If $G = (V, E)$ is an undirected graph and $\{v, w\} \in E$, we will often use $vw$ as shorthand for $\{v, w\}$; we will also sometimes use $V(G)$ to denote its vertex set. Moreover, we let $N_G(v)$ denote the set of *neighbors* of $v$, i.e., $\{u \in V \mid vu \in E\}$. We extend this notation to sets as follows: $N_G(X) = \{u \in V \setminus X \mid \exists x \in X : ux \in E(G)$. For a set $X$ of vertices, let $A_X$ denote the set of all possible arcs over $X$.

If $D = (V, A)$ is a directed graph (i.e., a *digraph*) and $(v, w) \in A$, we will similarly use $vw$ as shorthand for $(v, w)$. We also let $P_D(v)$ denote the set of *parents* of $v$, i.e., $\{u \in V \mid uv \in A\}$ (there are sometimes called *in-neighbors* in the literature, while the notion of *out-neighbors* is defined analogously). In both cases, we may drop $G$ or $D$ from the subscript if the (di)graph is clear from the context. The *degree* of $v$ is $|N(v)|$, and for digraphs we use the notions of *in-degree* (which is equal to $|P(v)|$) and *out-degree* (the number of arcs originating from the given vertex).

The *skeleton* (sometimes called the *underlying undirected graph*) of a digraph $G = (V, A)$ is the undirected graph $G' = (V, E)$ such that $vw \in E$ if $vw \in A$ or $wv \in A$. A digraph is a *polytree* if its skeleton is a forest.

When comparing two numerical parameters $\alpha, \beta$ of graphs, we say that $\alpha$ is more *restrictive* than $\beta$ if there exists a function $f$ such that $\beta(G) \leq f(\alpha(G))$ holds for every graph $G$. In other words, $\alpha$ is

more restrictive than $\beta$ if and only if the following holds: whenever all graphs in some graph class $\mathcal{H}$ have $\alpha$ upper-bounded by a constant, all graphs in $\mathcal{H}$ also have $\beta$ upper-bounded by a constant. Observe that in this case a fixed-parameter algorithm parameterized by $\beta$ immediately implies a fixed-parameter algorithm parameterized by $\alpha$, while $\mathsf{W}[1]$-hardness behaves in the opposite way.

**Problem Definitions.** Let $V$ be a set of vertices and $\mathcal{F} = \{\, f_v : 2^{V \setminus \{v\}} \to \mathbb{N}_0 \mid v \in V \,\}$ be a family of *local score functions*. For a digraph $D = (V, A)$, we define its score as follows: $\texttt{score}(D) = \sum_{v \in V} f_v(P_D(v))$, where $P_D(v)$ is the set of vertices of $D$ with an outgoing arc to $v$ (i.e., the *parent set* of $v$ in $D$). We can now formalize our problem of interest [36, 25].

---

BAYESIAN NETWORK STRUCTURE LEARNING (BNSL)

| | |
|---|---|
| Input: | A set $V$ of vertices, a family $\mathcal{F}$ of local score functions, and an integer $\ell$. |
| Question: | Does there exist an acyclic digraph $D = (V, A)$ such that $\texttt{score}(D) \geq \ell$? |

---

POLYTREE LEARNING (PL) is defined analogously, with the only difference that there $D$ is additionally required to be a polytree [24]. We call $D$ a *solution* for the given instance.

Since both $V$ and $\mathcal{F}$ are assumed to be given on the input of our problems, an issue that arises here is that an explicit representation of $\mathcal{F}$ would be exponentially larger than $|V|$. A common way to potentially circumvent this is to use a *non-zero representation* of the family $\mathcal{F}$, i.e., where we only store values for $f_v(P)$ that are different than zero. This model has been used in a large number of works studying the complexity of BNSL and PL [29, 36, 30, 25, 22, 24] and is known to be strictly more general than, e.g., the bounded-arity representation where one only considers parent sets of arity bounded by a constant [36, Section 3]. Let $\Gamma_f(v)$ be the set of candidate parents of $v$ which yield a non-zero score; formally, $\Gamma_f(v) = \{\, Z \mid f_v(Z) \neq 0 \,\}$, and the input size $|\mathcal{I}|$ of an instance $\mathcal{I} = (V, \mathcal{F}, \ell)$ is simply defined as $|V| + \ell + \sum_{v \in V, P \in \Gamma_f(v)} |P|$.

Let $P_\rightarrow(v)$ be the set of all parents which appear in $\Gamma_f(v)$, i.e., $a \in P_\rightarrow(v)$ if and only if $\exists Z \in \Gamma_f(v) : a \in Z$. A natural way to think about and exploit the structure of inter-variable dependencies laid bare by the non-zero representation is to consider the *superstructure graph* $G_\mathcal{I} = (V, E)$ of a BNSL (or PL) instance $\mathcal{I} = (V, \mathcal{F}, \ell)$, where $ab \in E$ if and only if either $a \in P_\rightarrow(b)$, or $b \in P_\rightarrow(a)$, or both.

Naturally, families of local score functions may be exponentially larger than $|V|$ even when stored using the non-zero representation. In this paper, we also consider a second representation of $\mathcal{F}$ which is guaranteed to be polynomial in $|V|$: in the *additive representation*, we require that for every vertex $v \in V$ and set $Q = \{q_1, \ldots, q_m\} \subseteq V \setminus \{v\}$, $f_v(Q) = f_v(\{q_1\}) + \cdots + f_v(\{q_m\})$. Hence, each cost function $f_v$ can be fully characterized by storing at most $|V|$-many entries of the form $f_v(x) := f_v(\{x\})$ for each $x \in V \setminus \{v\}$. To avoid overfitting, one may optionally impose an additional constraint: an upper bound $q$ on the size of any parent set in the solution(or, equivalently, $q$ is a maximum upper-bound on the in-degree of the sought-after acyclic digraph $D$).

While not every family of local score functions admits an additive representation, the additive model is similar in spirit to the models used by some practical algorithms for BNSL. For instance, the algorithms of Scanagatta, de Campos, Corani and Zaffalon [43, 42], which can process BNSL instances with up to thousands of variables, approximate the real score functions by adding up the known score functions for two parts of the parent set and applying a small, logarithmic correction. Both of these algorithms also use the aforementioned bound $q$ for the parent set size. In spite of this connection to practice and the representation's streamlined nature, we are not aware of any prior works that considered the additive representation in complexity-theoretic studies of BNSL and PL.

As before, in the additive representation we will also only store scores for parents of $v$ which yield a non-zero score, and can thus define $P_\rightarrow(v) = \{\, z \mid f_v(z) \neq 0 \,\}$, as for the non-zero representation. This in turn allows us to define the superstructure graphs in an analogous way as before: $G_\mathcal{I} = (V, E)$ where $ab \in E$ if and only if $a \in P_\rightarrow(b)$, $b \in P_\rightarrow(a)$, or both.

To distinguish between these models, we use $\text{BNSL}^{\neq 0}$, $\text{BNSL}^+$, and $\text{BNSL}^+_\leq$ to denote BAYESIAN NETWORK STRUCTURE LEARNING with the non-zero representation, the additive representation, and the additive representation and the parent set size bound $q$, respectively. The same notation will also be used for POLYTREE LEARNING—for example, an instance of $\text{PL}^+_\leq$ will consist of $V$, a family

181 $\mathcal{F}$ of local score functions in the additive representation, and integers $\ell, q$, and the question is whether
182 there exists a polytree $D = (V, A)$ with in-degree at most $q$ and $\texttt{score}(D) \geq \ell$.

183 In our algorithmic results, we will often use $G = (V, E)$ to denote the superstructure graph of
184 the input instance $\mathcal{I}$. Without any loss of generality, we will also assume that $G$ is connected.
185 Indeed, given an algorithm $\mathbb{A}$ that solves BNSL on connected instances, we may solve disconnected
186 instances of BNSL by using $\mathbb{A}$ to find the maximum score $\ell_C$ for each connected component $C$ of $G$
187 independently, and we may then simply compare $\sum_{C \text{ is a connected component of } G} \ell_C$ with $\ell$.

**Parameterized Complexity.** In parameterized algorithmics [8, 12, 35] the running-time of an
189 algorithm is studied with respect to a parameter $k \in \mathbb{N}_0$ and input size $n$. The basic idea is to find
190 a parameter that describes the structure of the instance such that the combinatorial explosion can
191 be confined to this parameter. In this respect, the most favorable complexity class is FPT (*fixed-*
192 *parameter tractable*) which contains all problems that can be decided by an algorithm running in
193 time $f(k) \cdot n^{\mathcal{O}(1)}$, where $f$ is a computable function. Algorithms with this running-time are called
194 *fixed-parameter algorithms*. A less favorable outcome is an XP *algorithm*, which is an algorithm
195 running in time $\mathcal{O}(n^{f(k)})$; problems admitting such algorithms belong to the class XP.

196 Showing that a problem is W[1]-hard rules out the existence of a fixed-parameter algorithm under
197 the well-established assumption that $W[1] \neq \mathsf{FPT}$. This is usually done via a *parameterized*
198 *reduction* [8, 12] to some known W[1]-hard problem. A parameterized reduction from a parameterized
199 problem $\mathcal{P}$ to a parameterized problem $\mathcal{Q}$ is a function:

200 • which maps Yes-instances to Yes-instances and No-instances to No-instances,
201 • which can be computed in time $f(k) \cdot n^{\mathcal{O}(1)}$, where $f$ is a computable function, and
202 • where the parameter of the output instance can be upper-bounded by some function of the
203 parameter of the input instance.

**Treewidth.** A *nice tree-decomposition* $\mathcal{T}$ of a graph $G = (V, E)$ is a pair $(T, \chi)$, where $T$ is a tree
205 (whose vertices we call *nodes*) rooted at a node $r$ and $\chi$ is a function that assigns each node $t$ a set
206 $\chi(t) \subseteq V$ such that the following holds:

207 • For every $uv \in E$ there is a node $t$ such that $u, v \in \chi(t)$.
208 • For every vertex $v \in V$, the set of nodes $t$ satisfying $v \in \chi(t)$ forms a subtree of $T$.
209 • $|\chi(\ell)| = 1$ for every leaf $\ell$ of $T$ and $|\chi(r)| = 0$.
210 • There are only three kinds of non-leaf nodes in $T$:
211     – **Introduce node:** a node $t$ with exactly one child $t'$ such that $\chi(t) = \chi(t') \cup \{v\}$ for
212        some vertex $v \notin \chi(t')$.
213     – **Forget node:** a node $t$ with exactly one child $t'$ such that $\chi(t) = \chi(t') \setminus \{v\}$ for some
214        vertex $v \in \chi(t')$.
215     – **Join node:** a node $t$ with two children $t_1, t_2$ such that $\chi(t) = \chi(t_1) = \chi(t_2)$.

216 The *width* of a nice tree-decomposition $(T, \chi)$ is the size of a largest set $\chi(t)$ minus 1, and the
217 *treewidth* of the graph $G$, denoted $\mathrm{tw}(G)$, is the minimum width of a nice tree-decomposition
218 of $G$. Fixed-parameter algorithms are known for computing a nice tree-decomposition of optimal
219 width [4, 27]. For $t \in V(T)$ we denote by $T_t$ the subtree of $T$ rooted at $t$.

**Graph Parameters Based on Edge Cuts.** Traditionally, the bulk of graph-theoretic research on
221 structural parameters has focused on parameters that guarantee the existence of small vertex separators
222 in the graph; these are inherently tied to the theory of *graph minors* [40, 39] and the vertex deletion
223 distance. This approach gives rise not only to the classical notion of treewidth, but also to its
224 well-known restrictions and refinements such as *pathwidth* [40], *treedepth* [34] and the *vertex cover*
225 *number* [15, 28]. The vertex cover number is the most restrictive parameter in this hierarchy.

226 However, there are numerous problems of interest that remain intractable even when parameterized
227 by the vertex cover number. A recent approach developed for attacking such problems has been to
228 consider parameters that guarantee the existence of small edge cuts in the graph; these are typically
229 based on the edge deletion distance or, more broadly, tied to the theory of *graph immersions* [48, 32].
230 The parameter of choice for the latter is *treecut width* (tcw) [48, 32, 17, 18], a counterpart to
231 treewidth which has been successfully used to tackle some problems that remained intractable when

232 parameterized by the vertex cover number [20]. For the purposes of this manuscript, it will be useful
233 to note that graphs containing a vertex cover $X$ such that every vertex outside of $X$ has degree at
234 most 2 have treecut width at most $|X|$ [20, Section 3].

235 On the other hand, the by far most prominent parameter based on edge deletion distance is the
236 *feedback edge number* of a connected graph $G = (V, E)$, which is the minimum caardinality of a
237 set $F \subseteq E$ of edges (called the *feedback edge set*) such that $G - F$ is acyclic. The feedback edge
238 number can be computed in quadratic time and has primarily been used to obtain fixed-parameter
239 algorithms and polynomial kernels for problems where other parameterizations failed [20, 3, 2, 47].

240 Up to now, these were the only two edge-cut based graph parameters that have been considered in
241 the broader context of algorithm design. This situation could be seen as rather unstisfactory in view
242 of the large gap between the complexity of the richer class of graphs of bounded treecut width, and
243 the significantly simpler class of graphs of bounded feedback edge number—for instance, the latter
244 class is not even closed under disjoint union. Here, we propose a new parameter that lies "between"
245 the feedback edge number and treecut width, and which can be seen as a localized relaxation of the
246 feedback edge number: instead of measuring the total size of the feedback edge set, it only measures
247 how many feedback edges can "locally interfere with" any particular part of the graph.

248 Formally, for a connected graph $G = (V, E)$ and a spanning tree $T$ of $G$, let the *local feedback edge*
249 *set* at $v \in V$ be

$$E_{\mathrm{loc}}^T(v) = \{uw \in E \setminus E(T) \mid \text{ the unique path between } u \text{ and } w \text{ in } T \text{ contains } v\}.$$

250 The *local feedback edge number of* $(G, T)$ (denoted $\mathrm{lfen}(G, T)$) is then equal to $\max_{v \in V} |E_{\mathrm{loc}}^T(v)|$,
251 and the *local feedback edge number of* $G$ is simply the smallest local feedback edge number among
252 all possible spanning trees of $G$, i.e., $\mathrm{lfen}(G) = \min_{T \text{ is a spanning tree of } G} \mathrm{lfen}(G, T)$.

253 It is not difficult to show that the local feedback edge number is "sandwiched" between the feedback
254 edge number and treecut width. We also show that computing it is FPT.

255 **Proposition 1.** *For every graph $G$, $\mathrm{tcw}(G) \leq \mathrm{lfen}(G) + 1$ and $\mathrm{lfen}(G) \leq \mathrm{fen}(G)$.*

256 *Proof.* Let us begin with the second inequality. Consider an arbitrary spanning tree $T$ of $G$. Then for
257 every $v \in V(G)$, $E_{\mathrm{loc}}^T(v)$ is a subset of a feedback edge set corresponding to the spanning tree $T$, so
258 $|E_{\mathrm{loc}}^T(v)| \leq \mathrm{fen}(G)$ and the claim follows.

259 To establish the first inequality, we will use the notation and definition of treecut width from previous
260 work [18, Subsection 2.4]. Let $T$ be the spanning tree of $G$ with $\mathrm{lfen}(G, T) = \mathrm{lfen}(G)$. We construct
261 a treecut decomposition $(T, \mathcal{X})$ where each bag contains precisely one vertex, notably by setting
262 $X_t = \{t\}$ for each $t \in V(T)$. Fix any node $t$ in $T$ other than root, let $u$ be the parent of $t$ in $T$. All
263 the edges in $G \setminus ut$ with one endpoint in the rooted subtree $T_t$ and another outside of $T_t$ belong to
264 $E_{loc}^T(t)$, so $\mathrm{adh}_T(t) = |\mathrm{cut}(t)| \leq |E_{loc}^T(t)| \leq \mathrm{lfen}(G)$.

266 Let $H_t$ be the torso of $(T, \mathcal{X})$ in $t$, then $V(H_t) = \{t, z_1 ... z_l\}$ where $z_i$ correspond to connected
267 components of $T \setminus t$, $i \in [l]$. In $\tilde{H}(t)$, only $z_i$ with degree at least 3 are preserved. But all such $z_i$ are
268 the endpoints of at least 2 edges in $|E_{loc}^T(t)|$, so $\mathrm{tor}(t) = |V(\tilde{H}_t)| \leq 1 + |E_{loc}^T(t)| \leq 1 + \mathrm{lfen}(G)$.
269 Thus $\mathrm{tcw}(G) \leq \mathrm{lfen}(G) + 1$. $\qquad\square$

270 **Theorem 2.** *The problem of determining whether $\mathrm{lfen}(G) \leq k$ for an input graph $G$ parameterized*
271 *by an integer $k$ is fixed-parameter tractable. Moreover, if the answer is positive, we may also output*
272 *a spanning tree $T$ such that $\mathrm{lfen}(G, T) \leq k$ as a witness.*

273 *Proof.* Observe that since $\mathrm{tcw}(G) \leq \mathrm{lfen}(G) + 1$ by Proposition 1 and $\mathrm{tw}(G) \leq 2\,\mathrm{tcw}(G)^2 +$
274 $3\,\mathrm{tcw}(G)$ [17], we immediately see that no graph of treewidth greater than $k' = 2k^2 + 5k + 3$ can
275 have a local feedback edge set of at most $k$. Hence, let us begin by checking that $\mathrm{tw}(G) \leq k'$ using
276 the classical fixed-parameter algorithm for computing treewidth [4]; if not, we can safely reject the
277 instance.

278 Next, we use the fact that $\mathrm{tw}(G) \leq k'$ to invoke Courcelle's Theorem [6, 12], which provides a
279 fixed-parameter algorithm for model-checking any *Monadic Second-Order Logic* formula on $G$ when
280 parameterized by the size of the formula and the treewidth of $G$. We refer interested readers to the
281 appropriate books [7, 12] for a definition of Monadic Second Order Logic; intuitively, the logic

allows one to make statements about graphs using variables for vertices and edges as well as their sets, standard logical connectives, set inclusions, and atoms that check whether an edge is incident to a vertex. If the formula contains a free set variable $X$ and admits a model on $G$, Courcelle's Theorem allows us to also output an interpretation of $X$ on $G$ that satisfies the formula.

The formula $\phi$ we will use to check whether $\mathrm{lfen}(G) \leq k$ will be constructed as follows. $\phi$ contains a single free edge set variable $X$ (which will correspond to the sought-after feedback edge set). $\phi$ then consists of a conjunction of two parts, where the first part simply ensures that $X$ is a minimal feedback edge set using a well-known folklore construction [31, 1]; this also ensures that $G - X$ is a spanning tree. In the second part, $\phi$ quantifies over all vertices in $G$, and for each such vertex $v$ it says there exist edges $e_1, \ldots, e_k$ in $X$ such that for every edge $ab \in X$ distinct from all of $e_1, \ldots, e_k$, there exists a path $P$ between $a$ and $b$ in $G - X$ which is disjoint from $v$. (Note that since the path $P$ is unique in $G - X$, one could also quantify $P$ universally and achieve the same result.)

It is easy to verify that $\phi(X)$ is satisfied in $G$ if and only if $\mathrm{lfen}(G, G - X) \leq k$, and so the proof follows. Finally, we remark that—as with every algorithmic result arising from Courcelle's Theorem—one could also use the formula as a template to build an explicit dynamic programming algorithm that proceeds along a tree-decomposition of $G$. $\qquad\square$

# 3 Solving BNSL$^{\neq 0}$ with Parameters Based on Edge Cuts.

In this section we provide tractability and lower-bound results for BNSL$^{\neq 0}$ from the viewpoint of superstructure parameters based on edge cuts. Together with the previous lower bound that rules out fixed-parameter algorithms based on all vertex-separator parameters [36, Theorem 3], the results presented here provide a comprehensive picture of the complexity of BNSL$^{\neq 0}$ with respect to superstructure parameterizations.

## 3.1 Using the Feedback Edge Number for BNSL$^{\neq 0}$

We say that two instances $\mathcal{I}, \mathcal{I}'$ of BNSL are *equivalent* if (1) they are either both Yes-instances or both No-instances, and furthermore (2) a solution to one instance can be transformed into a solution to the other instance in polynomial time. Our aim here is to prove the following theorem:

**Theorem 3.** *There is an algorithm which takes as input an instance $\mathcal{I}$ of BNSL$^{\neq 0}$ whose superstructure has* fen *$k$, runs in time $\mathcal{O}(|\mathcal{I}|^2)$, and outputs an equivalent instance $\mathcal{I}' = (V', \mathcal{F}', \ell')$ of BNSL$^{\neq 0}$ such that $|V'| \leq 16k$.*

In parameterized complexity theory, such data reduction algorithms with performance guarantees are called *kernelization algorithms* [12, 8]. These may be applied as a polynomial-time preprocessing step before, e.g., more computationally expensive methods are used. The fixed-parameter tractability of BNSL$^{\neq 0}$ when parameterized by the fen of the superstructure follows as an immediate corollary of Theorem 3 (one may solve $\mathcal{I}$ by, e.g., exhaustively looping over all possible DAGs on $V'$ via a brute-force procedure). We also note that even though the number of variables of the output instance is polynomial in the parameter $k$, the instance $\mathcal{I}'$ need not have size polynomial in $k$.

We begin our path towards a proof of Theorem 3 by computing a feedback edge set $E_F$ of $G$ of size $k$ in time $\mathcal{O}(|\mathcal{I}|^2)$ by, e.g., Prim's algorithm. Let $T$ be the spanning tree of $G$, $E_F = E(G) \setminus E(T)$. The algorithm will proceed by the recursive application of certain reduction rules, which are polynomial-time operations that alter ("simplify") the input instance in a certain way. A reduction rule is *safe* if it outputs an instance which is equivalent to the input instance. We start by describing a rule that will be used to prune $T$ until all leaves are incident to at least one edge in $E_F$.

**Reduction Rule 1.** *Let $v \in V$ be a vertex and let $Q$ be the set of neighbors of $v$ with degree 1 in $G$. We construct a new instance $\mathcal{I}' = (V', \mathcal{F}', \ell)$ by setting:* **1.** *$V' := V \setminus Q$;* **2.** *$\Gamma_{f'}(v) := \{\emptyset\} \cup \{ (P \setminus Q) \mid P \in \Gamma_f(v) \}$;* **3.** *for all $w \in V' \setminus \{v\}$, $f'_w = f_w$;* **4.** *for every $P' \in \Gamma_{f'}(v)$:*

$$f'_v(P') := \max_{P: P \setminus Q = P'} \Big( f_v(P) + \sum_{v_{\mathrm{in}} \in P \cap Q} f_{v_{\mathrm{in}}}(\emptyset) + \sum_{v_{\mathrm{out}} \in Q \setminus P} \max(f_{v_{\mathrm{out}}}(\emptyset), f_{v_{\mathrm{out}}}(v)) \Big).$$

**Lemma 4.** *Reduction Rule 1 is safe.*

*Proof.* For the forward direction, assume that $\mathcal{I}'$ admits a solution $D'$, and let $\lambda$ be the score $D'$ achieves on $v$. By the construction of $\mathcal{I}'$, there must be a parent set $Z \in \Gamma_f(v)$ such that $Z \cap V' = P_{D'}(v)$ (i.e., $Z$ agrees with $v$'s parents in $D'$) and $\lambda$ is the sum of the following scores: (1) $f_v(Z)$, (2) the maximum achievable score for each vertex in $Q \setminus Z$, and (3) the score of $\{\emptyset\}$ for each vertex in $Z \cap Q$. Let $D$ be obtained from $D'$ by adding the following arcs: $zv$ for each $z \in Z$, and $vq$ for each $q \in Q \setminus Z$ such that $q$ achieves its maximum score with $v$ as its parent. By construction, $\lambda = \sum_{w' \in \{v\} \cup Q} f_w(P_D(w))$. Since the scores of $D$ and $D'$ coincide on all vertices outside of $\{v\} \cup Q$ and $D$, we conclude that $\texttt{score}(D) = \texttt{score}(D')$, and hence $\mathcal{I}$ is a **Yes**-instance.

For the converse direction, assume that $\mathcal{I}$ admits a solution $D$. Let $D' = D - Q$. By the construction of $f'_v$, it follows that $f'_v(P_{D'}(v))$ is greater or equal to the score $D$ achieves on $\{v\} \cup Q$. Thus, $D'$ is a solution to $\mathcal{I}'$, and we conclude that Reduction Rule 1 is safe. $\qquad\square$

Observe that the superstructure graph $G'$ obtained after applying one step of Reduction Rule 1 is simply $G - Q$; after its exhaustive application we obtain an instance $\mathcal{I}$ such that all the leaves of the tree $T$ are endpoints of $E_F$. Our next step is to get rid of long paths in $G$ whose internal vertices have degree 2. We note that this step is more complicated than in typical kernelization results using feedback edge set as the parameter, since a directed path $Q$ in $G$ can serve multiple "roles" in a hypothetical solution $D$ and our reduction gadget needs to account for all of these. Intuitively, $Q$ may or may not appear as a directed path in $D$ (which impacts what other arcs can be used in $D$ due to acyclicity), and in addition the total score achieved by $D$ on the internal vertices of $Q$ needs to be preserved while taking into account whether the endpoints of $Q$ have a neighbor in the path or not. Because of this (and unlike in many other kernelization results of this kind [20, 46, 18]), we will not be replacing $Q$ merely by a shorter path, but by a more involved gadget.

**Reduction Rule 2.** *Let $a, b_1, \ldots, b_m, c$ be a path in $G$ such that for each $i \in [m]$, $b_i$ has degree precisely 2. For each $B \subseteq \{a, c\}$, let $\ell_{\max}(B)$ be the maximum sum of scores that can be achieved by $b_1, \ldots, b_m$ under the condition that $b_1$ (and analogously $b_m$) takes $a$ ($c$) into its parent set if and only if $a \in B$ ($c \in B$). In other words, $\ell_{\max}(B) = \max_{D_B} \sum_{b_i \mid i \in [m]} f_{b_i}(P_{D_B}(b_i))$ where $D_B$ is a DAG on $\{b_1, \ldots, b_m\} \cup B$ such that $B$ does not contain any vertices of out-degree 0 in $D_B$. Moreover, let $\ell_{\mathrm{noPath}}(a)$ (and analogously $\ell_{\mathrm{noPath}}(c)$) be the maximum score that can be achieved on the vertices $b_1, \ldots, b_m$ by a DAG on $a, b_1, \ldots, b_m, c$ with the following properties: $a$ ($c$) has out-degree 1, $c$ ($a$) has out-degree 0, and there is no directed path from $a$ to $b_m$ (from $c$ to $b_1$).*

*We construct a new instance $\mathcal{I}' = (V', \mathcal{F}', \ell)$ as follows:*

- $V' := V \cup \{b\} \setminus \{b_2 \ldots b_{m-1}\}$;

- $\Gamma_{f'}(b) = \{B \cup \{b_1, b_m\} \mid B \subseteq \{a, c\}\}$ *with scores* $f'_b(B \cup \{b_1, b_m\}) := \ell_{\max}(B)$;

- *The scores for $a$ and $c$ are obtained from $\mathcal{F}$ by simply adding $b$ to any parent set containing either $b_1$ or $b_m$; formally:*

  - $\Gamma_{f'}(a)$ *is a union of* $\{P \in \Gamma_f(a) \mid b_1 \notin P\}$, *where* $f'_a(P) := f_a(P)$ *and* $\{P \cup \{b\} \mid b_1 \in P, P \in \Gamma_f(a)\}$, *where* $f'_a(P \cup \{b\}) := f_a(P)$;
  - $\Gamma_{f'}(c)$ *is a union of* $\{P \in \Gamma_f(c) \mid b_m \notin P\}$, *where* $f'_c(P) := f_c(P)$, *and* $\{P \cup \{b\} \mid b_m \in P, P \in \Gamma_f(c)\}$, *where* $f'_c(P \cup \{b\}) := f_c(P)$.

- $\Gamma_{f'}(b_1)$ *contains only* $\{a, b, b_m\}$ *with score* $\ell_{\mathrm{noPath}}(a)$;

- $\Gamma_{f'}(b_m)$ *contains only* $\{c, b, b_1\}$ *with score* $\ell_{\mathrm{noPath}}(c)$;

- *for all* $w \in V' \setminus \{a, b_1, b, b_m, c\}$, $f'_w = f_w$.

An Illustration of Reduction Rule 2 is provided in Figure 2. The rule can be applied in linear time, since the 6 values of $\ell_{noPath}$ and $\ell_{max}$ can be computed in linear time by a simple dynamic programming subroutine that proceeds along the path $a, b_1, \ldots, b_m, c$ (alternatively, one may instead invoke the fact that paths have treewidth 1 [36]).

**Lemma 5.** *Reduction Rule 2 is safe.*

*Proof.* Note that the superstructure graph of reduced instance is obtained from $G_{\mathcal{I}}$ by contracting $b_2 \ldots b_{m-1}$, adding $b$ and connecting it by edges to $a, c, b_1, b_m$. We will show that a score of at least $\ell$

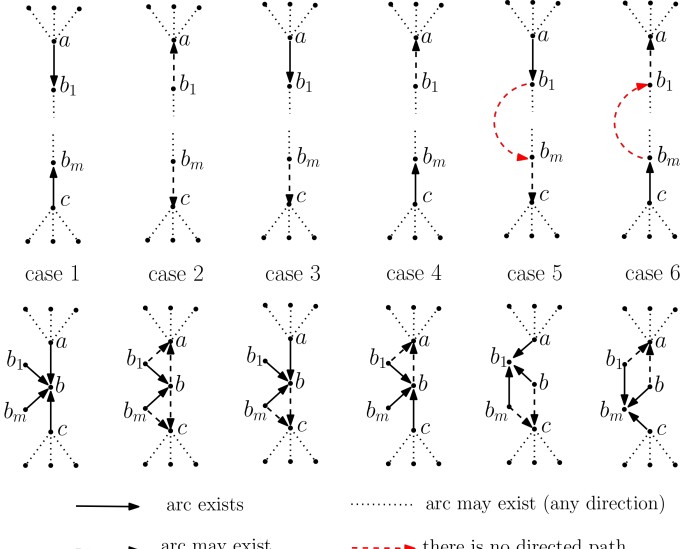

Figure 2:
Top: The six possible scenarios that give rise to the values of $\ell_{max}$ (Cases 1-4) and $\ell_{noPath}$ (Cases 5-6).
Bottom: The corresponding arcs in the gadget after the application of Reduction Rule 2.

→ arc exists ⋯⋯⋯ arc may exist (any direction)

⤏ arc may exist ⇢ there is no directed path

can be achieved in the original instance $\mathcal{I}$ if and only if a score of at least $\ell$ can be achieved in the reduced instance $\mathcal{I}'$.

Assume that $D$ is a DAG that achieves a score of $\ell$ in $\mathcal{I}$. We will construct a DAG $D'$, called the *reduct* of $D$, with $f'(D') \geq \ell$. To this end, we first modify $D$ by removing the vertices $b_2...b_{m-1}$ and adding $b$ (let us denote the DAG obtained at this point $D^*$). Further modifications of $D^*$ depend only on $D[a, b_1...b_m, c]$, and we distinguish the 6 cases listed below (see also Figure 2):

- case 1: $D$ contains both arcs $ab_1$ and $cb_m$. We add to $D^*$ arcs from $a, c, b_1, b_m$ to $b$, denote resulting graph by $D'$. As $D'$ is obtained from DAG by making $b$ a sink, it is a DAG as well. Parent set of $b$ in $D'$ is $\{a, c, b_1, b_m\}$, so its score is $\ell_{max}(a, c) \geq \sum_{i=1}^{m} f_{b_i}(P_D(b_i))$, which means that it achieves the highest scores all of $b_i$'s can achieve in $D$. The remaining vertices in $V(D') \setminus \{b_1, b_m, b\}$ have the same scores as in $D$, so $f'(D') \geq f(D) = \ell$.

- case 2: $D$ contains none of the arcs $ab_1$ and $cb_m$. To keep the scores of $a$ and $c$ the same as in $D$, we add to $D^*$ the arc $ba$ iff $D$ contains $b_1a$, add arc $bc$ iff $D$ contains $b_mc$. Furthermore, we add arcs $b_1b$ and $b_mb$ and denote resulting graph $D'$. As $D'$ is obtained from $D$ by making $b$ a source and then adding sources $b_1$ and $b_m$, it is a DAG as well. The parent set of $b$ in $D'$ is $\{b_1, b_m\}$, so its score is $\ell_{max}(\emptyset) \geq \sum_{i=1}^{m} f_{b_i}(P_D(b_i))$. Rest of vertices in $V(D') \setminus \{b_1, b_m, b\}$ have the same scores as in $D$, so $f'(D') \geq f(D) = \ell$.

- case 3: $D$ doesn't contain arc $ab_1$, but contains $cb_m$ and all the arcs $b_{i+1}b_i$, $i \in [m-1]$. We add to $D^*$ arcs $cb$, $b_1b$ and $b_mb$. We also add $ba$ iff $D$ contains $b_1a$, to preserve the score of $a$. Denote resulting graph by $D'$. $D'$ can be considered as $D$ where long directed path $c \to b_m \to ... \to b_1$ was replaced by $c \to b$ and then sources $b_1$ and $b_m$ were added, so it is a DAG. Arguments for scores are similar to cases 1 and 2.

- case 4: $D$ doesn't contain arc $cb_m$, but contains $ab_1$ and all the arcs $b_ib_{i+1}$, $i \in [m-1]$. This case is symmetric to case 3.

- case 5: $D$ contains the arc $ab_1$ but does not contain the arc $cb_m$ and at least one of the arcs $b_ib_{i+1}$, $i \in [m-1]$ is also missing (i.e., there is no directed path from $a$ to $b_m$). We add to $D'$ arcs $bb_1$ and $b_mb_1$. If $b_mc \in A(D)$, add also $bc$. Denote the resulting graph $D'$. As $D'$ is obtained from $D^*$ by making $b_1$ a sink and $b$ a source, it is a DAG. $b_1$ has parent set $\{a, b, b_m\}$ in $D'$, so its score is $\ell_{noPath}(a) \geq \sum_{i=1}^{m} f_{b_i}(P_D(b_i))$. Rest of vertices in $V(D') \setminus \{b_1, b_m, b\}$ have the same scores as in $D$, so $f'(D') \geq f(D) = \ell$.

- case 6: $D$ contains the arc $cb_m$ but does not contain the arc $ab_1$ and at least one of the arcs $b_{i+1}b_i$, $i \in [m-1]$ is also missing. This case is symmetric to case 5.

The considered cases exhaustively partition all possible configurations of $D[a, b_1...b_m, c]$, so we always can construct $D'$ with a score at least $\ell$. For the converse direction, note that the DAGs constructed in cases 1-6 cover all optimal configurations on $\{a, b_1, b, b_m, c\}$: if there is a DAG $D''$ in $\mathcal{I}'$ with a score of $\ell'$, we can always reverse the construction to obtain a DAG $D'$ with score at least $\ell'$ such that $D'[a, b_1, b, b_m, c]$ has one of the forms depicted at the bottom line of the figure. The claim for the converse direction follows from the fact that every such $D'$ is a reduct of some DAG $D$ of the original instance with the same score. $\square$

We are now ready to prove the desired result.

*Proof of Theorem 3.* We begin by exhaustively applying Reduction Rule 1 on an instance whose superstructure graph has a feedback edge set of size $k$, which results in an instance with the same feedback edge set but whose spanning tree $T$ has at most $2k$ leaves. It follows that there are at most $2k$ vertices with a degree greater than 2 in $T$.

Let us now "mark" all the vertices that either are endpoints of the edges in $E_F$ or have a degree greater then 2 in $T$; the total number of marked vertices is upper-bounded by $4k$. We now proceed to the exhaustive application of Reduction Rule 2, which will only be triggered for sufficiently long paths in $T$ that connect two marked vertices but contain no marked vertices on its internal vertices; there are at most $4k$ such paths due to the tree structure of $T$. Reduction Rule 2 will replace each such path with a set of 3 vertices, and therefore after its exhaustive application we obtain an equivalent instance with at most $4k + 4k \cdot 3 = 16k$ vertices, as desired. Correctness follows from the safeness of Reduction Rules 1, 2, and the runtime bound follows by observing that the total number of applications of each rule as well as the runtime of each rule are upper-bounded by a linear function of the input size. $\square$

## 3.2 Fixed-Parameter Tractability of BNSL$^{\neq 0}$ using the Local Feedback Edge Number

Our aim here will be to lift the fixed-parameter tractability of BNSL$^{\neq 0}$ established by Theorem 3 by relaxing the parameterization to lfen. In particular, we will prove:

**Theorem 6.** BNSL$^{\neq 0}$ *is fixed-parameter tractable when parameterized by the local feedback edge number of the superstructure.*

Since fen is a more restrictive parameter than lfen, this results in a strictly larger class of instances being identified as tractable. However, the means we will use to establish Theorem 6 will be fundamentally different: we will not use a polynomial-time data reduction algorithm as the one provided in Theorem 3, but instead apply a dynamic programming approach. Since the kernels constructed by Theorem 3 contain only polynomially-many variables w.r.t. fen, that result is incomparable to Theorem 6.

In fact one can use standard techniques to prove that, under well-established complexity assumptions, a data reduction result such as the one provided in Theorem 3 *cannot* exist for lfen. The intuitive reason for this is that lfen is a "local" parameter that does not increase by, e.g., performing a disjoint union of two distinct instances (the same property is shared by many other well-known parameters such as treewidth, pathwidth, treedepth, clique-width, and treecut width). We provide a formal proof of this claim at the end of Subsection 3.3.

As our first step towards proving Theorem 6, we provide general conditions for when the union of two DAGs is a DAG as well. Let $D = (V, A)$ be a directed graph and $V' \subseteq V$. Denote by $\mathtt{Con}(V', D)$ the binary relation on $V' \times V'$ which specifies whether vertices from $V'$ are connected by a path in $D$: $\mathtt{Con}(V', D) = \{(v_1, v_2) \subseteq V' \times V' | \exists \text{ directed path from } v_1 \text{ to } v_2 \text{ in } D\}$. Similarly to arcs, we will use $v_1 v_2 \in$ as shorthand for $(v_1, v_2)$; we will also use $\mathtt{trcl}$ to denote the transitive closure.

**Lemma 7.** *Let $D_1$, $D_2$ be directed graphs with common vertices $V_{\text{com}} = V(D_1) \cap V(D_2)$, $V_{\text{com}} \subseteq V_1 \subseteq V(D_1)$, $V_{\text{com}} \subseteq V_2 \subseteq V(D_2)$. Then:*

- *(i) $\mathtt{Con}(V_1 \cup V_2, D_1 \cup D_2) = \mathtt{trcl}(\mathtt{Con}(V_1, D_1) \cup \mathtt{Con}(V_2, D_2))$;*

- *(ii) If $D_1$, $D_2$ are DAGs and $\mathtt{Con}(V_1 \cup V_2, D_1 \cup D_2)$ is irreflexive, then $D_1 \cup D_2$ is a DAG.*

*Proof.* (i) Denote $R_i := \mathtt{Con}(V_i, D_i)$, $i = 1, 2$. Obviously $\mathtt{trcl}(R_1 \cup R_2)$ is a subset of $\mathtt{Con}(V_1 \cup V_2, D_1 \cup D_2)$. Assume that for some $x, y \in V_1 \cup V_2$ there exists a directed path $P$ from $x$ to $y$ in $D_1 \cup D_2$. We will show (by induction on the length $l$ of shortest $P$) that $xy \in \mathtt{trcl}(R_1 \cup R_2)$.

- $l = 1$: in this case there is an arc $xy$ in some $D_i$, so $xy \in R_i \subseteq \mathtt{trcl}(R_1 \cup R_2)$

- $l \to l+1$. If P is completely contained in some $D_i$, then $xy \in R_i \subseteq \mathtt{trcl}(R_1 \cup R_2)$. Otherwise $P$ must contain arcs $e \notin A(D_1)$, $f \notin A(D_2)$. Then there is $w \in V_{com} \subseteq V_1 \cup V_2$ between them. By the induction hypothesis $xw \in \mathtt{trcl}(R_1 \cup R_2)$ and $wy \in \mathtt{trcl}(R_1 \cup R_2)$, so $xy \in \mathtt{trcl}(R_1 \cup R_2)$

(ii) The precondition implies that the digraph $D_1 \cup D_2$ induced on $V_1 \cup V_2$ is a DAG. Assume that $D_1 \cup D_2$ is not a DAG and let $C$ be a shortest directed cycle in $D_1 \cup D_2$. As $D_1$ and $D_2$ are DAGs, $C$ must contain arcs $e \notin A(D_1)$, $f \notin A(D_2)$. So there are least 2 different vertices $x, y$ from $V_{com}$ in $C$. By (i) we have that $xy \in \mathtt{trcl}(R_1 \cup R_2)$ and $yx \in \mathtt{trcl}(R_1 \cup R_2)$, then also $xx \in \mathtt{trcl}(R_1 \cup R_2)$, which contradicts irreflexivity. $\qquad \square$

Towards proving Theorem 6, assume that we are given an instance $\mathcal{I} = (V, \mathcal{F}, \ell)$ of BNSL$^{\neq 0}$ with connected superstructure graph $G = (V, E)$. Let $T$ be a fixed rooted spanning tree of $G$ such that $\mathrm{lfen}(G, T) = \mathrm{lfen}(G) = k$, denote the root by $r$. For $v \in V(T)$, let $T_v$ be the subtree of $T$ rooted at $v$, let $V_v = V(T_v)$, and let $\bar{V}_v = N_G(V_v) \cup V_v$. We define the *boundary* $\delta(v)$ of $v$ to be the set of endpoints of all edges in $G$ with precisely one endpoint in $V_v$ (observe that the boundary can never have a size of 1). $v$ is called *closed* if $|\delta(v)| \leq 2$ and *open* otherwise. We begin by establishing some basic properties of the local feedback edge set.

**Observation 8.** *Let $v$ be a vertex of $T$. Then:*

1. *For every closed child $w$ of $v$ in $T$, it holds that $\delta(w) = \{v, w\}$ and $vw$ is the only edge between $V_w$ and $V \setminus V_w$ in $G$.*

2. $|\delta(v)| \leq 2k + 2$.

3. *Let $\{v_i | i \in [t]\}$ be the set of all open children of $v$ in $T$. Then $t \leq 2k$ and*
$$\delta(v) \subseteq \cup_{i=1}^{t} \delta(v_i) \cup \{v\} \cup N_G(v)$$

*Proof.* The first claim follows by the connectivity assumption on $G$ and the definition of boundary.

For the second claim, clearly $\delta(r) = \emptyset$. Let $v \neq r$ have the parent $u$, and consider an arbitrary $w \in \delta(v) \setminus \{u, v\}$. Then there is an edge $ww' \in E(G)$ with precisely one endpoint in $V_v$ and $ww' \neq uv$. Hence $ww' \notin E(T)$ and the path between $w$ and $w'$ in T contains $v$, and this implies $ww' \in E_{loc}^T(v)$ by definition. Consequently, $w \in V_{loc}^T(v)$. For the claimed bound we note that $|V_{loc}^T(v)| \leq 2|E_{loc}^T(v)| \leq 2k$.

For the third claim, let $w = v_i$ for some $i \in [t]$. As $w$ is open, there exists an edge $e \neq vw$ between $V_w$ and $V \setminus V_w$ in $G$. By definition of local feedback edge set, $e \in E_{loc}^T(v)$. Let $x_w$ be the endpoint of $e$ that belongs to $V_w$, then $x_w \in V_{loc}^T(v)$ and $x_w \notin V_{w'}$ for any open child $w' \neq w$ of $v$. But $|V_{loc}^T(v)| \leq 2k$, which yields the bound on number $t$ of open children.
For the boundary inclusion, consider any edge $c$ in $G$ with precisely one endpoint $x_v$ in $V_v$. Note that $x_v$ can not belong to $V_w$ for any closed child $w$ of $v$. If $x_v \in V_{v_i}$ for some $i \in [t]$, then endpoints of $c$ belong to $\delta(v_i)$. Otherwise $x_v = v$ and therefore the second endpoint of $c$ is in $N_G(v)$. $\qquad \square$

With Observation 8 in hand, we can proceed to a definition of the records used in our dynamic program. Intuitively, these records will be computed in a leaf-to-root fashion and will store at each vertex $v$ information about the best score that can be achieved by a partial solution that intersects the subtree rooted at $v$.

Let $R$ be a binary relation on $\delta(v)$ and $s$ an integer. For $s \in \mathbb{Z}$, we say that $(R : s)$ is a *record* for a vertex $v$ if and only if there exists a DAG $D$ on $\bar{V}_v$ such that (1) $w \in V_v$ for each arc $uw \in A(D)$, (2) $R = \mathtt{Con}(\delta(v), D)$ and (3) $\sum_{u \in V_v} f_u(P_D(u)) = s$. The records $(R, s)$ where $s$ is maximal for fixed $R$ are called *valid*. Denote the set of all valid records for $v$ by $\mathcal{R}(v)$, and note that $|\mathcal{R}(v)| \leq 2^{\mathcal{O}(k^2)}$.

Observe that if $v_i$ is a closed child of $v$, then by Observation 8.1 $\mathcal{R}(v_i)$ consists of precisely two valid records: one for $R = \emptyset$ and one for $R = \{vv_i\}$. Moreover, the root $r$ of $T$ has only a single valid record $(\emptyset : s_{\mathcal{I}})$, where $s_{\mathcal{I}}$ is the maximum score that can be achieved by a solution in $\mathcal{I}$. The following lemma lies at the heart of our result and shows how we can compute our records in a leaf-to-root fashion along $T$.

**Lemma 9.** *Let $v \in V(G)$ have $m$ children in $T$ where $m > 0$, and assume we have computed $\mathcal{R}(v_i)$ for each child $v_i$ of $v$. Then $\mathcal{R}(v)$ can be computed in time at most $m \cdot |\Gamma_f(v)| \cdot 2^{\mathcal{O}(k^3)}$.*

*Proof.* Without loss of generality, let the open children of $v \in V(G)$ be $v_1, \ldots, v_t$ and let the remaining (i.e., closed) children of $v$ be $v_{t+1}, \ldots, v_m$; recall that by Point 3. of Observation 8, $t \leq 2k$. For each closed child $v_j$, $j \in [m] \setminus [t]$, let $s_j^\emptyset$ be the second component of the valid record for $\emptyset \in \mathcal{R}(v_j)$, and let $s_j^\times$ be the second component of the valid record for the single non-empty relation in $\mathcal{R}(v_j)$. Consider the following procedure $\mathbb{A}$.

First, $\mathbb{A}$ branches over all choices of $P \in \Gamma_f(v)$ and all choices of $(R_i, s_i) \in \mathcal{R}(v_i)$ for each individual open child $v_i$ of $v$. Let $R_0 = \{ pv \mid p \in P \}$ and let $R' = \bigcup_{j \in [t]_0} R_j$. If $\texttt{trcl}(R')$ is not irreflexive, we discard this branch; otherwise, we proceed as follows. Let $R_{new}$ be the subset of $R'$ containing all arcs $uw$ such that $w \in V_v$. Moreover, let $s_{new} = f_v(P) + (\sum_{i \in [t]} s_i) + (\sum_{i \in [m] \setminus [t] \mid v_i \in P} s_i^\emptyset) + (\sum_{i \in [m] \setminus [t] \mid v_i \notin P} (\max(s_i^\emptyset, s_i^\times)))$.

The algorithm $\mathbb{A}$ gradually constructs a set $\mathcal{R}^*(v)$ as follows. At the beginning, $\mathcal{R}^*(v) = \emptyset$. For each newly obtained tuple $(R_{new}, s_{new})$, $\mathbb{A}$ checks whether $\mathcal{R}^*(v)$ already contains a tuple with $R_{new}$ as its first element; if not, we add the new tuple to $\mathcal{R}^*(v)$. If there already exists such a tuple $(R_{new}, s_{old}) \in \mathcal{R}^*(v)$, we replace it with $(R_{new}, \max(s_{old}, s_{new}))$.

For the running time, recall that in order to construct $\mathcal{R}^*(v)$ the algorithm branched over $|\Gamma_f(v)|$-many possible parent sets of $v$ and over the choice of at most $2k$-many binary relations $R_i$ on the boundaries of open children. According to Observation 8.2, there are at most $3^{(2k+2)^2}$ options for every such relation, so we have at most $\mathcal{O}((3^{(2k+2)^2})^{2k} \cdot |\Gamma_f(v)|) \leq 2^{\mathcal{O}(k^3)} \cdot |\Gamma_f(v)|$ branches. In every branch we compute $\texttt{trcl}(R')$ in time $k^{\mathcal{O}(1)}$ and then compute the value of $s_{new}$ using the equation provided above before updating $\mathcal{R}^*(v)$, which takes time at most $\mathcal{O}(m)$.

Finally, to establish correctness it suffices to prove following claim:

**Claim 1.** *$(R : s)$ is a record for $v$ if and only if there exist $P \in \Gamma_f(v)$ and records $(R_i : s_i)$ for $v_i$, $i \in [m]$, such that:*

- *$\texttt{trcl}(\cup_{i=0}^t R_i)$ is irreflexive;*

- *$R_i = \emptyset$ for any closed child $v_i \in P$;*

- *$\sum_{i=1}^m s_i + f_v(P) = s$;*

- *$R = (\texttt{trcl}(\cup_{i=0}^t R_i))|_{\delta(v) \times \delta(v)}$.*

*Moreover, if $(R : s) \in \mathcal{R}(v)$ then in addition:*

- *$(R_i : s_i) \in \mathcal{R}(v_i)$, $i \in [t]$;*

- *for every closed child $v_i \notin P$, $s_i = \max(s_i^\emptyset, s_i^\times)$.*

*Proof of the Claim.* (a) ($\Leftarrow$) Denote $V_i = V_{v_i}$ and $\bar{V}_i = \bar{V}_{v_i}$, $i \in [m]$. For every $i \in [m]$ there exists DAG $D_i$ on $\bar{V}_i$ such that all its arcs finish in $V_i$, $R_i = \texttt{Con}(\delta(v_i), D_i)$ and $\sum_{u \in V_i} f_u(P_{D_i}(u)) = s_i$. Denote by $D_0$ DAG on $V_0 = v \cup N_G(v)$ with arc set $R_0$. We will construct the witness $D$ of $(R, s)$ by gluing together all $D_i$, $i \in [m]_0$.

We start from $D_0$ and DAGs of open children. Note that $\texttt{Con}(V_0, D_0) = R_0$ and $\texttt{Con}(\delta(v_i), D_i) = R_i$ for $i \in [t]$. Inductive application of Lemma 7 to DAGs $D_i$, $i \in [t]$, yields $\texttt{Con}(\cup_{i=1}^t \delta(v_i) \cup V_0, D^*) = \texttt{trcl}(\cup_{i=0}^t R_i)$. In particular, as $\delta(v) \subseteq \cup_{i=1}^t \delta(v_i) \cup V_0$ by Observation 8.3, we have that $\texttt{Con}(\delta(v), D^*) = (\texttt{trcl}(\cup_{i=0}^t R_i))|_{\delta(v) \times \delta(v)} = R$. As $\texttt{trcl}(\cup_{i=0}^t R_i)$ is irreflexive, $D^* = \cup_{i=0}^t D_i$ is DAG by Lemma 7.

Now we add to $D^*$ DAGs for closed children and finally obtain $D = \cup_{i=t+1}^m D_i \cup D^*$. For every closed child $v_i$, $D_i$ is by Observation 8.1 the union of $v$ and $D_i \setminus v$, plus at most one of arcs $vv_i$, $v_iv$ between them (recall $R_i = \emptyset$ for any closed child $v_i \in P$). Note that $D_i \setminus v$ can share only

$v_i$ with $D_0$ and doesn't have common vertices with any other $D_j$. Therefore any directed path in $D$ starting and finishing outside outside of $V_i$, $i > t$, doesn't intersect $V_i$. In particular, acyclicity of $D^*$ and $D_i$, $i \in [m] \setminus [t]$, implies acyclicity of $D$; $\mathtt{Con}(\delta(v), D) = \mathtt{Con}(\delta(v), D^*) = R$.

All the arcs in $D_i$ finish in $V_i$, so parent set for every $x_i \in D_i$ in $D$ is the same as in $D_i$, $i \in [m]$. Also parent set of $v$ in $D$ is the same as in $D_0$. So

$$\sum_{u \in V_v} f_u(P_D(u)) = \sum_{i=1}^{m} \sum_{u \in V_i} f_u(P_{D_i}(u)) + f_v(P_{D_0}(v)) = \sum_{i=1}^{m} s_i + f_v(P) = s$$

($\Rightarrow$) Let $D$ be a witness for $(R : s)$, i.e. $D$ is DAG on $\bar{V}_v$ with all arcs finishing in $V_v$ such that $\sum_{u \in V_v} f_u(P_D(u)) = s$ and $\mathtt{Con}(\delta(v), D) = R$. For $i = 1 \in [m]$ define $D_i' = D[\bar{V}_i]$ and let $D_i$ be obtained from $D_i'$ by deleting arcs that finish outside $V_i$. Note that $\cup_{i=1}^{m} D_i = D$. Let $R_i = \mathtt{Con}(\delta(v_i), D_i)$, as in ($\Leftarrow$) we have that $R = \mathtt{Con}(\delta(v), D) = \mathtt{trcl}(\cup_{i=0}^{t} R_i))|_{\delta(v) \times \delta(v)}$. As $D$ is DAG, $\mathtt{trcl}(\cup_{i=0}^{t} R_i)$ is irreflexive and $R_i = \emptyset$ for any closed child $v_i \in P$. Local score for $D_i$ is

$$s_i = \sum_{u \in V_i} f_u(P_{D_i}(u)) = \sum_{u \in V_i} f_u(P_{D_i'}(u)) = \sum_{u \in V_i} f_u(P_D(u))$$

So $v_i$ has record $(R_i : s_i)$. Denote $P = P_D(v)$. Then:

$$s = \sum_{u \in V_v} f_u(P_D(u)) = \sum_{i=1}^{m} \sum_{u \in V_i} f_u(P_D(u)) + f_v(P_D(v)) = \sum_{i=1}^{m} s_i + f_v(P)$$

(b) Let $(R : s) \in \mathcal{R}(v)$ and all $D, P, D_i, R_i, s_i$ are as in $(a)(\Rightarrow)$. Assume that for some $i$ $(R_i, s_i)$ is not valid record of $v_i$. In this case $v_i$ must have a record $(R_i : s_i + \Delta)$ with $\Delta > 0$. But then $(a)(\Leftarrow)$ implies that $v$ has record $(R : s + \Delta)$, which contradicts to validity of $(R : s)$

Assume that some closed $v_i \notin P$ has valid record $(R_i', s_i + \Delta)$ with $\Delta > 0$. $R'$ and $R$ differ only by arc $vv_i$, so addition or deletion of the arc to $D$ would increase the total score by $\Delta > 0$ without creating cycles. This would result in record $(R : s + \Delta)$ and yield a contradiction with validity of $(R : s)$. ■ □

We are now ready to prove the main result of this subsection.

*Proof of Theorem 6.* We provide an algorithm that solves $\mathrm{BNSL}^{\neq 0}$ in time $2^{O(k^3)} \cdot n^3$, where $n = |\mathcal{I}|$, assuming that a spanning tree $T$ of $G$ such that $\mathrm{lfen}(G, T) = k$ is provided as part of the input. Once that is done, the theorem will follow from Theorem 2.

The algorithm computes $\mathcal{R}(v)$ for every node $v$ in $T$, moving from leaves to the root:

- For a leaf $v$, compute $\mathcal{R}^*(v) := \{(R_P : f_v(P)) | P \in \Gamma_f(v), R_P = \{uv | u \in P\}\}$. This can be done by simply looping over $\Gamma_f(v)$ in time $\mathcal{O}(n)$. Note that $\mathcal{R}^*(v)$ is the set of all records of $v$, so we can correctly set $\mathcal{R}(v) := \{(R : s) \in \mathcal{R}^*(v) |$ there is no $(R : s') \in \mathcal{R}^*(v)$ with $s' > s\}$.

- Let $v \in V(G)$ have at least one child in $T$, and assume we have computed $\mathcal{R}(v_i)$ for each child $v_i$ of $v$. Then we invoke Lemma 9 to compute $\mathcal{R}(v)$ in time at most $m \cdot |\Gamma_f(v)| \cdot 2^{\mathcal{O}(k^2)} \leq 2^{\mathcal{O}(k^2)} \cdot n^2$. □

## 3.3 Lower Bounds for BNSL$^{\neq 0}$

Since $\mathrm{lfen}$ lies between $\mathrm{fen}$ and treecut width in the parameter hierarchy (see Proposition 1) and $\mathrm{BNSL}^{\neq 0}$ is FPT when parameterized by $\mathrm{lfen}$, the next step would be to ask whether this tractability result can be lifted to treecut width. Below, we answer this question negatively.

**Theorem 10.** BNSL$^{\neq 0}$ *is* W[1]-*hard when parameterized by the treecut width of the superstructure graph.*

563  In fact, we show an even stronger result: $\text{BNSL}^{\neq 0}$ is $\mathsf{W}[1]$-hard when parameterized by the vertex
564  cover number of the superstructure even when all vertices outside of the vertex cover are required to
565  have degree at most 2. We remark that while $\text{BNSL}^{\neq 0}$ was already shown to be $\mathsf{W}[1]$-hard when
566  parameterized by the vertex cover number [36], in that reduction the degree of the vertices outside of
567  the vertex cover is not bounded by a constant and, in particular, the graphs obtained in that reduction
568  have unbounded treecut width.

569  *Proof of Theorem 10.* We reduce from the following well-known $\mathsf{W}[1]$-hard problem [12, 8]:

---

**REGULAR MULTICOLORED CLIQUE (RMC)**

570

Input:      A $k$-partite graph $G = (V_1 \cup ... \cup V_k, E)$ such that $|N_G(v)| = m$ for every $v \in V$

Parameter:   The integer $k$

Question:    Are there nodes $v^i$ that form a $k$-colored clique in $G$, i.e. $v^i \in V_i$ and $v^i v^j \in E$ for all $i, j \in [k], i \neq j$?

---

571  We say that vertices in $V_i$ have color $i$. Let $G = (V_1 \cup ... \cup V_k, E)$ be an instance of RMC. We
572  will construct an instance $(V, \mathcal{F}, \ell)$ of $\text{BNSL}^{\neq 0}$ such that $\mathcal{I}$ is a **Yes**-instance if and only if $G$ is a
573  **Yes**-instance of RMC. $V$ consists of one vertex $v_i$ for each color $i \in [k]$ and one vertex $v_e$ for every
574  edge $e \in E$. For each edge $e \in E$ that connects a vertex of color $i$ with a vertex of color $j$, the
575  constructed vertex $v_e$ will have precisely one element in its score function that achieves a non-zero
576  score, in particular: $f_{v_e}(\{v_i, v_j\}) = 1$.

577  Next, for each $i \in [k]$, we define the scores for $v_i$ as follows. For every $v \in V_i$, let $E_v$ be the set of all
578  edges incident to $v$ in $G$, and let $P_i^v = \{v_e : e \in E_v\}$. We now set $f_{v_i}(P_i^v) = m + 1$ for each such
579  $v$; all other parent sets will receive a score of 0. Note that $\{ v_i \mid i \in [k] \}$ forms a vertex cover of the
580  superstructure graph and that all vertices outside of this vertex cover have degree at most 2, as desired.
581  We will show that $G$ has a $k$-colored clique if and only if there is a Bayesian network $D$ with score at
582  least $\ell = |E| + k + \binom{k}{2}$. (In fact, it will later become apparent that the score can never exceed $\ell$.)

583  Assume first that $G$ has a $k$-colored clique on $v^i, i \in [k]$, consisting of a set $E_X$ of $\binom{k}{2}$ edges.
584  Consider the digraph $D$ on $V$ obtained as follows. For each vertex $v_i$, $i \in [k]$, and each vertex
585  $v_e$ where $e \in E$, $D$ contains the arc $v_e v_i$ if $v_e$ is incident to $v^i$ and otherwise $D$ contains the arc
586  $v_i v_e$. This completes the construction of $D$. Now notice that the construction guarantees that each
587  $v_i$ receives the parent set $P_i^{v^i}$ and hence contributes a score of $m + 1$. Moreover, for every edge $e$
588  not incident to a vertex in the clique, the vertex $v_e$ contributes a score of 1; note that the number
589  of such edges is $|E| - km + \binom{k}{2}$; indeed, every $v_i$ is incident to $m$ edges but since $v^i, i \in [k]$,
590  was a clique we are guaranteed to double-count precisely $\binom{k}{2}$ many edges. Hence the total score is
591  $k(m + 1) + |E| - km + \binom{k}{2} = |E| + k + \binom{k}{2}$, as desired.

592  Assume that $\mathcal{I} = (V, \mathcal{F}, \ell)$ is a **Yes**-instance and let $s_{\texttt{opt}} \geq \ell = |E| + k + \binom{k}{2}$ be the maximum score
593  that can be achieved by a solution to $\mathcal{I}$; let $D$ be a dag witnessing such a score. Then all $v_i$, $i \in [k]$,
594  must receive a score of $m + 1$ in $D$. Indeed, assume that some $v_i$ receives a score of 0 and let $P_v$ be
595  any parent set of $v_i$ with a score $m + 1$. Modify $D$ by orienting edges $v_i v_e$ for every $v_e \in P_v$ inside
596  $v_i$. Now local score of $v_i$ is $m + 1$, total score of the rest of vertices decreased by at most $m$ (maximal
597  number of $v_e$ that had local score 1 in $D$ and lost it after the modification). So the modified DAG has
598  a score of at least $s_{\texttt{opt}} + 1$, which contradicts the optimality of $s_{\texttt{opt}}$. Therefore all $v_i$, $i \in [k]$, get
599  score $m + 1$ in $D$.

600  Let $P_i$ be parent set of $v_i$ in $D$, then $|P_i| = m$, $P_i = P_i^{v^i}$ for some $v^i \in V_i$. For every $v_e \in P_i$,
601  the local score of $v_e$ in $D$ is 0. Denote by $E_{unsat}$ the set of all $v_e$ that have a score of 0 in $D$. Every
602  $v_e$ belongs to at most 2 different $P_i$ and $P_i \cap P_j \leq 1$ for every $i \neq j$, so $|E_{unsat}| \geq km - \binom{k}{2}$. If
603  $|E_{unsat}| > km - \binom{k}{2}$, sum of local scores of $e_v$ in $D$ would be smaller then $|E| - km + \binom{k}{2}$, which
604  results in $s_{\texttt{opt}} < |E| + k + \binom{k}{2}$. Therefore $|E_{unsat}| = km - \binom{k}{2}$. But this means that $P_i \cap P_j \neq \emptyset$ for
605  any $i \neq j$, i.e. $v^i, i \in [k]$ form a $k$-colored clique in $G$. In particular $s_{\texttt{opt}} = \ell$.      $\square$

606  For our second result, we note that the construction in the proof of Theorem 10 immediately implies
607  that $\text{BNSL}^{\neq 0}$ is $\mathsf{NP}$-hard even under the following two conditions: (1) $\ell + \sum_{v \in V} |\Gamma_f(v)| \in \mathcal{O}(|V|^2)$

608 (i.e., the size of the parent set encoding is quadratic in the number of vertices), and (2) the instances
609 are constructed in a way which makes it impossible to achieve a score higher than $\ell$. Using this,
610 as a fairly standard application of *AND-cross-compositions* [8] we can exclude the existence of an
611 efficient data reduction algorithm for $\mathrm{BNSL}^{\neq 0}$ parameterized by lfen:

**Theorem 11.** *Unless* $\mathsf{NP} \subseteq \mathsf{co\text{-}NP/poly}$*, there is no polynomial-time algorithm which takes as*
613 *input an instance $\mathcal{I}$ of* $\mathrm{BNSL}^{\neq 0}$ *whose superstructure has* lfen $k$ *and outputs an equivalent instance*
614 $\mathcal{I}' = (V', \mathcal{F}', \ell')$ *of* $\mathrm{BNSL}^{\neq 0}$ *such that* $|V'| \in k^{\mathcal{O}(1)}$*. In particular,* $\mathrm{BNSL}^{\neq 0}$ *does not admit a*
615 *polynomial kernel when parameterized by* lfen*.*

*Proof Sketch.* We describe an AND-cross-composition for the problem while closely following the
617 terminology and intuition introduced in Section 15 in the book [8]. Let the input consist of instances
618 $\mathcal{I}_1, \ldots, \mathcal{I}_t$ of (unparameterized) instances of $\mathrm{BNSL}^{\neq 0}$ which satisfy conditions (1) and (2) mentioned
619 above, and furthermore all have the same size and same target value of $\ell_1$ (which is ensured through
620 the use of the polynomial equivalence relation $\mathcal{R}$ [8, Definition 15.7]). The instance $\mathcal{I}$ produced
621 on the output is merely the disjoint union of instances $\mathcal{I}_1, \ldots, \mathcal{I}_t$ where we set $\ell := t \cdot \ell_1$, and we
622 parameterize $\mathcal{I}$ by lfen.

Observe now that condition (a) in Definition 15.7 [8] is satisfied by the fact that the local feedback
624 edge number of $\mathcal{I}$ is upper-bounded by the number of edges in a connected component of $\mathcal{I}$. Moreover,
625 the AND- variant of condition (b) in that same definition (see Subsection 15.1.3 [8]) is satisfied as
626 well: since none of the original instances can have a score greater than $\ell_1$, $\mathcal{I}$ achieves a score of $\ell_1 \cdot t$
627 if and only if each of the original instances was a Yes-instance.

This completes the construction of an AND-cross-composition for $\mathrm{BNSL}^{\neq 0}$ parameterized by lfen,
629 and the claim follows by Theorem 15.12 [8]. □

# 4 Additive Scores and Treewidth

While the previous section focused on the complexity of BNSL when the non-zero representation
632 was used (i.e., $\mathrm{BNSL}^{\neq 0}$), here we turn our attention to the complexity of the problem with respect to
633 the additive representation. Recall from Subsection 2 that there are two variants of interest for this
634 representation: $\mathrm{BNSL}^+$ and $\mathrm{BNSL}^+_{\leq}$. We begin by showing that, unsurprisingly, both of these are
635 NP-hard.

**Theorem 12.** $\mathrm{BNSL}^+$ *is* NP-*hard. Moreover,* $\mathrm{BNSL}^+_{\leq}$ *is* NP-*hard for every* $q \geq 3$.

*Proof.* We provide a direct reduction from the following NP-hard problem [23, 10]:

---

MINIMUM FEEDBACK ARC SET ON BOUNDED-DEGREE DIGRAPHS (MFAS)

Input:      Digraph $D = (V, A)$ whose skeleton has degree at most 3, integer $m \leq |A|$.
Question:   Is there a subset $A' \subseteq A$ where $|A'| \leq m$ such that $D - A'$ is a DAG?

---

Let $(D, m)$ be an instance of MFAS. We construct an instance $\mathcal{I}$ of $\mathrm{BNSL}^+_{\leq}$ as follows:

640 • $V = V(D)$,

641 • $f_y(x) = 1$ for every $xy \in A(D)$,

642 • $f_y(x) = 0$ for every $xy \in A_V \setminus A(D)$,

643 • $\ell = |A| - m$, and

644 • $q = 3$.

Assume that $(D, m)$ is a Yes-instance and $A'$ is any feedback arc set of size $m$. Let $D'$ be the DAG
646 obtained from $D$ after deleting arcs in $A'$. Then $\mathrm{score}(D')$ is equal to the number of arcs in $D'$,
647 which is $|A| - m$, so $\mathcal{I}$ is a Yes-instance. On the other hand, if $\mathcal{I}$ is a Yes-instance of $\mathrm{BNSL}^+$,
648 pick any DAG $D'$ with $\mathrm{score}(D') \geq \ell = |A| - m$. Without loss of generality we may assume that
649 $A(D') \subseteq A$, as the remaining arcs have a score of zero and may hence be removed. All the arcs in $A$
650 have a score 1 and hence the DAG $D'$ contains at least $|A| - m$ arcs, i.e., it can be obtained from $D$

651 by deleting at most $m$ arcs. Hence $(D, m)$ is also a Yes-instance. To establish the NP-hardness of
652 BNSL$^+$, simply disregard the bound $q$ on the input. $\qquad\square$

653 While the use of the additive representation did not affect the classical complexity of BNSL, it makes
654 a significant difference in terms of parameterized complexity. Indeed, in contrast to BNSL$^{\neq 0}$:

655 **Theorem 13.** BNSL$^+$ *is* FPT *when parameterized by the treewidth of the superstructure. Moreover,*
656 BNSL$^+_{\leq}$ *is* FPT *when parameterized by q plus the treewidth of the superstructure.*

657 *Proof.* We begin by proving the latter statement, and will then explain how that result can be
658 straightforwardly adapted to obtain the former. As our initial step, we apply Bodlaender's algorithm [4,
659 27] to compute a nice tree-decomposition $(\mathcal{T}, \chi)$ of $G_{\mathcal{I}}$ of width $k = \mathrm{tw}(G_{\mathcal{I}})$. In this proof we use
660 $T$ to denote the set of nodes of $\mathcal{T}$ and $r \in T$ be the root of $\mathcal{T}$. Given a node $t \in T$, let $\chi_t^{\downarrow}$ be the set of
661 all vertices occurring in bags of the rooted subtree $T_t$, i.e., $\chi_t^{\downarrow} = \{u \mid \exists t' \in T_t \text{ such that } u \in \chi(t')\}$.
662 Let $G_t^{\downarrow}$ be the subgraph of $G_{\mathcal{I}}$ induced on $\chi_t^{\downarrow}$.

663 To prove the theorem, we will design a leaf-to-root dynamic programming algorithm which will
664 compute and store a set of records at each node of $T$, whereas once we ascertain the records for $r$
665 we will have the information required to output a correct answer. Intuitively, the records will store
666 all information about each possible set of arcs between vertices in each bag, along with relevant
667 connectivity information provided by arcs between vertices in $\chi_t^{\downarrow}$ and information about the partial
668 score. They will also keep track of parent set sizes in each bag.

669 Formally, the records will have the following structure. For a node $t$, let $S(t) =$
670 $\{(\mathrm{loc}, \mathrm{con}, \mathrm{inn}) \mid \mathrm{loc}, \mathrm{con} \subseteq A_{\chi(t)}, \mathrm{inn} : \chi(t) \to [q]_0\}$ be the set of *snapshots* of $t$. The record $\mathcal{R}_t$
671 of $t$ is then a mapping from $S(t)$ to $\mathbb{N}_0 \cup \{\bot\}$. Observe that $|S(t)| \leq 4^{k^2}(q+1)^k$. To introduce the
672 semantics of our records, let $\Upsilon_t$ be the set of all directed acyclic graphs over the vertex set $\chi_t^{\downarrow}$ with
673 maximal in-degree at most $q$, and let $D_t = (\chi_t^{\downarrow}, A)$ be a directed acyclic graph in $\Upsilon_t$. We say that the
674 *snapshot of $D_t$ in $t$* is the tuple $(\alpha, \beta, p)$ where $\alpha = A \cap A_{\chi(t)}$, $\beta = \mathrm{Con}(\chi(t), D_t)$ and $p$ specifies
675 numbers of parents of vertices from $\chi(t)$ in $D$, i.e. $p(v) = |\{w \in \chi_t^{\downarrow} | wv \in A\}|$, $v \in \chi(t)$. We are
676 now ready to define the record $\mathcal{R}_t$. For each snapshot $(\mathrm{loc}, \mathrm{con}, \mathrm{inn}) \in S(t)$:

677 - $\mathcal{R}_t(\mathrm{loc}, \mathrm{con}, \mathrm{inn}) = \bot$ if and only if there exists no directed acyclic graph in $\Upsilon_t$ whose
678 snapshot is $(\mathrm{loc}, \mathrm{con}, \mathrm{inn})$, and

679 - $\mathcal{R}_t(\mathrm{loc}, \mathrm{con}, \mathrm{inn}) = \tau$ if $\exists D_t \in \Upsilon_t$ such that

680     – the snapshot of $D_t$ is $(\mathrm{loc}, \mathrm{con}, \mathrm{inn})$,
681     – score$(D_t) = \tau$, and
682     – $\forall D_t' \in \Upsilon_t$ such that the snapshot of $D_t'$ is $(\mathrm{loc}, \mathrm{con}, \mathrm{inn})$: score$(D_t) \geq$ score$(D_t')$.

683 Recall that for the root $r \in T$, we assume $\chi(r) = \emptyset$. Hence $\mathcal{R}_r$ is a mapping from the one-element
684 set $\{(\emptyset, \emptyset, \emptyset)\}$ to an integer $\tau$ such that $\tau$ is the maximum score that can be achieved by any DAG
685 $D = (V, A)$ with all in-degrees of vertices upper bounded by $q$. In other words, $\mathcal{I}$ is a YES-instance
686 if and only if $\mathcal{R}_r(\emptyset, \emptyset, \emptyset) \geq \ell$. To prove the theorem, it now suffices to show that the records can be
687 computed in a leaf-to-root fashion by proceeding along the nodes of $T$. We distinguish four cases:

688 $t$ **is a leaf node.** Let $\chi(t) = \{v\}$. By definition, $S(t) = \{(\emptyset, \emptyset, \emptyset)\}$ and $\mathcal{R}_t(\emptyset, \emptyset, \emptyset) = f_v(\emptyset)$.

689 $t$ **is a forget node.** Let $t'$ be the child of $t$ in $\mathcal{T}$ and let $\chi(t) = \chi(t') \setminus \{v\}$. We initiate by setting
690 $\mathcal{R}_t^0(\mathrm{loc}, \mathrm{con}, \mathrm{inn}) = \bot$ for each $(\mathrm{loc}, \mathrm{con}, \mathrm{inn}) \in S(t)$.

691 For each $(\mathrm{loc}', \mathrm{con}', \mathrm{inn}') \in S(t')$, let $\mathrm{loc}_v, \mathrm{con}_v$ be the restrictions of $\mathrm{loc}', \mathrm{con}'$ to tu-
692 ples containing $v$. We now define $\mathrm{loc} = \mathrm{loc}' \setminus \mathrm{loc}_v$, $\mathrm{con} = \mathrm{con}' \setminus \mathrm{con}_v$, $\mathrm{inn} =$
693 $\mathrm{inn}'|_{\chi(t)}$ and say that $(\mathrm{loc}, \mathrm{con}, \mathrm{inn})$ is *induced* by $(\mathrm{loc}', \mathrm{con}', \mathrm{inn}')$. Set $\mathcal{R}_t^0(\mathrm{loc}, \mathrm{con}, \mathrm{inn}) :=$
694 $\max(\mathcal{R}_t^0(\mathrm{loc}, \mathrm{con}, \mathrm{inn}), \mathcal{R}_{t'}(\mathrm{loc}', \mathrm{con}', \mathrm{inn}'))$, where $\bot$ is assumed to be a minimal element.

695 For correctness, it will be useful to observe that $\Upsilon_t = \Upsilon_{t'}$. Consider our final computed value of
696 $\mathcal{R}_t^0(\mathrm{loc}, \mathrm{con}, \mathrm{inn})$ for some $(\mathrm{loc}, \mathrm{con}, \mathrm{inn}) \in S(t)$.

697 If $\mathcal{R}_t(\mathrm{loc}, \mathrm{con}, \mathrm{inn}) = \tau$ for some $\tau \neq \bot$, then there exists a DAG $D$ which wit-
698 nesses this. But then $D$ also admits a snapshot $(\mathrm{loc}', \mathrm{con}', \mathrm{inn}')$ at $t'$ and witnesses

$\mathcal{R}_{t'}(\text{loc}', \text{con}', \text{inn}') \geq \tau$. Note that $(\text{loc}, \text{con}, \text{inn})$ is induced by $(\text{loc}', \text{con}', \text{inn}')$. So in our algorithm $\mathcal{R}_t^0(\text{loc}, \text{con}, \text{inn}) \geq \mathcal{R}_{t'}(\text{loc}', \text{con}', \text{inn}') \geq \tau$.

If on the other hand $\mathcal{R}_t^0(\text{loc}, \text{con}, \text{inn}) = \tau$ for some $\tau \neq \perp$, then there exists a snapshot $(\text{loc}', \text{con}', \text{inn}')$ such that $(\text{loc}, \text{con}, \text{inn})$ is induced by $(\text{loc}', \text{con}', \text{inn}')$ and $\mathcal{R}_{t'}(\text{loc}', \text{con}', \text{inn}') = \tau$. $\mathcal{R}_t(\text{loc}, \text{con}, \text{inn}) \geq \tau$ now follows from the existence of a DAG witnessing the value of $\mathcal{R}_{t'}(\text{loc}', \text{con}', \text{inn}')$.

Hence, we can correctly set $\mathcal{R}_t = \mathcal{R}_t^0$.

$t$ **is an introduce node.** Let $t'$ be the child of $t$ in $\mathcal{T}$ and let $\chi(t) = \chi(t') \cup \{v\}$. We initiate by setting $\mathcal{R}_t^0(\text{loc}, \text{con}, \text{inn}) = \perp$ for each $(\text{loc}, \text{con}, \text{inn}) \in S(t)$.

For each $(\text{loc}', \text{con}', \text{inn}') \in S(t')$ and each $Q \subseteq \{ab \in A_{\chi(t)} \mid \{a, b\} \cap \{v\} \neq \emptyset\}$, we define:

- $\text{loc} := \text{loc}' \cup Q$

- $\text{con} := \texttt{trcl}(con' \cup Q)$

- $\text{inn}(x) := \text{inn}'(x) + |\{y \in \chi(t) | yx \in Q\}|$ for every $x \in \chi(t) \setminus \{v\}$
  $\text{inn}(v) := |\{y \in \chi(t) | yv \in Q\}|$

If con is not irreflexive or $\text{inn}(x) > q$ for some $x \in \chi(t)$, discard this branch. Otherwise, let $\mathcal{R}_t^0(\text{loc}, \text{con}, \text{inn}) := \max(\mathcal{R}_t^0(\text{loc}, \text{con}, \text{inn}), \texttt{new})$ where $\texttt{new} = \mathcal{R}_{t'}(\text{loc}', \text{con}', \text{inn}') + \sum_{ab \in Q} f_b(a)$. As before, $\perp$ is assumed to be a minimal element here.

Consider our final computed value of $\mathcal{R}_t^0(\text{loc}, \text{con}, \text{inn})$ for some $(\text{loc}, \text{con}, \text{inn}) \in S(t)$.

For correctness, assume that $\mathcal{R}_t^0(\text{loc}, \text{con}, \text{inn}) = \tau$ for some $\tau \neq \perp$ and is obtained from $(\text{loc}', \text{con}', \text{inn}'), Q$ defined as above. Then $\mathcal{R}_{t'}(\text{loc}', \text{con}', \text{inn}') = \tau - \sum_{ab \in Q} f_b(a)$. Construct a directed graph $D$ from the witness $D'$ of $\mathcal{R}_{t'}(\text{loc}', \text{con}', \text{inn}')$ by adding the arcs specified in $Q$. As con $= \texttt{trcl}(con' \cup Q)$ is irreflexive and $D'$ is a DAG, $D$ is a DAG as well by 7. Moreover, $\text{inn}(x) \leq q$ for every $x \in \chi(t)$ and the rest of vertices have in $D$ the same parents as in $D'$, so $D \in \Upsilon_t$. In particular, $(\text{loc}, \text{con}, \text{inn})$ is a snapshot of $D$ in $t$ and $D$ witnesses $\mathcal{R}_t(\text{loc}, \text{con}, \text{inn}) \geq \mathcal{R}_{t'}(\text{loc}', \text{con}', \text{inn}') + \sum_{ab \in Q} f_b(a) = \tau$.

On the other hand, if $\mathcal{R}_t(\text{loc}, \text{con}, \text{inn}) = \tau$ for some $\tau \neq \perp$, then there must exist a directed acyclic graph $D = (\chi_t^{\downarrow}, A)$ in $\Upsilon_t$ that achieves a score of $\tau$. Let $Q$ be the restriction of $A$ to arcs containing $v$, and let $D' = (\chi_t^{\downarrow} \setminus v, A \setminus Q)$, clearly $D' \in \Upsilon_{t'}$. Let $(\text{loc}', \text{con}', \text{inn}')$ be the snapshot of $D'$ at $t'$. Observe that loc $= \text{loc}' \cup Q$, con $= \texttt{trcl}(con' \cup Q)$, inn differs from $\text{inn}'$ by the numbers of incoming arcs in $Q$ and the score of $D'$ is precisely equal to the score $\tau$ of $D$ minus $\sum_{(a,b) \in Q} f_b(a)$. Therefore $\mathcal{R}_{t'}(\text{loc}', \text{con}', \text{inn}') \geq \tau - \sum_{(a,b) \in Q} f_b(a)$ and in the algorithm $\mathcal{R}_t^0(\text{loc}, \text{con}, \text{inn}) \geq \mathcal{R}_{t'}(\text{loc}', \text{con}', \text{inn}') + \sum_{(a,b) \in Q} f_b(a) \geq \tau$. Equality then follows from the previous direction of the correctness argument.

Hence, at the end of our procedure we can correctly set $\mathcal{R}_t = \mathcal{R}_t^0$.

$t$ **is a join node.** Let $t_1, t_2$ be the two children of $t$ in $\mathcal{T}$, recall that $\chi(t_1) = \chi(t_2) = \chi(t)$. By the well-known separation property of tree-decompositions, $\chi_{t_1}^{\downarrow} \cap \chi_{t_2}^{\downarrow} = \chi(t)$ [12, 8]. We initiate by setting $\mathcal{R}_t^0(\text{loc}, \text{con}, \text{inn}) := \perp$ for each $(\text{loc}, \text{con}, \text{inn}) \in S(t)$.

Let us branch over each $\text{loc}, \text{con}_1, \text{con}_2 \subseteq A_{\chi(t)}$ and $\text{inn}_1, \text{inn}_2 : \chi(t) \to [q]_0$. For every $b \in \chi(t)$ set $\text{inn}(b) = \text{inn}_1(b) + \text{inn}_2(b) - |\{a | ab \in \text{loc}\}|$. If:

- $\texttt{trcl}(\text{con}_1 \cup \text{con}_2)$ is not irreflexive and/or

- $\mathcal{R}_{t_1}(\text{loc}, \text{con}_1, \text{inn}_1) = \perp$, and/or

- $\mathcal{R}_{t_2}(\text{loc}, \text{con}_2, \text{inn}_2) = \perp$, and/or

- $\text{inn}(b) > q$ for some $b \in \chi(t)$

743 then discard this branch. Otherwise, set $\mathrm{con} = \mathtt{trcl}(\mathrm{con}_1 \cup \mathrm{con}_2)$, $\mathtt{doublecount} = \sum_{ab \in \mathrm{loc}} f_b(a)$

744 and $\mathtt{new} = \mathcal{R}_{t_1}(\mathrm{loc}, \mathrm{con}_1) + \mathcal{R}_{t_2}(\mathrm{loc}, \mathrm{con}_2) - \mathtt{doublecount}$. We then set $\mathcal{R}_t^0(\mathrm{loc}, \mathrm{con}, \mathrm{inn}) :=$

745 $\max(\mathcal{R}_t^0(\mathrm{loc}, \mathrm{con}, \mathrm{inn}), \mathtt{new})$ where $\bot$ is once again assumed to be a minimal element.

746 At the end of this procedure, we set $\mathcal{R}_t = \mathcal{R}_t^0$.

747 For correctness, assume that $\mathcal{R}_t^0(\mathrm{loc}, \mathrm{con}, \mathrm{inn}) = \tau \neq \bot$ is obtained from $\mathrm{loc}, \mathrm{con}_1, \mathrm{con}_2, \mathrm{inn}_1, \mathrm{inn}_2$

748 as above. Let $D_1 = (\chi_{t_1}^\downarrow, A_1)$ and $D_2 = (\chi_{t_2}^\downarrow, A_2)$ be DAGs witnessing $\mathcal{R}_{t_1}(\mathrm{loc}, \mathrm{con}_1, \mathrm{inn}_1)$

749 and $\mathcal{R}_{t_2}(\mathrm{loc}, \mathrm{con}_2, \mathrm{inn}_2)$ correspondingly. Note that common vertices of $D_1$ and $D_2$ are precisely

750 $\chi(t)$. In particular, if $D_1$ and $D_2$ share an arc $ab$, then $a, b \in \chi(t)$ and therefore $ab \in \mathrm{loc}$. On

751 the other hand, $\mathrm{loc} \subseteq A_1$, $\mathrm{loc} \subseteq A_2$, so $loc = A_1 \cap A_2$. Hence $\mathrm{inn}$ specifies the number of

752 parents of every $b \in \chi(T)$ in $D = D_1 \cup D_2$. Rest of vertices $v \in V(D) \setminus \chi(t)$ belong to

753 precisely one of $D_i$ and their parents in $D$ are the same as in this $D_i$. As $\mathtt{trcl}(\mathrm{con}_1 \cup \mathrm{con}_2)$ is

754 irreflexive, $D$ is a DAG by Lemma 7, so $D \in \Upsilon_t$. The snapshot of $D$ in $t$ is $(\mathrm{loc}, \mathrm{con}, \mathrm{inn})$ and

755 $\mathtt{score}(D) = \sum_{ab \in A(D)} f_b(a) = \sum_{ab \in A_1} f_b(a) + \sum_{ab \in A_2} f_b(a) - \sum_{ab \in \mathrm{loc}} f_b(a) = \mathtt{score}(D_1) +$

756 $\mathtt{score}(D_2) - \mathtt{doublecount} = \mathcal{R}_{t_1}(\mathrm{loc}, \mathrm{con}_1, \mathrm{inn}_1) + \mathcal{R}_{t_2}(\mathrm{loc}, \mathrm{con}_2, \mathrm{inn}_2) - \mathtt{doublecount} = \tau$.

757 So $D$ witnesses that $\mathcal{R}_t(\mathrm{loc}, \mathrm{con}, \mathrm{inn}) \geq \tau$.

758 For the converse, assume that $\mathcal{R}_t(\mathrm{loc}, \mathrm{con}, \mathrm{inn}) = \tau \neq \bot$ and $D$ is a DAG witnessing this. Let

759 $D_1$ and $D_2$ be restrictions of $D$ to $\chi_{t_1}^\downarrow$ and $\chi_{t_2}^\downarrow$ correspondingly, then by the same arguments as

760 above $A(D_1) \cap A(D_2) = \mathrm{loc}$, in particular $D = D_1 \cup D_2$. Let $(\mathrm{loc}, \mathrm{con}_i, \mathrm{inn}_i)$ be the snapshot

761 of $D_i$ in $t_i$, $i = 1, 2$, then $\mathcal{R}_{t_i}(\mathrm{loc}, \mathrm{con}_i, \mathrm{inn}_i) \geq \mathtt{score}(D_i)$. By the procedure of our algorithm,

762 $\mathcal{R}_t^0(\mathrm{loc}, \mathrm{con}, \mathrm{inn}) \geq \mathcal{R}_{t_1}(\mathrm{loc}, \mathrm{con}_1, \mathrm{inn}_1) + \mathcal{R}_{t_2}(\mathrm{loc}, \mathrm{con}_2, \mathrm{inn}_2) - \mathtt{doublecount} \geq \mathtt{score}(D_1) +$

763 $\mathtt{score}(D_2) - \sum_{ab \in \mathrm{loc}} f_b(a) = \mathtt{score}(D) = \tau$.

764 Hence the resulting record $\mathcal{R}_t$ is correct, which concludes the correctness proof of the algorithm.

765 Since the nice tree-decomposition $\mathcal{T}$ has $\mathcal{O}(n)$ nodes, the runtime of the algorithm is upper-bounded

766 by $\mathcal{O}(n)$ times the maximum time required to process each node. This is dominated by the time

767 required to process join nodes, for which there are at most $(2^{k^2})^3((q+1)^k)^2 = 8^{k^2} \cdot (q+1)^{2k}$ branches

768 corresponding to different choices of $\mathrm{loc}, \mathrm{con}_1, \mathrm{con}_2, \mathrm{inn}_1, \mathrm{inn}_2$. Constructing $\mathtt{trcl}(\mathrm{con}_1 \cup \mathrm{con}_2)$

769 and verifying that it is irreflexive can be done in time $\mathcal{O}(k^3)$. Computing $\mathtt{doublecount}$ and $\mathrm{inn}$

770 takes time at most $\mathcal{O}(k^2)$. So the record for a join node can be computed in time $2^{\mathcal{O}(k^2)} \cdot q^{\mathcal{O}(k)}$.

771 Hence, after we have computed a width-optimal tree-decomposition for instance by Bodlaender's

772 algorithm [4], the total runtime of the algorithm is upper-bounded by $2^{\mathcal{O}(k^2)} \cdot q^{\mathcal{O}(k)} \cdot n$.

773 Finally, to obtain the desired result for $\mathrm{BNSL}^+$, we can simply adapt the above algorithm by

774 disregarding the entry $\mathrm{inn}$ and disregard all explicit bounds on the in-degrees (e.g., in the definition

775 of $\Upsilon_t$). The runtime for this dynamic programming procedure is then $2^{\mathcal{O}(k^2)} \cdot n$. $\qquad \square$

776 This completely resolves the parameterized complexity of $\mathrm{BNSL}^+$ w.r.t. all parameters depicted

777 on Figure 1. However, the same is not true for $\mathrm{BNSL}_{\leq}^+$: while a careful analysis of the algorithm

778 provided in the proof of Theorem 13 reveals that $\mathrm{BNSL}_{\leq}^+$ is XP-tractable when parameterized by the

779 treewidth of the superstructure alone, it is not yet clear whether it is FPT—in other words, do we

780 need to parameterize by both $q$ and treewidth to achieve fixed-parameter tractability?

781 We conclude this section by answering this question affirmatively. To do so, we will aim to reduce

782 from the following problem, which can be seen as a dual to the W[1]-hard MULTIDIMENSIONAL

783 SUBSET SUM problem considered in recent works [21, 18].

784

UNIFORM DUAL MULTIDIMENSIONAL SUBSET SUM (UDMSS)

| Input: | An integer $k$, a set $S = \{s_1, \ldots, s_n\}$ of item-vectors with $s_i \in \mathbb{N}^k$ for every $i$ with $1 \leq i \leq n$, a uniform target vector $t = (r, \ldots, r) \in \mathbb{N}^k$, and an integer $d$. |
|---|---|
| Parameter: | $k$. |
| Question: | Is there a subset $S' \subseteq S$ with $|S'| \geq d$ such that $\sum_{s \in S'} s \leq t$? |

785 We first begin by showing that this variant of the problem is W[1]-hard by giving a fairly direct

786 reduction from the originally considered problem, and then show how it can be used to obtain the

787 desired lower-bound result.

**Lemma 14.** DMSS *is* W[1]*-hard.*

*Proof.* The W[1]-hard MULTIDIMENSIONAL SUBSET SUM problem is stated as follows:

| MULTIDIMENSIONAL SUBSET SUM (MSS) | |
|---|---|
| Input: | An integer $k$, a set $S = \{s_1, \ldots, s_n\}$ of item-vectors with $s_i \in \mathbb{N}^k$ for every $i$ with $1 \le i \le n$, a target vector $t = (t^1, \ldots, t^k) \in \mathbb{N}^k$, and an integer $d$. |
| Parameter: | $k$. |
| Question: | Is there a subset $S' \subseteq S$ with $|S'| \le d$ such that $\sum_{s \in S'} s \ge t$? |

Consider its dual version, obtained by reversing both inequalities:

| DUAL MULTIDIMENSIONAL SUBSET SUM (DMSS) | |
|---|---|
| Input: | An integer $k$, a set $S = \{s_1, \ldots, s_n\}$ of item-vectors with $s_i \in \mathbb{N}^k$ for every $i$ with $1 \le i \le n$, a target vector $t = (t^1, \ldots, t^k) \in \mathbb{N}^k$, and an integer $d$. |
| Parameter: | $k$. |
| Question: | Is there a subset $S' \subseteq S$ with $|S'| \ge d$ such that $\sum_{s \in S'} s \le t$? |

Given an instance $\mathcal{I} = (S, t, k, d)$ of MSS, we construct an instance $\mathcal{I}_d = (S, z - t, k, n - d)$ of DMSS, where $z = \sum_{s \in S} s$. Note that $S'$ is a witness of $\mathcal{I}$ if and only if $S \setminus S'$ is a witness of $\mathcal{I}_d$. The observation establishes W[1]-hardness of DMSS.

Now it remains to show that DMSS is W[1]-hard even if we require all the components of the target vector $t$ to be equal. Let $\mathcal{I} = (S, t, k, d)$ be the instance of DMSS. We construct an equivalent instance $\mathcal{I}_{eq} = (S_{eq}, t_{eq}, k + 1, d + 1)$ of UDMSS with $t_{eq} = (d \cdot t_{max}, \ldots, d \cdot t_{max})$, where $t_{max} = \max\{t^i : i \in [k]\}$. $S_{eq}$ is obtained from $S$ by setting the $(k+1)$-th entries equal to $t_{max}$, plus one auxiliary vector to make the target uniform: $S_{eq} = \{(a^1, \ldots, a^k, t_{max}) | (a^1, \ldots, a^k) \in S\} \cup \{b\}$, where $b = (dt_{max} - t^1, \ldots, dt_{max} - t^k, 0)$.

For correctness, assume that $\mathcal{I}$ is a Yes-instance, in particular, we can choose $S'$ with $|S'| = d$ and $\sum_{s \in S'} s \le t$. Then $S'_{eq} = \{(a^1, \ldots, a^k, t_{max}) | (a^1, \ldots, a^k) \in S'\} \cup \{b\}$ witnesses that $\mathcal{I}_{eq}$ is a Yes-instance. For the converse direction, let $\mathcal{I}_{eq}$ be a Yes-instance, we choose $S'_{eq}$ with $|S'_{eq}| = d + 1$ and $\sum_{s \in S'_{eq}} \le t_{eq}$. If $b \notin S_{eq}$, sum of the $(k+1)$-th entries in $S'_{eq}$ would be at least $(d + 1)t_{max}$, so $b$ must belong to $S'_{eq}$. Then $S'_{eq} \setminus \{b\}$ consists of precisely $d$ vectors with sum at most $t_{eq} - b = (t^1, \ldots, t^k, dt_{max})$. Restrictions of these vectors to $k$ first coordinates witness that $\mathcal{I}$ is a Yes-instance. $\qquad\square$

**Theorem 15.** $\mathrm{BNSL}_{\le}^+$ *is* W[1]*-hard when parameterized by the treewidth of the superstructure.*

*Proof.* Let $\mathcal{I} = (S, t, k, d)$ be an instance of UDMSS with $t = (r, \ldots, r)$, and w.l.o.g. assume that $r$ is greater than the parameter $k$. We construct an equivalent instance $(V, \mathcal{F}, \ell, r)$ of $\mathrm{BNSL}_{\le}^+$. Let us start from the vertex set $V$. For every $i \in [k]$, we add to $V$ a vertex $v^i$ corresponding to the $i$-th coordinate of the target vector $t$. Further, for every $s = (s^1, \ldots, s^k) \in S$, we add vertices $a_s, b_s$ and $s^1 + \cdots + s^k$ many vertices $s_j^i$, $i \in [k], j \in [s^i]$. Intuitively, taking $s$ into $S'$ will correspond to adding arcs from $s_j^i$ to $v^i$ for every $i \in [k], j \in [s^i]$. The upper bound $r$ for each coordinate of the sum in $S'$ is captured by allowing $v^i$ to have at most $r$ many parents. Formally, for every $s \in S, i \in [k], j \in [s^i]$ the scores are defined as follows (for convenience we list them as scores per arc): $f(s_j^i v^i) = 2$, $f(b_s a_s) = M_s = 2 \cdot \sum_{i \in [k]} s^i - 1$. We call the arcs mentioned so far *light*. Note that for every fixed $s \in S$, $\sum_{i \in [k]} \sum_{j \in [s^i]} f(s_j^i v^i) = 2 \cdot \sum_{i \in [k]} s^i = M_s + 1$ so the sum of scores of light arcs is $L = \sum_{s \in S}(2M_s + 1)$. We finally set $f(a_s s_j^i) = f(v^i b_s) = L$ for every $s \in S, i \in [k]$ and $j \in [s^i]$. Now the number of arcs yielding the score of $L$ is $m = k|S| + \sum_{s \in S} \sum_{i \in [k]} s^i$; we call these arcs *heavy*. We set the scores of all arcs not mentioned above to zero and we set $\ell = mL + \sum_{s \in S} M_s + d$. This finishes our construction; see Figure 3 for an illustration. Note that the superstructure graph has treewidth of at most $k + 2$: the deletion of vertices $v^i$, $i \in [k]$, makes it acyclic.

19

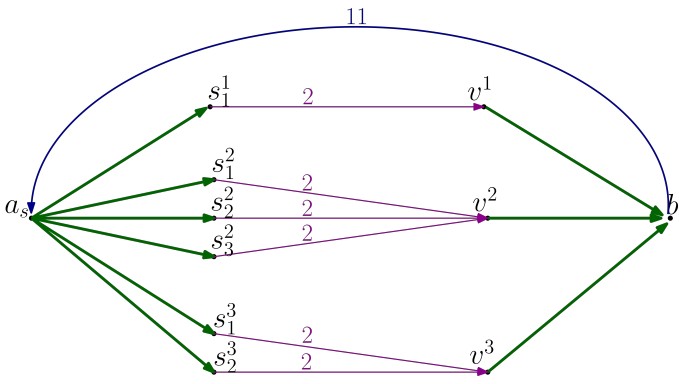

Figure 3: An example of our main gadget encoding the vector $s = (1, 3, 2)$ with $k = 3$. Heavy arcs are marked in green, while purple and blue arcs are light.

Intuitively, the reduction forces a choice between using the blue edge or all the purple edges; the latter case provides a total score that is 1 greater than the former, but is constrained by the upper bound $r$ on the in-degrees of $v^1$, $v^2$, $v^3$.

For correctness, assume that $\mathcal{I} = (S, t, k, d)$ is a Yes-instance of UDMSS, let $S'$ be a subset of $S$ of size $d$ witnessing it. We add all the heavy arcs, resulting in a total score of $mL$. Further, for every $s = (s^1, \ldots, s^k) \in S'$, we add all the arcs $s_j^i v^i$, $i \in [k]$, $j \in [s^i]$, which increases the total score by $M_s + 1$. For every $s \in S \setminus S'$, we add an arc $b_s a_s$, augmenting the total score by $M_s$. Denote the resulting digraph by $D$, then $\texttt{score}(D) = mL + \sum_{s \in S'}(M_s + 1) + \sum_{s \in S \setminus S'} M_s = mL + \sum_{s \in S} M_s + d = \ell$. We proceed by checking parent set sizes. Note that every $s_j^i$ has precisely one incoming arc $a_s s_j^i$ in $D$, every $a_s$ has at most one in-neighbour $b_s$ and in-neighbours of every $b_s$ are $v^i$, $i \in [k]$. Finally, for every $i \in [k]$, $P_D(v^i) = \{s_j^i | s \in S', j \in [s^i]\}$ by construction, so $|P_D(v^i)| = \sum_{s \in S'} s^i \leq r$ as $S'$ is a solution to UDMSS. Therefore all the vertices in $D$ have at most $r$ in-neighbours. It remains to show acyclicity of $D$. As any cycle in the superstructure contains $v^i$ for some $i \in k$, the same holds for any potential directed cycle $C$ in $D$. Two next vertices of $C$ after $v^i$ can be only $b_s$ and $a_s$ for some $s \in S$. In particular, by our construction, $s \in S \setminus S'$. Then, again by construction, $D$ doesn't contain an arc $s_j^i v^i$ for any $i \in [k]$, $j \in [s^i]$, so $v^i$ is not reachable from $a_s$, which contradicts to $C$ being a cycle. Therefore $D$ witnesses that $(V, \mathcal{F}, \ell, r)$ is a Yes-instance.

For the opposite direction, let $(V, \mathcal{F}, \ell, r)$ be a Yes-instance of $\text{BNSL}_{\leq}^+$ and let $D$ be a DAG witnessing this. Then $D$ contains all the heavy arcs. Indeed, sum of scores of all light arcs in $\mathcal{F}$ is $L$, so if at least one heavy arc is not in $A(D)$, then $\texttt{score}(D) \leq (m-1)L + L = mL < \ell$. For every $s \in S$, let $A^s = \{s_j^i v^i | i \in [k], j \in [s^i]\}$. If $D$ doesn't contain an arc $b_s a_s$ and some of arcs from $A^s$, the total score of $A(D) \cap A^s$ is at most $M_s - 1$. In this case we modify $D$ by deletion of $A(D) \cap A^s$ and addition of arc $b_s a_s$, which increases $\texttt{score}(D)$ and may only decrease the parent set sizes of $v^i$, $i \in k$. After these modifications, let $S'' = \{s \in S | D \text{ contains an arc } b_s a_s\}$. Note that whenever $s \in S''$, $D$ cannot contain any of the arcs $s_j^i v^i$, $i \in [k]$, $j \in [s^i]$, as this would result in directed cycle $v^i \to b_s \to a_s \to s_j^i \to v^i$. Therefore for every $s \in S$, $D$ contains either an arc $b_s a_s$ (yielding the score of $M_s$) or all of arcs $s_j^i v^i$, $i \in [k]$, $j \in [s^i]$ (yielding the score of $M_s + 1$ in total), so the sum of scores of light arcs in $D$ is $\sum_{s \in S \setminus S''}(M_s + 1) + \sum_{s \in S''} M_s = \sum_{s \in S} M_s + |S \setminus S''|$, which should be at least $\ell - mL = \sum_{s \in S} M_s + d$. So $|S \setminus S''| \geq d$, we claim that $S' = S \setminus S''$ is a solution to $\mathcal{I} = (S, t, k, d)$. Indeed, for every $i \in [k]$, $r \geq |P_D(v^i)| = |\{s_j^i | s \in S', j \in [s^i]\}| = \sum_{s \in S'} s^i$. □

## 5  Implications for Polytree Learning

Here, we discuss how the results of Sections 3 and 4 can be adapted to POLYTREE LEARNING (PL).

**Theorem 3: Data Reduction.** Recall that the proof of Theorem 3 used two data reduction rules. While Reduction Rule 1 carries over to $\text{PL}^{\neq 0}$, Reduction Rule 2 has to be completely redesigned to preserve the (non-)existence of undirected paths between $a$ and $c$. By doing so, we obtain:

**Theorem 16.** *There is an algorithm which takes as input an instance $\mathcal{I}$ of $\mathrm{PL}^{\neq 0}$ whose superstructure has feedback edge number $k$, runs in time $\mathcal{O}(|\mathcal{I}|^2)$, and outputs an equivalent instance $\mathcal{I}' = (V', \mathcal{F}', \ell')$ of $\mathrm{PL}^{\neq 0}$ such that $|V'| \leq 24k$.*

*Proof.* Note that Reduction Rule 1 acts on the superstructure graph by deleting leaves and therefore preserves not only optimal scores but also (non-)existance of polytrees achiving the scores. Hence we can safely apply the rule to reduce the instance of $\mathrm{PL}^{\neq 0}$. After the exhaustive application, all the leaves of the superstructure graph $G$ are the endpoints of edges in feedback edge set, so there can be at most $2k$ of them. To get rid of long induced paths in $G$, we introduce the following rule:

**Reduction Rule 3.** *Let $a, b_1, \ldots, b_m, c$ be a path in $G$ such that for each $i \in [m]$, $b_i$ has degree precisely 2. For every $B \subseteq \{a, c\}$ and $p \in \{0, 1\}$, let $\ell_p(B)$ be the maximum sum of scores that can be achieved by $b_1, \ldots, b_m$ under the conditions that (1) there exists an undirected path between $b_1$ and $b_m$ if and only if $p = 1$; (2) $b_1$ (and analogously $b_m$) takes $a$ ($c$) into its parent set if and only if $a \in B$ ($c \in B$).*

*We construct a new instance $\mathcal{I}' = (V', \mathcal{F}', \ell)$ as follows:*

- $V' := V \cup \{b, b'_1, b''_1, b'_m, b''_m\} \setminus \{b_1 \ldots b_m\}$;

- $\Gamma_{f'}(b'_1) = \Gamma_{f'}(b''_1) = \Gamma_{f'}(b'_m) = \Gamma_{f'}(b''_m) = \emptyset$;

- *The scores for $a$ (analagously $c$) are obtained from $\mathcal{F}$ by simply replacing every occurence of $b_1$ by $b'_1$ and $b''_1$ ($b_m$ by $b'_m$ and $b''_m$), formally:*

  - $\Gamma_{f'}(a)$ *is a union of* $\{P \in \Gamma_f(a) | b_1 \notin P\}$, *where* $f'_a(P) := f_a(P)$ *and* $\{P \setminus b_1 \cup \{b'_1, b''_1\} | b_1 \in P, P \in \Gamma_f(a)\}$, *where* $f'_a(P \setminus b_1 \cup \{b'_1, b''_1\}) := f_a(P)$;
  - $\Gamma_{f'}(c)$ *is a union of* $\{P \in \Gamma_f(c) | b_m \notin P\}$, *where* $f'_c(P) := f_c(P)$, *and* $\{P \setminus b_m \cup \{b'_m, b''_m\} | b_m \in P, P \in \Gamma_f(c)\}$, *where* $f'_c(P \setminus b_m \cup \{b'_m, b''_m\}) := f_c(P)$.

- $\Gamma_{f'}(b)$ *consists of eight sets, yielding corresponding scores $f'_b$:* $\{a, c, b'_1, b''_1, b'_m, b''_m\} \to l_1(\{a, c\})$, $\{b'_1, b''_1, b'_m, b''_m\} \to l_0(\{a, c\})$, $\{b'_1, b'_m\} \to l_1(\emptyset)$, $\emptyset \to l_0(\emptyset)$, $\{a, b'_1, b''_1, b'_m\} \to l_1(\{a\})$, $\{b'_1, b''_1\} \to l_0(\{a\})$, $\{b'_m, b''_m\} \to l_1(\{c\})$, $\{b'_1, b'_m, b''_m, c\} \to l_0(\{c\})$.

Parent sets of $b$ are defined in a way to cover all the possible configurations on solutions to $\mathcal{I}$ restricted to $a, b_1, \ldots, b_m, c$; the corresponding scores of $b$ are intuitively the sums of scores that $b_i$, $i \in [m]$, receive in the solutions. The eight cases that may arise are illustrated in Figure 4.

**Claim 2.** *Reduction Rule 3 is safe.*

*Proof.* We will show that a score of at least $\ell$ can be achieved in the original instance $\mathcal{I}$ if and only if a score of at least $\ell$ can be achieved in the reduced instance $\mathcal{I}'$.

Assume that $D$ is a polytree that achieves a score of $\ell$ in $\mathcal{I}$. We will construct a polytree $D'$, called the *reduct* of $D$, with $f'(D') \geq \ell$. To this end, we first modify $D$ by removing the vertices $b_1, \ldots, b_m$ and adding $b, b'_1, b''_1, b'_m, b''_m$. We also add arcs $b'_1 a$ and $b''_1 a$ ($b'_m c$ and $b''_m c$ correspondingly) if and only if $b_1 a \in A(D)$ ($b_m c \in A(D)$). Let us denote the DAG obtained at this point $D^*$. Note that scores of $a$ and $c$ in $D^*$ are the same as in $D$. Further modifications of $D^*$ depend only on $D[a, b_1 \ldots b_m, c]$ and change only the parent set of $b$. We distinguish the 8 cases listed below (see also Figure 4):

- case 1.1 (1.2): $ab_1, cb_m \in A(D)$, $b_1$ and $b_m$ are (not) connected by path in $D$. We add incoming arcs to $b$ from $a, c, b'_1, b''_1, b'_m, b''_m$ ($b'_1, b''_1, b'_m, b''_m$ only) resulting in $f'_b(P_{D'}(b)) = l_1(\{a, c\})$ ($f'_b(P_{D'}(b)) = l_0(\{a, c\})$).

- case 2.1 (2.2): $ab_1, cb_m \notin A(D)$, $b_1$ and $b_m$ are (not) connected by path in $D$. We add incoming arcs to $b$ from $b'_1$ and $b'_m$ (leave $D^*$ unchanged) yielding $f'_b(P_{D'}(b)) = l_1(\emptyset)$ ($f'_b(P_{D'}(b)) = l_0(\emptyset)$).

- case 3.1 (3.2): $ab_1 \in A(D)$, $cb_m \notin A(D)$, $b_1$ and $b_m$ are (not) connected by path in $D$. We add incoming arcs to $b$ from $a, b'_1, b''_1, b'_m$ ($b'_1$ and $b''_1$ only), then $f'_b(P_{D'}(b)) = l_1(\{a\})$ ($f'_b(P_{D'}(b)) = l_0(\{a\})$).

- case 4.1 (4.2): $ab_1 \notin A(D)$, $cb_m \in A(D)$, $b_1$ and $b_m$ are (not) connected by path in $D$. The cases are symmetric to 3.1 (3.2)

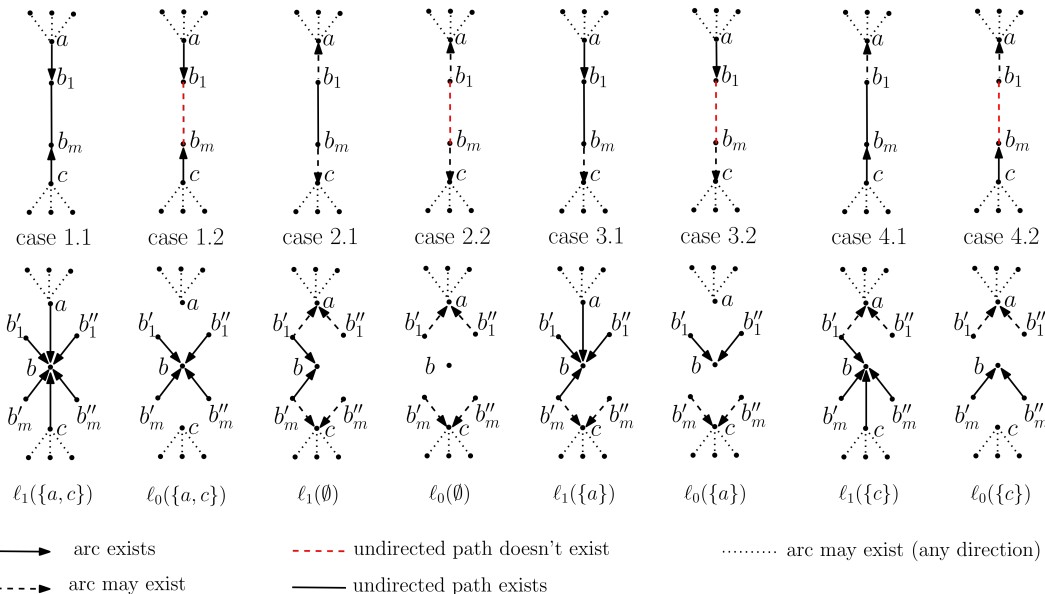

Figure 4: Top: The eight possible scenarios for solutions to $\mathcal{I}$. Bottom: The corresponding arcs in the gadget after the application of Reduction Rule 2' (the scores of $b$ are specified below).

Note that $D'$ contains a path between $a$ and $c$ if and only if $D$ does. By definition of $l_0$ and $l_1$, the score of $b$ in $D'$ is at least as large as the sum of scores of $b_i$, $i \in [m]$, in $D$. Moreover, each vertice in $V(D) \cap V(D')$ receives equal scores in $D$ and $D'$. Hence $D'$ is a polytree with $f'(D') \geq \ell$, as desired. ith a score at least $\ell$. For the converse direction, note that the polytrees constructed in cases 1.1-4.2 cover all optimal configurations which may arise in $\mathcal{I}'$: if there is a polytree $D''$ in $\mathcal{I}'$ with a score of $\ell'$, we can always modify it to a polytree $D'$ with a score of at least $\ell'$ such that $D'[a, b_1', b_1'', b, b_m', b_m'', c]$ has one of the forms depicted at the bottom line of the figure. But every such $D'$ is a reduct of some polytree $D$ of the original instance with the same score. $\qquad\square$

We apply Reduction Rule 3 exhaustively, until there is no more path to shorten. Bounds on the running time of the procedure and size of the reduced instance can be obtained similarly to the case of $\text{BNSL}^{\neq 0}$. In particular, every long path is replaced with a set of 5 vertices, resulting in at most $4k + 4k \cdot 5 = 24k$ vertices. $\qquad\square$

**Theorem 6: Fixed-parameter tractability.** Analogously to $\text{BNSL}^{\neq 0}$ a data reduction procedure as the one provided in Theorem 16 does not exist for $\text{PL}^{\neq 0}$ parametrized by lfen unless $\mathsf{NP} \subseteq \mathsf{co\text{-}NP/poly}$, since the lower-bound result provided in Theorem 11 can be straightforwardly adapted to $\text{PL}^{\neq 0}$. But similarly as for BNSL we can provide an FPT algorithm using the same ideas as in the proof of Theorem 6. The algorithm proceeds by dynamic programming on the spanning tree $T$ of $G$ with $\text{lfen}(G, T) = \text{lfen}(G) = k$. The records will, however, need to be modified: for each vertex $v$, instead of the path-connectivity relation on $\delta(v)$, we store connected components of the *inner boundary* $\delta(v) \cap V_v$ and incoming arcs to $T_v$. We provide a full description of the algorithm below.

**Theorem 17.** $\text{PL}^{\neq 0}$ *is fixed-parameter tractable when parameterized by the local feedback edge number of the superstructure.*

*Proof.* As before, given an instance $\mathcal{I}$ with a superstructure graph $G = G_{\mathcal{I}}$ such that $\text{lfen}(G) = k$, we start from computing the spanning tree $T$ of $G$ with $\text{lfen}(G, T) = \text{lfen}(G) = k$; pick a root $r$ in $T$. We keep all the notations $T_v, V_v, \bar{V}_v, \delta(v)$ for $v \in V(T)$ from the subsection 3.2. In addition, we define the *inner boundary* of $v \in V(T)$ to be $\delta_{in}(v) := \delta(v) \cap V_v$ i.e. part of boundary that belongs to subtree of $T$ rooted in $v$. The remaining part we call the *outer boundary* of $v$ and denote by $\delta_{out}(v) := \delta(v) \setminus \delta_{in}(v)$. For any set $A$ of arcs, we define $\widetilde{A} = \{uv | uv \in A \text{ or } vu \in A\}$. Obviously,

934 the claims of Observation 8 still hold. Moreover, for every closed $v$, $\delta_{in}(v)$ contains only $v$ itself and
935 $\delta_{out}(v)$ is either the parent of $v$ in $T$ or $\emptyset$ (for $v = r$).

936 Let $R_v$ be binary relation on $\delta_{in}(v)$, $A_v \subseteq \delta_{out}(v) \times \delta_{in}(v)$, $s_v$ is integer. Then $(R_v, A_v, s_v)$ is a
937 *record* for $v$ if and only if there exist a polytree $D$ on $\bar{V}_v$ with all arcs oriented inside $V_v$ such that:

938 - $A_v = \{xy \in A(D) \mid x \in \delta_{out}(v), y \in \delta_{in}(v)\}$

939 - $R_v = \{xy \mid x, y \in \delta_{in}(v)$ are in the same connected component of $D[V_v]\}$

940 - $s_v = \sum_{u \in V_v} f_u(P_D(u))$

941 Note that $R_v$ is an equivalence relation on $\delta_{in}(v)$, number of its equivalence classes is equal to
942 number of connected components of $D[V_v]$ that intersect $\delta(v)$.

943 Record $(R_v, A_v, s_v)$ is called *valid* if and only if $s_v$ is maximal for fixed $R_v$, $A_v$ among all the records
944 for $v$. Denote by $\mathcal{R}(v)$ the set of all valid records for $v$, then $|\mathcal{R}(v)| \leq 2^{(2k+2)^2}$. Indeed, $R_v$ and $A_v$
945 can be uniquely determined by the choice of some relation on $\delta(v) \times \delta(v)$. As $|\delta(v)| \leq 2k + 2$, there
946 are at most $2^{(2k+2)^2}$ possible relations.

947 The root $r$ of $T$ has a single valid record $(\emptyset, \emptyset, s_{\mathcal{I}})$, where $s_{\mathcal{I}}$ is the maximum score that can be
948 achieved by a solution to $\mathcal{I}$. For any closed $v \neq r$, $\mathcal{R}(v)$ consists of precisely two valid records: one
949 for $A_v = \emptyset, R_v = \{vv\}$ and another for $A_v = \{wv\}, R_v = \{vv\}$, where $w$ is a parent of $v$ in $T$.

950 We proceed by computing our records in a leaf-to-root fashion along $T$.

951 Let $v$ be a leaf. Start by innitiating $\mathcal{R}^*(v) := \emptyset$, then for each $P \in \Gamma_f(v)$ add to $\mathcal{R}^*(v)$ the triple
952 $(\{vv\}, \{uv \mid u \in P\}, f_v(P))$. Note that $\mathcal{R}^*(v)$ is by definition precisely the set of all records for $v$, so
953 we can correctly set $\mathcal{R}(v) = \{(R_v, A_v, s_v) \in \mathcal{R}^*(v) \mid s_v$ is maximal for fixed $R_v, A_v\}$.

954 Assume that $v$ has $m$ children $\{v_i : i \in [m]\}$ in $T$, where $v_i$, $i \in [t]$, are open and $v_i$, $i \in [m] \setminus [t]$,
955 are closed. The following claim shows how (and under which conditions) the records of children of $v$
956 can be composed into a record of $v$.

957 **Claim 3.** *Let $P \in \Gamma_f(v)$, $D_0$ is a polytree on $V_0 = v \cup P$ with arc set $A_0 = \{uv \mid u \in P\}$, $(R_i, A_i, s_i)$*
958 *are records for $v_i$ witnessed by $D_i$, $i \in [m]$. Let $A_{loc}^{in}$ be the set of arcs in $\bigcup_{i \in [t]_0} A_i$ which have both*
959 *endpoints in $V_v$, $R = \texttt{trcl}(\widetilde{A}_{loc}^{in} \cup \bigcup_{i \in [t]_0} R_i)$. Then $D = \cup_{i=0}^m D_i$ is a polytree if and only if the*
960 *following two conditions hold:*

961     *1. $A_i = \emptyset$ for each closed child $v_i \in P$.*

962     *2. $\sum_{i=0}^t N_i - |A_{loc}^{in}| - \sum_{y \in Y}(n_y - 1) = N$, where*

963         - *$N$ is the number of equivalence classes in $\texttt{trcl}(\bigcup_{i \in [t]_0}(\widetilde{A}_i \cup R_i))$*
964         - *$N_i$ is the number of equivalence classes in $R_i$, $i \in [t]$*
965         - *$Y$ is the set of endpoints of arcs in $\bigcup_{i \in [t]_0} A_i$ which don't belong to any $V_i$, $i \in [m]$.*
966         - *For every $y \in Y$, $n_y$ is the number of arcs in $A_0 \cup ... \cup A_t$ having endpoint $y$.*

967 *In this case $D$ witnesses the record $(R_v, A_v, s_v)$, where:*
968 *$R_v = R|_{\delta_{in}(v) \times \delta_{in}(v)}$, $A_v = (\bigcup_{i \in [t]_0} A_i)|_{\delta_{out}(v) \times \delta_{in}(v)}$, $s_v = \sum_{i=0}^m s_i + f_v(P)$.*

969 *If $(R_v, A_v, s_v) \in \mathcal{R}(v)$, then $(R_i, A_i, s_i) \in \mathcal{R}(v_i)$, $i \in [m]$. Moreover, for any closed child $v_i \notin P$,*
970 *there is no $(R_i', A_i', s_i') \in \mathcal{R}(v_i)$ with $s_i' > s_i$.*

971 We will prove the claim at the end, let us show how it can be exploited to compute valid records of
972 $v$. We start from initial setting $\mathcal{R}^*(v) := \emptyset$, then branch over all parent sets $P \in \Gamma_f(v)$ and triples
973 $(R_i, A_i, s_i) \in \mathcal{R}(v_i)$ for open children $v_i$. For each closed child $v_i \notin P$ take $(R_i, A_i, s_i) \in \mathcal{R}(v_i)$
974 with maximal $s_i$, for each closed child $v_i \in P$ take $(R_i, A_i, s_i) \in \mathcal{R}(v_i)$ with $A_i = \emptyset$. Now the first
975 condition of Claim3 holds, if the second one holds as well, we add to $\mathcal{R}^*(v)$ the triple $(R_v, A_v, s_v)$.

According to Claim 3, $\mathcal{R}^*(v)$ computed in such a way consists only of records for $v$ and, in particular, contains all the valid records. Therefore we can correctly set $\mathcal{R}(v) = \{(R_v, A_v, s_v) \in \mathcal{R}^*(v) | s_v$ is maximal for fixed $R_v, A_v\}$.

To construct $\mathcal{R}^*(v)$ for node $v$ with children $v_i$, $i \in [m]$, we branch over at most $n$ possible parent sets of $v$ and at most $2^{(2k+2)^2}$ valid records for every open child of $v$. Number of open children is bounded by $2k$, so we have at most $\mathcal{O}((2^{(2k+2)^2})^{2k} \cdot n) \le 2^{\mathcal{O}(k^3)} \cdot n$ branches. In a fixed branch we compute scores for closed children in $\mathcal{O}(n)$, application of Claim 3 requires time polynomial in $k$. So $\mathcal{R}^*(v)$ is computed in time $2^{\mathcal{O}(k^3)} \cdot n^2$ that majorizes running time for leaves. As the number of vertices in $T$ is at most $n$, total running time of the algorithm is $2^{\mathcal{O}(k^3)} \cdot n^3$ assuming that $T$ is given as a part of the input.

*proof of Claim 3* ($\Leftarrow$). We start from checking whether $D = \cup_{i=0}^m D_i$ is a polytree. As the first condition implies that a polytree of every closed child $v_i$ is connected to the rest of $D$ by at most one arc $v_i v$ or $v v_i$, it is sufficient to check whether $D^t = \cup_{i=0}^t D_i$ is polytree. Number of connected components of $D^t$ is $N' + N$, where $N'$ is the total number of connected components of $D_i$ that don't intersect $\delta(v_i)$, $i \in [t]$. Note that $D^t$ can be constructed as follows:

1. Take a disjoint union of polytrees $D_i' = D_i[V_i]$, $i \in [t]_0$, then the resulting polytree has $N' + \sum_{i=0}^t N_i$ connected components.

2. Add arcs between $D_i'$ and $D_j'$ that occur in $D$ for every $i, j \in [t]_0$, i.e. the arcs specified by $A_{loc}^{in}$. Resulting digraph is a polytree if and only if every added arc decreases the number of connected components by 1, i.e. the number of connected components after this step is $N' + \sum_{i=0}^t N_i - |A_{loc}^{in}|$.

3. Add all remaining vertices $y$ of $D$ together with their adjacent arcs in $D$. Note that such $y$ precisely form the set $Y$, so $D^t$ is a polytree if and only if we obtained a polytree after the previous step and every $y \in Y$ decreased it's number of connected components by $(n_y - 1)$, i.e. the number $N' + N$ of connected components in $D^t$ is equal to $N' + \sum_{i=0}^t N_i - |A_{loc}^{in}| - \sum_{y \in Y}(n_y - 1)$. But this is precisely the condition 2 of the claim.

Now, assuming that $D$ is a polytree, we will show that it witnesses $(R_v, A_v, s_v)$. Parent sets of vertices from each $V_i$ in $D$ are the same as in $D_i$, parent set of $v$ in $D$ is $P$. So $s_v = \sum_{i=0}^m s_i + f_v(P)$ is indeed the sum of scores over $V_v$ in $D$.

There are two kinds of arcs in $D$ starting outside of $V_v$: incoming arcs to $v$ and incoming arcs to the subtrees of open children. Thus $A(D)|_{\delta_{out}(v) \times \delta_{in}(v)} = (\bigcup_{i \in [t]_0} A_i)|_{\delta_{out}(v) \times \delta_{in}(v)} = A_v$.

Take any $u, w \in \delta_{in}(v), u \ne w$, note that $u$ and $w$ can not belong to subtrees of closed children. So $u$ and $w$ are in the same connected component of $D[V_v]$ if and only if they are connected by some undirected path $\pi$ in the skeleton of $D$ using only vertices from $D^t \cap V_v$. In this case $R_i$ captures the segmens of $\pi$ which are completely contained in $D_i[V_i], i \in [t]$. Rest of edges in $\pi$ either connect $v$ to some $V_i$, $i \in [t]$, or have enpoints in different $V_i$ and $V_j$ for some $i, j \in [t]$. Edges of this kind precisely form the set $\widetilde{A}_{loc}^{in}$, so $uw$ belongs to $R = \texttt{trcl}(\bigcup_{i \in [t]} R_i \cup \widetilde{A}_{loc}^{in})$. Therefore $R_v = R|_{\delta_{in}(v) \times \delta_{in}(v)}$ indeed represents connected components of $\delta_{in}(v)$ in $D[V_v]$.

($\Rightarrow$) Condition 1 obviously holds, otherwise $D$ would contain a pair of arcs with the same endpoints and different directions. In ($\Leftarrow$) we actually showed the necessity of condition 2 when 1 holds.

For the last statement, assume that $(R_v, A_v, s_v) \in \mathcal{R}(v)$ but $(R_i, A_i, s_i) \notin \mathcal{R}(v_i)$ for some $i$. Then there is $(R_i, A_i, s_i + \Delta) \in \mathcal{R}(v_i)$ for some $\delta > 0$. Let $D_i'$ be a witness of $(R_i, A_i, s_i + \Delta)$, then $D' = \bigcup_{j \in [m] \setminus \{i\}} D_j \cup D_i'$ is a polytree witnessing $(R_v, A_v, s_v + \Delta)$. But this contradicts to validity of $(R_v, A_v, s_v)$. By the same arguments records for closed children $v_i \notin P$ are the ones with maximal $s_i$ among two $(R_i, A_i, s_i) \in \mathcal{R}(v_i)$. ∎ □

As for treecut width, we remark that a recent reduction for PL$^{\ne 0}$ [24, Theorem 4.2] immediately implies that the problem is $\mathsf{W}[1]$-hard when parameterized by the treecut width(the superstructure graphs obtained in that reduction have a vertex cover of size bounded in the parameter, and the vertices outside of the vertex cover have degree at most 2).

**Theorem 13: Additive Representation.**    We remark that, like $\mathrm{BNSL}^+$ and $\mathrm{BNSL}^+_{\leq}$, a simple reduction shows that $\mathrm{PL}^+_{\leq}$ is NP-hard for a fixed value of $q$, in this case $q = 1$.

**Theorem 18.** $\mathrm{PL}^+_{\leq}$ *is* NP-*hard when* $q = 1$.

*Proof.* We reduce from the classical HAMILTONIAN PATH problem. Given a graph $G$, we construct an instance $\mathcal{I}$ of $\mathrm{PL}^+_{\leq}$ with $q = 1$ and the same vertex set. Whenever $G$ contains an edge $ab$, we set $f_a(b) = f_b(a) = 1$; all other cost functions are set to 0. $\ell$ is set to $|V| - 1$.

Consider a solution $D$ for $\mathcal{I}$. Since $D$ is a DAG, it must contain a source; by construction, all other vertices in $D$ must have an in-degree of 1. This implies that the arcs of $D$ form a Hamiltonian path in $G$. Conversely, given a Hamiltonian path in $G$, one can construct a solution $D$ by choosing one endpoint of the path as the source and then adding all arcs along the path.    $\square$

Moreover, the dynamic programming algorithm for $\mathrm{BNSL}^+_{\leq}$ parameterized by treewidth and $q$ can be adapted to also solve $\mathrm{PL}^+_{\leq}$. For completeness, we provide a full proof below; however one should keep in mind that the ideas are very similar to the proof of Theorem 13.

**Theorem 19.** $\mathrm{PL}^+$ *is* FPT *when parameterized by the treewidth of the superstructure. Moreover,* $\mathrm{PL}^+_{\leq}$ *is* FPT *when parameterized by q plus the treewidth of the superstructure.*

*Proof.* We begin by proving the latter statement, and will then explain how that result can be straightforwardly adapted to obtain the former. As our initial step, we apply Bodlaender's algorithm [4, 27] to compute a nice tree-decomposition $(\mathcal{T}, \chi)$ of $G_{\mathcal{I}}$ of width $k = \mathrm{tw}(G_{\mathcal{I}})$. We keep the notations $T$, $r$, and $\chi^{\downarrow}_t$ $G^{\downarrow}_t$ from the proof of Theorem 13. For any arc set $A$ we denote $\widetilde{A} = \{uw, wu | uw \in A\}$.

We will design a leaf-to-root dynamic programming algorithm which will compute and store a set of records at each node of $T$, whereas once we ascertain the records for $r$ we will have the information required to output a correct answer. The set of snapshots and structure of records will be the same as in the proof of Theorem 13. However, semantics wil slightly differ: in contrast to information about directed paths via forgotten nodes, con will now specify whether vertices of the bag belong to the same connected component of the partial polytree. Formally, let $\Psi_t$ be the set of all polytrees over the vertex set $\chi^{\downarrow}_t$ with maximal in-degree at most $q$, and let $D_t = (\chi^{\downarrow}_t, A)$ be a polytree in $\Psi_t$. We say that the *snapshot of* $D_t$ in $t$ is the tuple $(\alpha, \beta, p)$ where $\alpha = A_{\chi(t)} \cap A$, $\beta = A_{\chi(t)} \cap \{uw | u \text{ and } w \text{ belong to the same connected component of } D_t\}$ and $p$ specifies numbers of parents of vertices from $\chi(t)$ in $D$, i.e. $p(v) = |\{w \in \chi^{\downarrow}_t | wv \in A\}|$, $v \in \chi(t)$. We will call a connected component of $D_t$ *active* if it intersects $\chi(t)$. Note that the number of equivalence classes of con is equal to the number of active connected components of $D_t$. We are now ready to define the record $\mathcal{R}_t$. For each snapshot $(\mathrm{loc}, \mathrm{con}, \mathrm{inn}) \in S(t)$:

- $\mathcal{R}_t(\mathrm{loc}, \mathrm{con}, \mathrm{inn}) = \perp$ if and only if there exists no polytree in $\Psi_t$ whose snapshot is $(\mathrm{loc}, \mathrm{con}, \mathrm{inn})$, and

- $\mathcal{R}_t(\mathrm{loc}, \mathrm{con}, \mathrm{inn}) = \tau$ if $\exists D_t \in \Psi_t$ such that

    – the snapshot of $D_t$ is $(\mathrm{loc}, \mathrm{con}, \mathrm{inn})$,
    – $\mathtt{score}(D_t) = \tau$, and
    – $\forall D'_t \in \Psi_t$ such that the snapshot of $D'_t$ is $(\mathrm{loc}, \mathrm{con}, \mathrm{inn})$: $\mathtt{score}(D_t) \geq \mathtt{score}(D'_t)$.

Recall that for the root $r \in T$, we assume $\chi(r) = \emptyset$. Hence $\mathcal{R}_r$ is a mapping from the one-element set $\{(\emptyset, \emptyset, \emptyset)\}$ to an integer $\tau$ such that $\tau$ is the maximum score that can be achieved by any polytree $D = (V, A)$ with all in-degrees of vertices upper bounded by $q$. In other words, $\mathcal{I}$ is a YES-instance if and only if $\mathcal{R}_r(\emptyset, \emptyset, \emptyset) \geq \ell$. To prove the theorem, it now suffices to show that the records can be computed in a leaf-to-root fashion by proceeding along the nodes of $T$. We distinguish four cases:

$t$ **is a leaf node.** Let $\chi(t) = \{v\}$. By definition, $S(t) = \{(\emptyset, \emptyset, \emptyset)\}$ and $\mathcal{R}_t(\emptyset, \emptyset, \emptyset) = f_v(\emptyset)$.

$t$ **is a forget node.** Let $t'$ be the child of $t$ in $\mathcal{T}$ and let $\chi(t) = \chi(t') \setminus \{v\}$. We initiate by setting $\mathcal{R}^0_t(\mathrm{loc}, \mathrm{con}, \mathrm{inn}) = \perp$ for each $(\mathrm{loc}, \mathrm{con}, \mathrm{inn}) \in S(t)$.

For each $(\mathrm{loc}', \mathrm{con}', \mathrm{inn}') \in S(t')$, let $\mathrm{loc}_v$, $\mathrm{con}_v$ be the restrictions of $\mathrm{loc}'$, $\mathrm{con}'$ to tuples containing $v$. We now define $\mathrm{loc} = \mathrm{loc}' \setminus \mathrm{loc}_v$, $\mathrm{con} = \mathrm{con}' \setminus \mathrm{con}_v$, $\mathrm{inn} = \mathrm{inn}'|_{\chi(t)}$ and set $\mathcal{R}_t^0(\mathrm{loc}, \mathrm{con}, \mathrm{inn}) := \max(\mathcal{R}_t^0(\mathrm{loc}, \mathrm{con}, \mathrm{inn}), \mathcal{R}_{t'}(\mathrm{loc}', \mathrm{con}', \mathrm{inn}'))$, where $\perp$ is assumed to be a minimal element. Finally we set $\mathcal{R}_t = \mathcal{R}_t^0$, correctness can be argued analogously to the case of $\mathrm{BNSL}_{\leq}^+$.

$t$ **is an introduce node.** Let $t'$ be the child of $t$ in $\mathcal{T}$ and let $\chi(t) = \chi(t') \cup \{v\}$. We initiate by setting $\mathcal{R}_t^0(\mathrm{loc}, \mathrm{con}, \mathrm{inn}) = \perp$ for each $(\mathrm{loc}, \mathrm{con}, \mathrm{inn}) \in S(t)$.

For each $(\mathrm{loc}', \mathrm{con}', \mathrm{inn}') \in S(t')$ and each $Q \subseteq \{ab \in A_{\chi(t)} \mid \{a, b\} \cap \{v\} \neq \emptyset\}$, we define:

- $\mathrm{loc} := \mathrm{loc}' \cup Q$

- $\mathrm{con} := \mathtt{trcl}(con' \cup \widetilde{Q})$

- $\mathrm{inn}(x) := \mathrm{inn}'(x) + |\{y \in \chi(t) | yx \in Q\}|$ for every $x \in \chi(t) \setminus \{v\}$
  $\mathrm{inn}(v) := |\{y \in \chi(t) | yv \in Q\}|$

Let $N$ and $N'$ be the numbers of equivalence classes in $con$ and $con'$ correspondingly. If $N \neq N' + 1 - |Q|$ or $\mathrm{inn}(x) > q$ for some $x \in \chi(t)$, discard this branch. Otherwise, let $\mathcal{R}_t^0(\mathrm{loc}, \mathrm{con}, \mathrm{inn}) := \max(\mathcal{R}_t^0(\mathrm{loc}, \mathrm{con}, \mathrm{inn}), \mathtt{new})$ where $\mathtt{new} = \mathcal{R}_{t'}(\mathrm{loc}', \mathrm{con}', \mathrm{inn}') + \sum_{ab \in Q} f_b(a)$. As before, $\perp$ is assumed to be a minimal element here.

Consider our final computed value of $\mathcal{R}_t^0(\mathrm{loc}, \mathrm{con}, \mathrm{inn})$ for some $(\mathrm{loc}, \mathrm{con}, \mathrm{inn}) \in S(t)$.

For correctness, assume that $\mathcal{R}_t^0(\mathrm{loc}, \mathrm{con}, \mathrm{inn}) = \tau$ for some $\tau \neq \perp$ and is obtained from $(\mathrm{loc}', \mathrm{con}', \mathrm{inn}')$, $Q$ defined as above. Then $\mathcal{R}_{t'}(\mathrm{loc}', \mathrm{con}', \mathrm{inn}') = \tau - \sum_{ab \in Q} f_b(a)$. Construct a directed graph $D$ from the witness $D'$ of $\mathcal{R}_{t'}(\mathrm{loc}', \mathrm{con}', \mathrm{inn}')$ by adding $v$ and the arcs specified in $Q$. The equality $N = N' + 1 - |Q|$ garantees that every such arc decreases the number of active connected components by one, so $D$ is a polytree. Moreover, $\mathrm{inn}(x) \leq q$ for every $x \in \chi(t)$ and the rest of vertices have in $D$ the same parents as in $D'$, so $D \in \Psi_t$. In particular, $(\mathrm{loc}, \mathrm{con}, \mathrm{inn})$ is a snapshot of $D$ in $t$ and $D$ witnesses $\mathcal{R}_t(\mathrm{loc}, \mathrm{con}, \mathrm{inn}) \geq \mathcal{R}_{t'}(\mathrm{loc}', \mathrm{con}', \mathrm{inn}') + \sum_{ab \in Q} f_b(a) = \tau$.

On the other hand, if $\mathcal{R}_t(\mathrm{loc}, \mathrm{con}, \mathrm{inn}) = \tau$ for some $\tau \neq \perp$, then there must exist a polytree $D = (\chi_t^{\downarrow}, A)$ in $\Psi_t$ that achieves a score of $\tau$. Let $Q$ be the restriction of $A$ to arcs containing $v$, and let $D' = (\chi_t^{\downarrow} \setminus v, A \setminus Q)$, clearly $D' \in \Psi_{t'}$. Let $(\mathrm{loc}', \mathrm{con}', \mathrm{inn}')$ be the snapshot of $D'$ at $t'$. Observe that $\mathrm{loc} = \mathrm{loc}' \cup Q$, $\mathrm{con} = \mathtt{trcl}(con' \cup \widetilde{Q})$, $\mathrm{inn}$ differs from $\mathrm{inn}'$ by the numbers of incoming arcs in $Q$ and the score of $D'$ is precisely equal to the score $\tau$ of $D$ minus $\sum_{(a,b) \in Q} f_b(a)$. Therefore $\mathcal{R}_{t'}(\mathrm{loc}', \mathrm{con}', \mathrm{inn}') \geq \tau - \sum_{(a,b) \in Q} f_b(a)$ and in the algorithm $\mathcal{R}_t^0(\mathrm{loc}, \mathrm{con}, \mathrm{inn}) \geq \mathcal{R}_{t'}(\mathrm{loc}', \mathrm{con}', \mathrm{inn}') + \sum_{(a,b) \in Q} f_b(a) \geq \tau$. Equality then follows from the previous direction of the correctness argument.

Hence, at the end of our procedure we can correctly set $\mathcal{R}_t = \mathcal{R}_t^0$.

$t$ **is a join node.** Let $t_1, t_2$ be the two children of $t$ in $\mathcal{T}$, recall that $\chi(t_1) = \chi(t_2) = \chi(t)$. We initiate by setting $\mathcal{R}_t^0(\mathrm{loc}, \mathrm{con}, \mathrm{inn}) := \perp$ for each $(\mathrm{loc}, \mathrm{con}, \mathrm{inn}) \in S(t)$.

Let us branch over each $\mathrm{loc}, \mathrm{con}_1, \mathrm{con}_2 \subseteq A_{\chi(t)}$ and $\mathrm{inn}_1, \mathrm{inn}_2 : \chi(t) \to [q]_0$. For every $b \in \chi(t)$ set $\mathrm{inn}(b) = \mathrm{inn}_1(b) + \mathrm{inn}_2(b) - |\{a | ab \in \mathrm{loc}\}|$. Let $N_1$ and $N$ be the numbers of equivalence classes in $\mathrm{con}_1$ and $\mathtt{trcl}(\mathrm{con}_1 \cup \mathrm{con}_2)$ correspondingly. If:

- $\mathrm{con}_1 \cap \mathrm{con}_2 \neq \mathtt{trcl}(\widetilde{\mathrm{loc}})$, and/or

- $N - N_1 \neq \frac{1}{2}|\mathrm{con}_2 \setminus \mathtt{trcl}(\widetilde{\mathrm{loc}})|$, and/or

- $\mathcal{R}_{t_1}(\mathrm{loc}, \mathrm{con}_1, \mathrm{inn}_1) = \perp$, and/or

- $\mathcal{R}_{t_2}(\mathrm{loc}, \mathrm{con}_2, \mathrm{inn}_2) = \perp$, and/or

- $\mathrm{inn}(b) > q$ for some $b \in \chi(t)$

then discard this branch. Otherwise, set $\mathrm{con} = \mathtt{trcl}(\mathrm{con}_1 \cup \mathrm{con}_2)$, $\mathtt{doublecount} = \sum_{ab \in \mathrm{loc}} f_b(a)$ and $\mathtt{new} = \mathcal{R}_{t_1}(\mathrm{loc}, \mathrm{con}_1) + \mathcal{R}_{t_2}(\mathrm{loc}, \mathrm{con}_2) - \mathtt{doublecount}$. We then set $\mathcal{R}_t^0(\mathrm{loc}, \mathrm{con}, \mathrm{inn}) := \max(\mathcal{R}_t^0(\mathrm{loc}, \mathrm{con}, \mathrm{inn}), \mathtt{new})$ where $\perp$ is once again assumed to be a minimal element.

At the end of this procedure, we set $\mathcal{R}_t = \mathcal{R}_t^0$.

For correctness, assume that $\mathcal{R}_t^0(\mathrm{loc}, \mathrm{con}, \mathrm{inn}) = \tau \neq \perp$ is obtained from $\mathrm{loc}, \mathrm{con}_1, \mathrm{con}_2, \mathrm{inn}_1, \mathrm{inn}_2$ as above. Let $D_1 = (\chi_{t_1}^\downarrow, A_1)$ and $D_2 = (\chi_{t_2}^\downarrow, A_2)$ be polytrees witnessing $\mathcal{R}_{t_1}(\mathrm{loc}, \mathrm{con}_1, \mathrm{inn}_1)$ and $\mathcal{R}_{t_2}(\mathrm{loc}, \mathrm{con}_2, \mathrm{inn}_2)$ correspondingly. Recall from the proof of Theorem 13 that common vertices of $D_1$ and $D_2$ are precisely $\chi(t)$, $\mathrm{loc} = A_1 \cap A_2$ and $\mathrm{inn}$ specifies the number of parents of every $b \in \chi(T)$ in $D = D_1 \cup D_2$. Numbers of active connected components of $D$ and $D_1$ are $N$ and $N_1$ correspondingly. Observe that $D$ can be constructed from $D_1$ by adding vertices and arcs of $D_2$. As $\mathrm{con}_1 \cap \mathrm{con}_2 = \mathtt{trcl}(\widetilde{\mathrm{loc}})$, we can only add a path between vertices in $\chi(t)$ if it didn't exist in $D_1$. Hence $\frac{1}{2}|\mathrm{con}_2 \setminus \mathtt{trcl}(\widetilde{\mathrm{loc}})|$ specifies the number of paths between vertices in $\chi(t)$ via forgotten vertices of $\chi_{t_2}^\downarrow$. The equality $N_1 - N = \frac{1}{2}|\mathrm{con}_2 \setminus \mathtt{trcl}(\widetilde{\mathrm{loc}})|$ means that adding every such path decreases the number of active connected components of $D_1$ by one. As $D_1$ is a polytree, $D$ is a polytree as well, so $D \in \Psi_t$. The snapshot of $D$ in $t$ is $(\mathrm{loc}, \mathrm{con}, \mathrm{inn})$ and $\mathtt{score}(D) = \sum_{ab \in A(D)} f_b(a) = \sum_{ab \in A_1} f_b(a) + \sum_{ab \in A_2} f_b(a) - \sum_{ab \in \mathrm{loc}} f_b(a) = \mathtt{score}(D_1) + \mathtt{score}(D_2) - \mathtt{doublecount} = \mathcal{R}_{t_1}(\mathrm{loc}, \mathrm{con}_1, \mathrm{inn}_1) + \mathcal{R}_{t_2}(\mathrm{loc}, \mathrm{con}_2, \mathrm{inn}_2) - \mathtt{doublecount} = \tau$. So $D$ witnesses that $\mathcal{R}_t(\mathrm{loc}, \mathrm{con}, \mathrm{inn}) \geq \tau$.

For the converse, assume that $\mathcal{R}_t(\mathrm{loc}, \mathrm{con}, \mathrm{inn}) = \tau \neq \perp$ and $D$ is a polytree witnessing this. Let $D_1$ and $D_2$ be restrictions of $D$ to $\chi_{t_1}^\downarrow$ and $\chi_{t_2}^\downarrow$ correspondingly, then $A(D_1) \cap A(D_2) = \mathrm{loc}$, in particular $D = D_1 \cup D_2$. Let $(\mathrm{loc}, \mathrm{con}_i, \mathrm{inn}_i)$ be the snapshot of $D_i$ in $t_i$, $i = 1, 2$. $D = D_1 \cup D_2$ is a polytree, so any pair of vertices in $\chi(t)$ can not be connected by different paths in $D_1$ and $D_2$, i.e. $\mathrm{con}_1 \cap \mathrm{con}_2 = \mathtt{trcl}(\widetilde{\mathrm{loc}})$. By the procedure of our algorithm, $\mathcal{R}_t^0(\mathrm{loc}, \mathrm{con}, \mathrm{inn}) \geq \mathcal{R}_{t_1}(\mathrm{loc}, \mathrm{con}_1, \mathrm{inn}_1) + \mathcal{R}_{t_2}(\mathrm{loc}, \mathrm{con}_2, \mathrm{inn}_2) - \mathtt{doublecount} \geq \mathtt{score}(D_1) + \mathtt{score}(D_2) - \sum_{ab \in \mathrm{loc}} f_b(a) = \mathtt{score}(D) = \tau$.

Hence the resulting record $\mathcal{R}_t$ is correct, which concludes the correctness proof of the algorithm.

Since the nice tree-decomposition $\mathcal{T}$ has $\mathcal{O}(n)$ nodes, the runtime of the algorithm is upper-bounded by $\mathcal{O}(n)$ times the maximum time required to process each node. This is dominated by the time required to process join nodes, for which there are at most $(2^{k^2})^3((q+1)^k)^2 = 8^{k^2} \cdot (q+1)^{2k}$ branches corresponding to different choices of $\mathrm{loc}, \mathrm{con}_1, \mathrm{con}_2, \mathrm{inn}_1, \mathrm{inn}_2$. Constructing $\mathtt{trcl}(\mathrm{con}_1 \cup \mathrm{con}_2)$ and computing numbers of active connected components can be done in time $\mathcal{O}(k^3)$. Computing $\mathtt{doublecount}$ and $\mathrm{inn}$ takes time at most $\mathcal{O}(k^2)$. So the record for a join node can be computed in time $2^{\mathcal{O}(k^2)} \cdot q^{\mathcal{O}(k)}$. Hence, after we have computed a width-optimal tree-decomposition for instance by Bodlaender's algorithm [4], the total runtime of the algorithm is upper-bounded by $2^{\mathcal{O}(k^2)} \cdot q^{\mathcal{O}(k)} \cdot n$.

Finally, to obtain the desired result for $\mathrm{PL}^+$, we can simply adapt the above algorithm by disregarding the entry $\mathrm{inn}$ and disregard all explicit bounds on the in-degrees (e.g., in the definition of $\Psi_t$). The runtime for this dynamic programming procedure is then $2^{\mathcal{O}(k^2)} \cdot n$. $\qquad \square$

The situation is, however, completely different for $\mathrm{PL}^+$: unlike $\mathrm{BNSL}^+$, this problem is in fact polynomial-time tractable. Indeed, it admits a simple reduction to the classical minimum edge-weighted spanning tree problem.

**Observation 20.** $\mathrm{PL}^+$ *is polynomial-time tractable.*

*Proof.* Consider an the superstructure graph $G$ of an instance $\mathcal{I} = (V, \mathcal{F}, \ell)$ of $\mathrm{PL}^+$ where we assign to each edge $ab \in E(G)$ a weight $w(ab) = \max f_a(b), f_b(a)$, and recall that we can assume w.l.o.g. that $G$ is connected. Each spanning tree $T$ of $G$ with weight $p$ can be transformed to a DAG $D$ over $V$ with a score of $p$ and whose skeleton is a tree by simply replacing each edge $ab$ with the arc $ab$ or $ba$, depending on which achieves a higher score. On the other hand, each solution to $\mathcal{I}$ can be transformed into a spanning tree $T$ of the same score by reversing this process. The claim then follows from the fact that a minimum-weight spanning tree of a graph can be computed in time $\mathcal{O}(|V| \cdot \log |V|)$. $\qquad \square$

This coincides with the intuitive expectation that learning simple, more restricted networks could be easier than learning general networks. We conclude our exposition with an example showcasing that this is not true in general when comparing PL to BNSL. Grüttemeier et al. [24] recently showed that $\text{PL}^{\neq 0}$ is $\mathsf{W}[1]$-hard when parameterized by the number of *dependent vertices*, which are vertices with non-empty sets of candidate parents in the non-zero representation. For $\text{BNSL}^{\neq 0}$ we can show:

**Theorem 21.** $\text{BNSL}^{\neq 0}$ *is fixed-parameter tractable when parameterized by the number of dependent vertices.*

*Proof.* Consider an algorithm $\mathbb{B}$ for $\text{BNSL}^{\neq 0}$ which proceeds as follows. First, it identifies the set $X$ of dependent vertices in the input instance $\mathcal{I} = (V, \mathcal{F}, \ell)$, and then it branches over all possible choices of arcs with both endpoints in $X$, i.e., it branches over each arc set $A \subseteq A_X$. This results in at most $3^{k^2}$ branches, where $k = |X|$. In each branch and for each vertex $x \in X$, it now finds the highest-scoring parent set among those which precisely match $A$ on $X$, i.e., it first computes $\Gamma_f^A(x) = \{\, P \in parentsets(x) \mid \forall w \in X \setminus \{x\} : w \in P \iff wp \in A \,\}$ and then computes $\mathtt{score}^A(x) = \max_{P \in \Gamma_f^A(x)}(f_x(P))$. It then compares $\sum_{x \in X} \mathtt{score}^A(x)$ to $\ell$; if the former is at least as large as the latter in at least one branch then $\mathbb{B}$ outputs "Yes", and otherwise it outputs no.

The runtime of this algorithm is upper-bounded by $\mathcal{O}(3^{k^2} \cdot k \cdot |\mathcal{I}|)$. As for correctness, if $\mathcal{I}$ admits a solution $D$ then we can construct a branch such that $\mathbb{B}$ will output "Yes": in particular, this must occur when $A$ is equal to the arcs of the subgraph of $D$ induced on $X$. On the other hand, if $\mathbb{B}$ outputs "Yes" for some choice of $A$, we can construct a DAG $D$ with a score of at least $\ell$ by extending $A$ as follows: for each $x \in X$ we choose a parent set $P \in \Gamma_f^A(x)$ which maximizes $f_x(P)$ and we add arcs from each vertex in $P \setminus X$ to $x$. The score of this DAG will be precisely $\sum_{x \in X} \mathtt{score}^A(x)$, which concludes the proof. $\square$

# 6 Concluding Remarks

Our results provide a new set of tractability results that counterbalance the previously established algorithmic lower bounds for BAYESIAN NETWORK STRUCTURE LEARNING and POLYTREE LEARNING on "simple" superstructures. In particular, even though the problems remain $\mathsf{W}[1]$-hard when parameterized by the vertex cover number of the superstructure [36, 24], we obtained fixed-parameter tractability and a data reduction procedure using the feedback edge number and its localized version. Together with our lower-bound result for treecut width, this completes the complexity map for BNSL w.r.t. virtually all commonly considered graph parameters of the superstructure. Moreover, we showed that if the input is provided with an additive representation instead of the non-zero representation considered in previous theoretical works, the problems admit a dynamic programming algorithm which guarantees fixed-parameter tractability w.r.t. the treewidth of the superstructure.

This theoretical work follows up on previous complexity studies of the considered problems, and as such we do not claim any immediate practical applications of the results. That being said, it would be interesting to see if the polynomial-time data reduction procedure introduced in Theorem 3 could be adapted and streamlined (and perhaps combined with other reduction rules which do not provide a theoretical benefit, but perform well heuristically) to allow for a speedup of previously introduced heuristics for the problem [43, 42], at least for some sets of instances.

Last but not least, we'd like to draw attention to the *local feedback edge number* parameter introduced in this manuscript specifically to tackle BNSL. This generalization of the feedback edge set has not yet been considered in graph-theoretic works; while it is similar in spirit to the recent push towards measuring the so-called *elimination distance* of a graph to a target class, it is not captured by that notion. Crucially, we believe that the applications of this parameter go beyond BNSL; all indications suggest that it may be used to achieve tractability also for purely graph-theoretic problems where previously only tractability w.r.t. fen was known.