# OpenReview forum: "The Complexity of Bayesian Network Learning: Revisiting the Superstructure"
_NeurIPS.cc/2021/Conference — NeurIPS 2021 Oral_

### Official Review · Reviewer_3mFx · 2021-07-10

**Rating:** 7
**Confidence:** 4

**Summary:**

This paper investigates the fixed-parameter tractability of the Bayesian Network Structure Learning (BNSL) problem, according to new parameters related to the network superstructure. Namely, since the BNSL problem is known to be W[1]-hard for various vertex-related parameters, the authors concentrate on edge-related parameters, by showing that fixed-parameter tractability can indeed be achieved in some cases. In doing so, the authors consider two parameters, the feedback edge number (fen) and its “local” version (lfen), which are less restrictive than the tree-cut width (tcw). Orthogonally, the authors examine three families of scoring (function) representations: non-zero representations, additive representations, and bounded additive representations. Finally, the authors explore the fixed-parameter tractability of polytrees.



**Ethical Concerns:**

Not Applicable.

**Limitations And Societal Impact:**

Limitations: Yes to some extent. As mentioned in the review, the paper is providing both positive and negative FPT results for related parameters, and this helps in circumscribing the theoretical aspects of the BNSL problem. On the other hand, the paper could be further improved by presenting concrete complexity results.

**Main Review:**

Overall, this paper is well-motivated, well-written, and its technical quality is remarkable. As far as I could check, the paper is well-positioned with respect to the related work, and the main proofs are well-detailed. Importantly, positive FPT results are balanced against negative ones, which together provide natural limitations of this study.

Unsurprisingly, the main concern with such a paper is the applicability of theoretical results. In fact, the lack of “immediate” practical applications is a limitation emphasized by the authors in the conclusion. Yet, I think that the paper could be improved in order to have a better insight into the pragmatic aspects of the theoretical results. Namely, instead of just claiming that “Problem X is FPT …”, I would suggest providing a detailed complexity result. For example, using the kernelization result in Theorem 3, together with an upper bound on the number of directed acyclic subgraphs (for the complexity of the brute-force enumeration algorithm in Lines 218-219), we would get a result in $n^{O(1)} 2^{O(k^2)}$. On the other hand, even if the “lfen” metric is more restricted than the “fen” one, it seems that in this case, the term in the exponential is in $O(k^3)$. So, for the moment, unless the superstructure of a BNSL instance has very small global/local feedback edge numbers, the practical aspects of FPT algorithms are questionable. Based on the sketch of proof of Theorem 11 (number of snapshots), it seems that a similar consideration holds for the treewidth parameter in the setting of additive scores.

Minor comments:
* Lines 135-136: The sentence “Let $\Gamma_f(v)$ be the set of candidate parents of $v$ [...]” is a bit ambiguous. I would suggest “Let $\Gamma_f(v)$ be the collection of candidate parent-sets of $v$ [...]”.
* Line 178: “caardinality” -> “cardinality”.


**Time Spent Reviewing:**

4

---

> ### Author Response · Authors · 2021-08-04
> **Response to Reviewer 3mFx**
>
> Thank you for your positive assessment. Based on your suggestion, we will make sure that the obtained runtime bounds for each of the algorithms are clearly stated in the final version, and we will of course fix all of the minor issues you identified.

---

### Official Review · Reviewer_9zsF · 2021-07-14

**Rating:** 7
**Confidence:** 2

**Summary:**

Summary.
The paper is about the parameterized complexity of Bayesian Network Structure Learning.
In particular, it is focused to analyze and develop previous contributions on superstructure of the input.
The author/s show/s that changing the parametrization yields to achieve fixed-parameter tractability.
The paper also anallyzes how the complexity of structural learning of bayesian networks depends on the input representation, with specific reference to non-zero representation, and additive representation.
Finally the paper shows how to extend the main contirbutions presented to the problem of Polytree Learning.

**Limitations And Societal Impact:**

Apparently the paper clarifies that the achievements are limited and suggests further developments.
In particular, the paper states that even if fixed-parameter tractability is obtained and a data reduction procedure using the feedback edge number and its localized version are provided, the problems remain W[1]-hard when parameterized by the vertex cover number of the superstructure.

**Main Review:**

Originality:
I'm under the impression that the paper is original because it proposes and investigates a new parametrization for solving the problem of structural learning in Bayesian networks, i.e., it investigates how such a parametrization impact on tractability of the problem of structural learning of bayesian networks.

Quality:
The paper is well structured and written in my humble opinion. SOme terms and wuantities are not defined in the paper but are left to reader to find them on reference books. Theoretical results seem to be interesting but it is difficult for me, not being specifically working in thei very specific subarea to fully appreciate how such results can bring practical benefit to solve the problem of learning bayesian network structure.
Personally, I think that this type of papers, containing many theorems with proof left to the supplementary material are very difficult to manage having just few weeks.

Clarity:
The paper is somewhat clear, even if my marginal knowledge of the research subject makes me unsecure about this.
I think that adding some practical example would have improved a lot the readability of the paper and had help the interested reader to better appreciate the quality of the work.

Significance:
The paper is significant to provide a different research direction to analyze the ocmplexity of lstructural learning of bayesian networks. Also in this case I think that some numerical experiments would have better clarified the significance and relevance of the different contributions described in the paper.

**Time Spent Reviewing:**

16

---

> ### Author Response · Authors · 2021-08-04
> **Response to Reviewer 9zsF**
>
> We are glad that you found the paper to be well structured and well written. We will try to add an additional example to help make the paper more accessible, but cannot promise this due to space constraints.

---

### Official Review · Reviewer_6TGC · 2021-07-15

**Rating:** 7
**Confidence:** 3

**Summary:**

This is a paper about the parametrized complexity of BN structural learning. To achieve that the authors consider the learning task as a decision problem and distinguish tasks wrt the way the local scores are represented (non-zero, additive, additive with bound on the number of parents).
Separately for each class, the authors discuss tractability wrt various parameters (and not only the treewidth) used to describe the superstructure. This in particular suggests a dynamic programming approach for the additive case that is fixed-parameter tractable wrt the treewidth. Those results are also specialized to the case of learning polytree-shaped models.



**Ethical Concerns:**

Nothing.

**Limitations And Societal Impact:**

Nothing.

**Main Review:**

This is a theoretical paper that might improve the understanding of the computational complexity of BN structural learning. I think the paper should be accepted. As also noticed by the authors, the work is very theoretical and using these result to practically implement algorithmic schemes might not be obvious. Yet, some more discussion in this direction could help as well as a quick taxonomy of the scores used in the literature wrt the categories considered by the authors.

**Time Spent Reviewing:**

1

---

> ### Author Response · Authors · 2021-08-04
> **Response to Reviewer 6TGC**
>
> Thank you for your positive review. The paper indeed aims at expanding our theoretical understanding of the complexity of BNSL. We will try to add a further discussion regarding the score functions that were considered in the literature.

---

### Official Review · Reviewer_F8cX · 2021-07-16

**Rating:** 8
**Confidence:** 4

**Summary:**

This paper contains a number of significant FPT results for BNSL, as
well as providing a concise summary of existing FPT results. A key
parameter considered is "local feedback edge number" which provides
FPT. Results for when local scores are additively decomposable are
given. The paper has many results and so many proofs are relegated to
the supplementary material.

**Limitations And Societal Impact:**

These issues are not addressed.

**Main Review:**

The paper is well-written and contains a number of significant
theoretical results. It will be interesting to see how/whether these
results can be used to inform better BNSL solving, but that is a
question for a different paper.

One can remove (i.e. not store) any parent set which has a proper
subset with a score that is not lower. Not storing zero scores is just
a special case of this, why not consider the more general case?

Concerning the additive representation: it is no surprise that *if* a
local score admits an additive representation then "BNSL becomes
significantly easier" (and thus more positive FPT results are
available). However, it is something of an understatement for the
authors to say that "not every family of local score functions admits
an additive representation", the only examples the authors give of local
scores which are "additive" are those which are approximations to
"real score functions". (I too don't know of any others.) So basically
this boils down to the fact that BNSL problems with real
(statistically-motivated) local scores are a lot easier to solve
approximately than exactly. Having tractability results for (exactly)
solving an approximation is of some use, but one needs to consider
also what is lost by the approximation.

The instances the authors consider, I=(V,F,l), have the parent sets F
explicitly given, which, as the authors state, could lead to very
large instances. It would be interesting to extend this line of work to
where the parent sets, rather than being precomputed in order to
create the instance, are supplied by some function which supplies
parent sets on demand. Evaluating this function might take, say, O(np)
where n is the number of datapoints and p-1 the size of the parent set.
(Of course, finding the superstructure would be more involved in this
setting.)


157: fix "dependencey"
178: fix "caardinality"

**Time Spent Reviewing:**

2

---

> ### Author Response · Authors · 2021-08-04
> **Response to Reviewer F8cX**
>
> We were very happy to read that you liked the presented results. We will, of course, fix the typos you mention.
>
> Regarding the removal of parent sets, you are completely correct. We adopted the non-zero representation from previous work, but one can indeed strengthen it by not storing any parent set which has a proper subset with a score that is not lower. In essence, this preprocessing step widens the possible application scenarios of the obtained algorithms (since the representations become smaller), and we will add a remark about this in the final version.
>
> The case where parent sets are not explicitly given with scores but rather via, e.g., an oracle is indeed an interesting direction for future work, and we will add a remark about this in the concluding remarks.

---

> > ### Comment · Reviewer_F8cX · 2021-08-30
> > **response to author reponse**
> >
> > Thanks for your response and deciding to add the remarks you mention. I don't think we have any real disagreement to discuss.

---

### Decision · Program_Chairs · 2021-09-27

**Decision:**

Accept (Oral)

**Comment:**

There is clear consensus among all involved in the assessment. I would be very glad to see the talk about this paper myself too. Please do take suggestions in consideration.